# Understanding the Generalization of In-Context Learning in Transformers: An Empirical Study

**Xingxuan Zhang**[†], **Haoran Wang**[†], **Jiansheng Li, Yuan Xue, Shikai Guan, Renzhe Xu**
**Hao Zou, Han Yu, Peng Cui***

Tsinghua University, [†]Equal Contribution, [*]Corresponding Author
`xingxuanzhang@hotmail.com`, `wang-hr22@mails.tsinghua.edu.cn`,
`cuip@tsinghua.edu.cn`

## Abstract

Large language models (LLMs) like GPT-4 and LLaMA-3 utilize the powerful in-context learning (ICL) capability of Transformer architecture to learn on the fly from limited examples. While ICL underpins many LLM applications, its full potential remains hindered by a limited understanding of its generalization boundaries and vulnerabilities. We present a systematic investigation of transformers' generalization capability with ICL relative to training data coverage by defining a task-centric framework along three dimensions: inter-problem, intra-problem, and intra-task generalization [1]. Through extensive simulation and real-world experiments, encompassing tasks such as function fitting, API calling, and translation, we find that transformers lack inter-problem generalization with ICL, but excel in intra-task and intra-problem generalization. When the training data includes a greater variety of mixed tasks, it significantly enhances the generalization ability of ICL on unseen tasks and even on known simple tasks. This guides us in designing training data to maximize the diversity of tasks covered and to combine different tasks whenever possible, rather than solely focusing on the target task for testing.

## 1 Introduction

Transformers (Vaswani et al., 2017) have become foundational architecture for most modern large language models (LLMs), such as GPT-3 (Brown et al., 2020), GPT-4 (Achiam et al., 2023), Gemini (Team et al., 2023) and LLaMA (Dubey et al., 2024). One of transformer's impressive capabilities, in-context learning (ICL), empowers LLMs with the remarkable ability to learn on the fly from limited test-time examples. While in-context learning forms the bedrock of numerous LLM capabilities and applications, a key obstacle to harnessing ICL safely and reliably lies in our limited understanding of its origins, limitations, and generalizability.

The intricate interplay of ICL with the immense and unstructured nature of raw text data in LLMs poses a substantial analytical hurdle. To elucidate the mechanisms and capabilities of in-context learning, researchers have explored its potential beyond the confines of language domains and transcended the raw nature of text data. For example, Garg et al. (2022) proposes a novel training paradigm where GPT-2 family models are trained on prompts enriched with both input and label demonstrations, which empowers the models to emulate the behavior of the ordinary least squares (OLS) algorithm. Ahuja & Lopez-Paz (2023) investigate the generality and limitations of transformers' in-context learning for linear regression by contrasting them with set-based MLPs under distribution shift, revealing stronger resilience and OLS-like performance for transformers while highlighting vulnerabilities at extremes. Yadlowsky et al. (2023) exhibit vulnerabilities and suffer generalization limitations for functions outside training domains, suggesting ICL capabilities are heavily reliant on training data coverage rather than generalizable inductive biases.

Although strides have been made in elucidating the mechanisms of in-context learning, key knowledge gaps still remain. Notably, a systematic investigation into the relationship between the boundary of

---

[1]Code is available at https://github.com/UbeCc/Generalization-of-Transformers

in-context learning's generalization capabilities and its training data is lacking. Furthermore, the robustness of in-context learning within the generalization boundary under distribution shift has yet to be thoroughly explored. Addressing these critical gaps is essential for comprehensively understanding the potential and limitations of this powerful learning paradigm.

We use a framework that abstracts training data into distinct problems with nested tasks. By leveraging this task-centric representation, we aim to systematically investigate the generalization of in-context learning along 3 dimensions: (1) inter-problem generalization, the ability to learn and perform on entirely new problems not encountered during training, (2) intra-problem generalization, the capacity to fully grasp familiar tasks after receiving minimal, targeted knowledge in training as guidance, and (3) intra-task generalization, the ability to perform on the trained tasks with similar test samples. We design a series of experiments to investigate generalization along these 3 dimensions. To ensure the consistency of experimental conclusions with the dimension distinctions and the validity of comparisons, we conduct function-fitting experiments on models including GPT-2 (Radford et al., 2019) and LLaMA-3 (Dubey et al., 2024), as well as real-world experiments involving Tool calling and translation tasks on LLaMA-2 (Touvron et al., 2023) and Qwen-2 (Yang et al., 2024).

Our findings are as follows.

1. Transformers fail to show inter-problem generalization yet succeed in achieving strong intra-problem generalization during ICL. Even with minimal, targeted knowledge in training as guidance, the model learns to generalize to other tasks within the problem.

2. Contrary to expectations, an abundance of pretraining data does not necessarily confer upon a model the capacity of inter-problem generalization. Our experiments with LLaMA-3 reveal that similar to a randomly initialized GPT-2 model, the model struggles to generalize across problems despite demonstrating exceptional intra-problem and intra-task generalization capabilities with ICL.

3. In real-world experiments, incorporating a combination of different tasks into the training data not only improves the generalization ability of the model's ICL on unseen tasks, but in some scenarios, it can even enhance performance on known, simple tasks.

4. In real-world experiments, the inclusion of a wider variety of tasks in the training data can amplify the improvement achieved by ICL on complex or hard tasks. Even if the target task is not present in the training data, a model exposed to a greater diversity of tasks during training can quickly learn novel tasks through ICL and achieve better performance.

## 2 PRELIMINARIES

### 2.1 NOTATIONS

Define $f(x; \phi)$ as a mapping from $\mathbb{R}$ to $\mathbb{R}$, parameterized by $\phi$. A function class $\mathcal{F}$, is specified as a tuple $(f, \Phi)$, encompassing all functions parameterized by elements from the set $\Phi$, that is, $\mathcal{F} = (f, \Phi) = \{f(x; \phi) : \phi \in \Phi\}$. Let $\mathcal{G}$ denote the set comprising all such function classes $\mathcal{F}$.

**Definition 2.1** (In-context Learning Task). For a *given function class* $\mathcal{F} \in \mathcal{G}$, in-context learning tasks involve presenting a model such that for any $f(x; \phi) \in \mathcal{F}$ and $x_i \in \mathbb{R}, i = 1, 2, \ldots, n + 1$, given a prompt sequence

$$s = (x_1, f(x_1; \phi), x_2, f(x_2; \phi), \ldots, x_n, f(x_n; \phi), x_{n+1}), \tag{1}$$

the model could accurately predict $\hat{y}$, an estimate of $f(x_{n+1}; \phi)$.

**Definition 2.2** (In-context Learning Problem). For a *given set of function classes* $\mathcal{T} \subseteq \mathcal{G}$, in-context learning problem requires a model, such as a transformer, to successfully address all in-context learning tasks corresponding to each $\mathcal{F} \in \mathcal{T}$.

Furthermore, we define $\mathcal{T}_{base}$ and $\mathcal{T}_{com}$ intuitively in order of increasing complexity. We define $\mathcal{T}_{base}$ as the set of base functions, and $\mathcal{T}_{com}$ as the set of combinations of multiple base functions from $\mathcal{T}_{base}$. Subsequently, we present the generalization task and problem defined by $\mathcal{F}$ and $\mathcal{T}$.

### 2.2 RESEARCH QUESTIONS ON IN-CONTEXT LEARNING GENERALIZATION

Our objective is to investigate the generalization capability of models in in-context learning at various levels. To this end, we pose the following specific research questions.

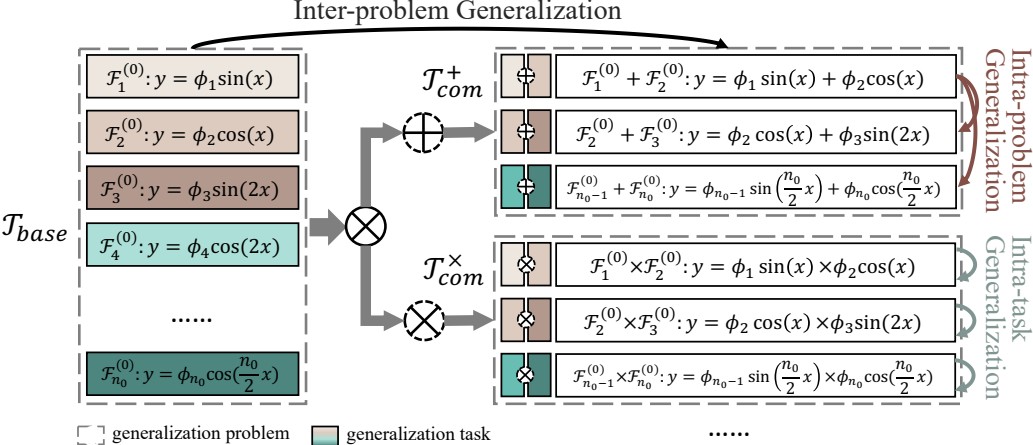

Figure 1: Illustration of inter-problem, intra-problem, and intra-task generalization. Here we use single function fitting as the base problem $\mathcal{T}_{base}$, and addition and multiplication as the combination problems $\mathcal{T}_{com}$ for example.

**Research Question 2.1** (Inter-problem Generalization)**.** Given two distinct in-context learning problems $\mathcal{T}_{base}, \mathcal{T}_{com} \subseteq \mathcal{G}$, where $\mathcal{T}_{base} \neq \mathcal{T}_{com}$, can a model trained on tasks from $\mathcal{T}_{base}$ generalize to tasks in $\mathcal{T}_{com}$?

**Research Question 2.2** (Intra-problem Generalization)**.** Given two different in-context learning problems $\mathcal{T}_{base}, \mathcal{T}_{com} \subseteq \mathcal{G}$, where $\mathcal{T}_{base} \neq \mathcal{T}_{com}$, and a subset of tasks $\mathcal{F}_1, \mathcal{F}_2, \ldots, \mathcal{F}_k \in \mathcal{T}_{com}$, if the model is trained on tasks from both $\mathcal{T}_{base}$ and this subset $\{\mathcal{F}_1, \mathcal{F}_2, \ldots, \mathcal{F}_k\}$ of $\mathcal{T}_{com}$, can it generalize to all the tasks in $\mathcal{T}_{com}$?

**Research Question 2.3** (Intra-task Generalization)**.** Given an in-context learning task $\mathcal{F}$ and a set of functions $\{f(x; \phi_1), f(x; \phi_2), \ldots, f(x; \phi_n)\} \in \mathcal{F}$, if a model is trained on these specific tasks, can it generalize to encompass all the functions in $\mathcal{F}$?

To facilitate understanding of the aforementioned definitions and research questions, we provide concrete definitions of generalization problem and generalization task based on function fitting tasks below for example.

## 2.3 CONSTRUCTING SPECIFIC IN-CONTEXT LEARNING TASKS AND PROBLEMS

As shown in Figure 1, to evaluate models' capability concerning Research Questions 2.1, 2.2, and 2.3, we devise specific in-context learning tasks and problems based on function fitting tasks.

**An Example of Base Function In-context Learning Problem** We select a set of $n_0$ base function classes $\{\mathcal{F}_1^{(0)}, \mathcal{F}_2^{(0)}, \ldots, \mathcal{F}_{n_0}^{(0)}\}$, constituting the base function in-context learning problem $\mathcal{T}_{base}$. These function classes are defined as:

$$\begin{aligned} \mathcal{F}_1^{(0)} &= \{\phi \sin(x) : \phi \in \mathbb{R}\}, & \mathcal{F}_2^{(0)} &= \{\phi \cos(x) : \phi \in \mathbb{R}\}, \\ \mathcal{F}_3^{(0)} &= \{\phi \sin(2x) : \phi \in \mathbb{R}\}, & \mathcal{F}_4^{(0)} &= \{\phi \cos(2x) : \phi \in \mathbb{R}\}, \cdots . \end{aligned} \tag{2}$$

We use sinusoidal functions as base functions and leverage the inherent advantage of orthogonality possessed by them to verify the model's ability to accurately fit them. This orthogonality property ensures that sinusoidal functions of different frequencies are mutually perpendicular in function space, minimizing interference and preventing inaccurate evaluation due to function overlap.

**Additive/Multiplicative/Compositional Combination In-context Learning Problems** From $\mathcal{T}_{base}$, we generate more complex problems using combination operations on functions. Specifically, for any in-context learning tasks $\mathcal{F}_1$ and $\mathcal{F}_2$, we define the additive, multiplicative, and compositional

Table 1: SE of models tested on the convex combinations of base functions. *Random* indicates the SE of random sampling within the function's codomain and serves as a measurement of SE. *Range* indicates the range in each sequence to calculate SE. $s(\cdot)$ and $c(\cdot)$ represent $\sin(\cdot)$ and $\cos(\cdot)$, respectively. $Mean_B$ and $Mean_C$ indicate the mean SE of base functions and combinations of base functions, respectively. $Com_{All}$ indicates the combination of all base functions.

| Range | Model | $s(x)$ | $c(x)$ | $s(2x)$ | $c(2x)$ | $Mean_B$ | $s(x)\&c(x)$ | $s(x)\&s(2x)$ | $s(x)\&c(2x)$ | $c(x)\&s(2x)$ | $c(x)\&c(2x)$ | $s(2x)\&c(2x)$ | $Com_{All}$ | $Mean_C$ |
|---|---|---|---|---|---|---|---|---|---|---|---|---|---|---|
| 1-10 | Baseline | **4.38e-3** | **2.43e-3** | 5.34e-3 | 6.49e-3 | 4.66e-3 | **6.01e-3** | 6.18e-2 | 8.93e-2 | 8.96e-2 | 4.39e-2 | 1.25e-2 | 2.18e-1 | 7.96e-2 |
|  | CFL | 5.20e-3 | 3.50e-3 | **3.61e-3** | **4.55e-3** | **4.22e-3** | 1.35e-2 | **7.37e-3** | **1.59e-2** | **4.80e-2** | **2.86e-2** | **1.05e-2** | **3.09e-2** | **2.21e-2** |
| 11-20 | Baseline | **1.38e-5** | **9.77e-6** | **3.38e-5** | **5.58e-5** | **2.85e-5** | **2.20e-3** | 6.65e-2 | 8.44e-2 | 8.43e-2 | 2.04e-2 | 5.63e-3 | 2.06e-1 | 6.71e-2 |
|  | CFL | 2.78e-5 | 1.01e-5 | 7.81e-5 | 8.12e-5 | 4.93e-5 | 5.30e-3 | **1.76e-4** | **9.22e-4** | **1.39e-2** | **4.41e-3** | **1.88e-3** | **4.05e-3** | **4.39e-3** |
| 21-30 | Baseline | **7.98e-6** | **6.62e-6** | 3.44e-5 | 4.40e-5 | 2.36e-5 | **1.86e-3** | 4.25e-2 | 6.90e-2 | 5.59e-2 | 9.57e-3 | 4.92e-3 | 2.07e-1 | 5.58e-2 |
|  | CFL | 2.13e-5 | 6.78e-6 | 7.41e-5 | 6.70e-5 | 4.23e-5 | 3.13e-3 | **1.29e-4** | **3.46e-4** | **6.20e-3** | **1.41e-3** | **7.43e-4** | **1.23e-3** | **1.88e-3** |
| 31-40 | Baseline | **8.84e-6** | 6.60e-6 | 3.42e-5 | 4.32e-5 | 2.39e-5 | **1.71e-3** | 3.94e-2 | 6.08e-2 | 4.70e-2 | 7.82e-3 | 4.77e-3 | 1.84e-1 | 4.94e-2 |
|  | CFL | 1.87e-5 | **6.02e-6** | 7.32e-5 | 6.58e-5 | 4.12e-5 | 2.66e-3 | **1.17e-4** | **2.27e-4** | **5.27e-3** | **9.45e-4** | **5.38e-4** | **6.96e-4** | **1.49e-3** |
| - | Random | 4.03e-2 | 4.27e-2 | 4.22e-2 | 4.14e-2 | 4.17e-2 | 9.27e-2 | 9.80e-2 | 9.45e-2 | 1.01e-1 | 9.10e-2 | 9.72e-2 | 3.59e-1 | 1.56e-1 |

operations on tasks as:

$$\mathcal{F}_1 + \mathcal{F}_2 = \{x \mapsto f_1(x; \phi_1) + f_2(x; \phi_2) : f_1 \in \mathcal{F}_1, f_2 \in \mathcal{F}_2\},$$
$$\mathcal{F}_1 \times \mathcal{F}_2 = \{x \mapsto f_1(x; \phi_1) \times f_2(x; \phi_2) : f_1 \in \mathcal{F}_1, f_2 \in \mathcal{F}_2\}, \tag{3}$$
$$\mathcal{F}_1 \circ \mathcal{F}_2 = \{x \mapsto f_1(f_2(x; \phi_2); \phi_1) : f_1 \in \mathcal{F}_1, f_2 \in \mathcal{F}_2\}.$$

For an operator $\diamond \in \{+, \times, \circ\}$, the respective in-context learning problem is formulated as:

$$\mathcal{T}_{com}^{\diamond} = \{\mathcal{F}_1 \diamond \mathcal{F}_2 : \mathcal{F}_1, \mathcal{F}_2 \in \mathcal{T}_{base}\}. \tag{4}$$

The problems $\mathcal{T}_{com}^{+}$, $\mathcal{T}_{com}^{\times}$, and $\mathcal{T}_{com}^{\circ}$ are thus termed as additive, multiplicative, and compositional combination in-context learning problems, respectively.

## 3 GENERALIZATION OF ICL ON FUNCTION FITTING

In this section, we investigate the generalization ability of ICL on function-fitting tasks and study the research questions raised in the previous section.

Previous studies (Garg et al., 2022; Akyürek et al., 2022) have demonstrated the ability of transformers to select the correct function from one or two function classes based on in-context examples. However, their ability to generalize to convex combinations of these learned functions within the training data remains limited (Yadlowsky et al., 2023). This difficulty arises because convex combinations constitute a new function class, requiring additional knowledge beyond solely the constituent functions. Moreover, real-world applications rarely involve separate training and test regimes with distinct function composition patterns. Instead, models are typically trained and deployed on data containing diverse, intermingled function (task) combinations. Hence, our investigation focuses on the more practical and generalizable scenario: the ability of transformers to extrapolate from specific function combinations to new, unseen combinations.

In Sections 3.1, 3.2, and 3.3, We use randomly initialized GPT-2-Standard as our base model, which is a relatively small model enough to reveal the results. In Section 3.4, we utilize a pretrained LLaMA-3 model for finetuning on function fitting to investigate whether large-scale pretraining influences the generalization of ICL. Since inputs are not limited by natural language, we directly perform function fitting on GPT-2 at the embedding level to enable floating-point calculations. Large language models map tokenized sentences into embeddings and then perform computations at the same level. Therefore, the underlying principle of function fitting is naturally consistent with the behavior of large language models. Our empirical results further confirm this connection.

### 3.1 GENERALIZATION TO CONVEX COMBINATIONS OF LEARNED FUNCTIONS

In the context of function approximation, convex combinations serve as the basis for representing functions. Here we explore Question 2.1, 2.2, and 2.3 with additive combination ICL problems, i.e., $\mathcal{T}_{com}^{+}$. In our base setting, we use four base functions, i.e., $n_0 = 4$ unless otherwise stated.

We denote the model trained on the convex combination of functions as the ComFuncLearner, while the model trained without convex combinations serves as the Baseline. The ComFuncLearner model leverages all base functions in its training alongside a specific convex combination of $\mathcal{F}_1^{(0)}$ and $\mathcal{F}_3^{(0)}$,

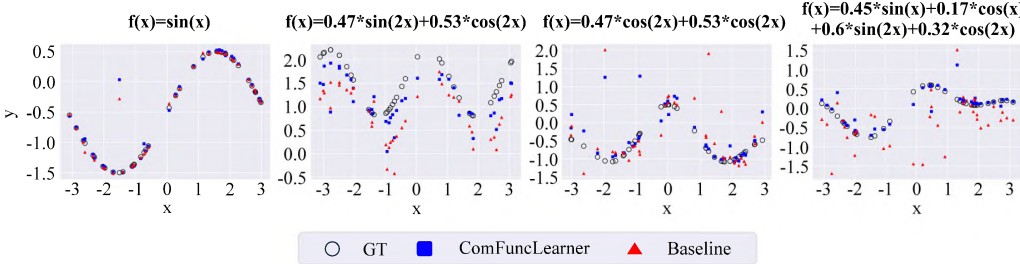

Figure 2: Function curves on the convex combinations of base functions fitted by the Baseline and ComFuncLearner models after ICL. *Com* stands for combination and the weights of base functions are randomly sampled and labeled on the figure.

i.e., $\mathcal{F}_1 + \mathcal{F}_3$. To maintain parity in the training set size and the number of seen functions, the Baseline model is trained on all base function classes and an additional one, $\mathcal{F}_5^{(0)} = \{\phi \sin(3x) : \phi \in \mathbb{R}\}$. Thus the Baseline model is trained on samples generated by functions sampled from $\{\mathcal{F}_1^{(0)}, \mathcal{F}_2^{(0)}, \mathcal{F}_3^{(0)}, \mathcal{F}_4^{(0)}, \mathcal{F}_5^{(0)}\}$ while the ComFuncLearner on $\{\mathcal{F}_1^{(0)}, \mathcal{F}_2^{(0)}, \mathcal{F}_3^{(0)}, \mathcal{F}_4^{(0)}, \mathcal{F}_1^{(0)} + \mathcal{F}_3^{(0)}\}$. The performance of the Baseline model on base function classes answers the Question 2.3 intra-task generalization, while its performance on combinations of base function classes with $\mathcal{T}_+$ answers the Question 2.1 inter-problem generalization. The performance of the ComFuncLearner on combinations of base function classes with $\mathcal{T}_+$ answers the Question 2.2 intra-task generalization.

We report the average squared error (SE) of the predicted values compared to ground-truth labels and averaged across 128 independent runs. Each data point thus represents the average SE computed over 128 test samples. As shown in Table 1, when tested with functions sampled from familiar classes seen in the training data, both models attain low squared error (SE) after 10 in-context examples. However, when presented with unseen convex combinations, the ComFuncLearner model exhibits a remarkable improvement in generalization ability within the ICL paradigm, even with only a single exposure to a convex combination pattern during training. Specifically, the ComFuncLearner consistently outperforms the Baseline model across all convex combinations, except for the combination of $\mathcal{F}_1^{(0)} = \{\phi \sin(x) : \phi \in \mathbb{R}\}$ and $\mathcal{F}_2^{(0)} = \{\phi \cos(x) : \phi \in \mathbb{R}\}$. In this combination, both models achieve low SEs (less than 0.002 after 20 in-context examples). This observed convergence could potentially be due to the close functional similarity between the two base functions, making their individual contributions subtle within the convex mixture.

To gain deeper insights into the model's internal function representation, we employ a progressive visualization method. Following each exposure to N in-context examples (i.e., samples), we fully traverse the X-axis space, feeding each $x$ as the input for the N+1st sample and recording the predicted $y$. This approach effectively captures the model's evolving understanding of the underlying function as it accumulates training data. Figure 2 presents the function curves learned after 39 examples, revealing the gradual refinement of the model's predictions as it progressively learns function. Please see Appendix D.1 for more visualization results.

For the individual basis functions, both models closely approximate the true curves across the entire X-axis range. However, for the convex combinations, the Baseline model exhibits significantly poorer performance compared to the ComFuncLearner model. While the ComFuncLearner accurately approximates almost all convex combinations, the Baseline model's fits are often suboptimal. Please note that the ComFuncLearner model does not learn the combination of $\mathcal{F}_3^{(0)} = \{\phi \sin(2x) : \phi \in \mathbb{R}\}$ and $\mathcal{F}_4^{(0)} = \{\phi \cos(2x) : \phi \in \mathbb{R}\}$, and the combination of all base functions. Thus this indicates that **when we select convex combination as the operation on base functions, the transformers show intra-problem generalization** (the ComFunclearner model fits the combinations it never learned in the training data) **yet fail to achieve inter-problem generalization** (the Baseline model fails to fit most combinations of base functions).

For a detailed experimental setup, please refer to Appendix B.1. Except for convex combinations, we also observe remarkably similar conclusions regarding the model's ability to learn multiplicative combinations of functions as shown in the following section.

Table 2: SE of the ComFuncLearner and Baseline model tested on the product combinations of base functions. *CFL1*, *CFL2* and *CFL4* indicate ComFuncLearner trained with 1, 2, and 4 combinations during training, respectively.

| Range | Model | s(x) | c(x) | s(2x) | c(2x) | Mean$_B$ | s(x)&c(x) | s(x)&s(2x) | s(x)&c(2x) | c(x)&s(2x) | c(x)&c(2x) | s(2x)&c(2x) | Com$_{All}$ | Mean$_C$ |
|---|---|---|---|---|---|---|---|---|---|---|---|---|---|---|
| 1-10 | Baseline | 3.50e-3 | 3.65e-3 | 5.34e-3 | 5.29e-3 | 4.45e-3 | 2.87e-2 | 1.89e-2 | 2.99e-2 | 2.10e-2 | 2.85e-2 | 1.44e-2 | 4.14e-2 | 3.21e-2 |
| | CFL1 | 4.04e-3 | 2.71e-3 | 4.01e-3 | 4.87e-3 | 3.91e-3 | 2.83e-2 | **5.90e-3** | 4.28e-2 | 1.45e-2 | 2.65e-2 | 1.98e-2 | 3.59e-2 | 2.48e-2 |
| | CFL2 | 4.85e-3 | 3.46e-3 | **2.35e-3** | **3.23e-3** | **3.47e-3** | **4.96e-3** | 2.31e-2 | 3.52e-2 | 1.37e-2 | 2.42e-2 | **4.85e-3** | **2.27e-2** | 1.84e-2 |
| | CFL4 | **3.42e-3** | **2.63e-3** | 4.13e-3 | 4.19e-3 | 3.59e-3 | 5.95e-3 | 5.94e-3 | **2.61e-2** | **1.35e-2** | **6.01e-3** | 7.16e-3 | 2.70e-2 | **1.31e-2** |
| 11-20 | Baseline | 4.54e-5 | 1.17e-5 | 2.13e-5 | 6.46e-5 | 3.61e-5 | 2.81e-2 | 9.07e-3 | 2.61e-2 | 1.35e-2 | 1.69e-2 | 7.65e-3 | 4.87e-2 | 2.52e-2 |
| | CFL1 | **2.77e-5** | 9.16e-6 | 4.04e-5 | **5.06e-5** | **3.23e-5** | 8.57e-3 | **9.40e-5** | 1.78e-2 | 5.78e-3 | 1.96e-2 | 3.23e-3 | 2.16e-2 | 1.10e-2 |
| | CFL2 | 3.53e-5 | **5.23e-6** | **1.93e-5** | 8.22e-5 | 3.61e-5 | **4.49e-5** | 1.22e-2 | **1.15e-2** | 8.44e-3 | 1.59e-2 | **1.30e-4** | **1.36e-2** | 8.85e-3 |
| | CFL4 | 4.08e-5 | 1.18e-5 | 4.34e-5 | 6.92e-5 | 4.12e-5 | 5.09e-5 | 1.88e-4 | 6.92e-3 | **5.27e-3** | **1.07e-3** | 2.55e-4 | 1.59e-2 | **4.11e-3** |
| 21-30 | Baseline | 3.28e-5 | 7.39e-6 | 1.58e-5 | 4.18e-5 | 2.43e-5 | 2.67e-2 | 7.69e-3 | 3.29e-2 | 8.85e-3 | 1.56e-2 | 5.70e-3 | 5.16e-2 | 2.13e-2 |
| | CFL1 | **1.94e-5** | 6.75e-6 | 3.23e-5 | 3.42e-5 | 2.38e-5 | 4.45e-3 | **5.41e-5** | 9.81e-3 | **3.05e-3** | 1.45e-2 | 1.68e-3 | 1.42e-2 | 6.83e-3 |
| | CFL2 | 2.48e-5 | **3.28e-6** | **1.17e-5** | 6.98e-5 | 2.73e-5 | 1.03e-2 | 8.66e-3 | 6.21e-3 | 1.35e-2 | 9.78e-5 | | **1.06e-2** | 7.05e-3 |
| | CFL4 | 2.33e-5 | 6.69e-6 | 1.56e-5 | **3.36e-5** | **2.03e-5** | **2.76e-5** | 1.16e-4 | **2.99e-3** | 3.39e-3 | **4.23e-5** | 1.08e-4 | 1.38e-2 | **2.94e-3** |
| 31-40 | Baseline | 2.52e-5 | 7.24e-6 | 1.61e-5 | 3.71e-5 | 2.17e-5 | 2.69e-2 | 7.76e-3 | 2.29e-2 | 8.86e-3 | 8.52e-3 | 5.21e-3 | 5.06e-2 | 1.98e-2 |
| | CFL1 | 1.82e-5 | 6.75e-6 | 3.65e-5 | 3.07e-5 | 2.31e-5 | 3.16e-3 | **4.99e-5** | 7.58e-3 | 2.84e-2 | 1.16e-2 | 1.59e-3 | 9.88e-3 | 5.24e-3 |
| | CFL2 | 2.44e-5 | **3.26e-6** | **1.13e-5** | 6.31e-5 | 2.61e-5 | 2.71e-5 | 8.78e-3 | 6.57e-3 | 5.52e-3 | 1.33e-2 | **7.83e-5** | 9.02e-3 | 6.21e-3 |
| | CFL4 | **1.79e-5** | 5.44e-6 | 1.25e-5 | **2.49e-5** | **1.57e-5** | **2.02e-5** | 9.73e-5 | **2.23e-3** | **2.77e-3** | **3.40e-5** | 8.77e-5 | **7.01e-3** | **2.24e-3** |
| - | Random | 4.25e-2 | 4.14e-2 | 4.68e-2 | 4.81e-2 | 4.47e-2 | 4.87e-2 | 5.01e-2 | 4.28e-2 | 4.59e-2 | 5.14e-2 | 5.28e-2 | 4.80e-2 | 1.98e-1 |

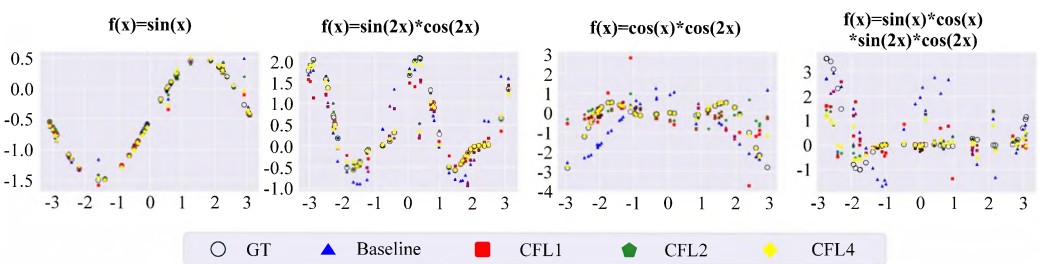

Figure 3: Function curves on the product combinations of base functions fitted by the Baseline and ComFuncLearner models after ICL.

## 3.2 Generalization to Products of Learned Functions

Moving beyond convex combinations, we explore the generalization of transformers when selecting product combinations (i.e., $\mathcal{T}_{com}^{\times}$) as the operation on base function classes. Unlike convex combinations, where the model typically aligns with one constituent function to maintain control over squared error (SE), product multiplication allows more freedom in fitting individual function classes, potentially leading to greater divergence from the product true curve. We also consider the combinations of $\mathcal{F}_1^{(0)} = \{\phi \sin(x) : \phi \in \mathbb{R}\}$ and $\mathcal{F}_3^{(0)} = \{\phi \sin(2x) : \phi \in \mathbb{R}\}$ as the extra function classes to train the ComFuncLearner model here.

Table 2 reveals the Baseline model's persistent struggle to accurately predict product combinations within the ICL. Despite having witnessed only one combinatorial pattern, the ComFuncLearner model consistently surpasses the Baseline in the combination problem. However, compared to the substantial improvement observed in convex combination experiments, the ComFuncLearner's advantage in product combinations is less pronounced. We hypothesize that this disparity arises from the inherent difficulty of fitting product combinations compared to convex combinations.

To investigate this further, we progressively increase the number of product combinations presented to the ComFuncLearner model in subsequent experiments (1, 2, 3, and 4). Specifically, we report the results of the ComFuncLearner model combinational function classes including $\{\mathcal{F}_1^{(0)} \times \mathcal{F}_3^{(0)}, \mathcal{F}_1^{(0)} \times \mathcal{F}_2^{(0)}, \mathcal{F}_2^{(0)} \times \mathcal{F}_4^{(0)}, \mathcal{F}_3^{(0)} \times \mathcal{F}_4^{(0)}\}$. Please note that we only add more patterns of combinations to the training yet the number of all training samples and training steps are fixed the same for the Baseline model, the ComFuncLearner model with 1, 2, and 4 combinations in its training.

This manipulation aims to determine whether exposure to a wider range of combinatorial patterns during training could enhance the model's generalization ability to unseen product combinations. As shown in Table 2, we observe a consistent decrease of SE as the number of product combination patterns presented to the ComFuncLearner model in the training increases. For example, we observe the mean SE among 21-30 points drop from 0.0213 (the Baseline model) to 0.00294 (the ComFuncLearner with 4 combination patterns in its training data). Figure 3 presents the function

Table 3: SE of the ComFuncLearner and Baseline model tested on the compositional combinations of base functions.

| Range | Model | s(x) | c(x) | s(3x) | s(x)&c(x) | Mean$_B$ | s(2x)&c(2x) | s(2x)&c(x) | s(2x)&s(3x) | s(3x)&c(x) | c(2x)&c(3x) | c(2x)&s(x) | c(2x)&s(2x) | Mean$_C$ |
|---|---|---|---|---|---|---|---|---|---|---|---|---|---|---|
| 1-10 | Baseline | 2.56e-3 | 2.94e-3 | 3.35e-3 | 4.19e-3 | 3.26e-3 | 3.09e-2 | 3.65e-2 | **3.48e-2** | 3.97e-2 | 3.91e-2 | 1.45e-2 | 3.99e-2 | 3.36e-2 |
|  | CFL1 | 2.50e-3 | **2.09e-3** | 2.85e-3 | **2.13e-3** | **2.39e-3** | 3.18e-2 | 3.69e-2 | 3.64e-2 | 3.89e-2 | 4.06e-2 | 1.29e-2 | 4.22e-2 | 3.43e-2 |
|  | CFL2 | **2.19e-3** | 2.52e-3 | 3.07e-3 | 2.25e-3 | 2.51e-3 | **2.99e-2** | 3.54e-2 | 3.69e-2 | 3.73e-2 | 4.24e-2 | 1.62e-2 | 4.07e-2 | 3.47e-2 |
|  | CFL4 | 2.69e-3 | 2.71e-3 | **3.58e-5** | 2.39e-3 | 2.84e-2 | 3.14e-2 | **2.96e-2** | 3.58e-2 | **3.39e-2** | **3.84e-2** | 1.19e-2 | **3.89e-2** | **3.11e-2** |
| 11-20 | Baseline | 6.41e-6 | 4.43e-6 | 1.69e-5 | 6.41e-5 | 2.34e-5 | 3.18e-2 | 3.83e-2 | 3.17e-2 | 3.56e-2 | 4.19e-2 | 7.55e-3 | 4.43e-2 | 3.30e-2 |
|  | CFL1 | 1.33e-5 | 4.69e-6 | 3.21e-5 | 5.93e-6 | 1.47e-5 | 3.43e-2 | 3.44e-2 | 3.19e-2 | 3.68e-2 | 4.61e-2 | 6.63e-3 | 4.36e-2 | 3.34e-2 |
|  | CFL2 | 1.64e-5 | **2.18e-6** | 2.97e-5 | 3.11e-6 | 1.43e-5 | 3.46e-2 | 3.49e-2 | 3.58e-2 | 4.08e-2 | 4.72e-2 | 7.71e-3 | 4.28e-2 | 3.48e-2 |
|  | CFL4 | **2.32e-6** | 2.29e-6 | **9.63e-6** | **2.19e-6** | **4.75e-6** | **2.27e-2** | **2.44e-2** | **2.69e-2** | **2.32e-2** | **3.04e-2** | **4.75e-3** | **2.94e-2** | **2.31e-2** |
| 21-30 | Baseline | 5.56e-6 | 4.20e-6 | 1.70e-5 | 6.40e-5 | 2.31e-5 | 2.99e-2 | 3.60e-2 | 2.87e-2 | 3.46e-2 | 4.01e-2 | 6.90e-3 | 4.49e-2 | 3.16e-2 |
|  | CFL1 | 1.29e-5 | 4.99e-6 | 3.31e-5 | 6.30e-6 | 1.43e-5 | 3.17e-2 | 3.23e-2 | 2.99e-2 | 3.89e-2 | 4.38e-2 | 6.10e-3 | 4.79e-2 | 2.48e-2 |
|  | CFL2 | 1.58e-5 | **1.73e-6** | 2.84e-5 | 2.44e-6 | 1.39e-5 | 3.29e-2 | 3.25e-2 | 3.35e-2 | 4.15e-2 | 4.81e-2 | 6.34e-3 | 4.50e-2 | 3.42e-2 |
|  | CFL4 | **1.51e-6** | 1.82e-6 | **8.89e-6** | **1.49e-6** | **4.35e-6** | **1.67e-2** | **1.94e-2** | **1.97e-2** | **1.79e-2** | **2.40e-2** | **3.85e-3** | **2.45e-2** | **1.80e-2** |
| 31-40 | Baseline | 5.43e-6 | 4.04e-6 | 1.72e-5 | 6.82e-5 | 2.43e-5 | 3.14e-2 | 3.66e-2 | 2.99e-2 | 3.85e-2 | 4.23e-2 | 7.54e-3 | 4.39e-2 | 3.29e-2 |
|  | CFL1 | 1.24e-5 | 4.65e-6 | 3.05e-5 | 6.18e-6 | 1.38e-5 | 3.28e-2 | 3.16e-2 | 3.02e-2 | 3.88e-2 | 4.64e-2 | 6.67e-3 | 4.53e-2 | 3.31e-2 |
|  | CFL2 | 1.45e-5 | **1.64e-6** | 2.97e-5 | 2.26e-6 | 1.34e-5 | 3.24e-2 | 3.35e-2 | 3.46e-2 | 3.94e-2 | 4.94e-2 | 7.12e-3 | 4.29e-2 | 3.42e-2 |
|  | CFL4 | **1.21e-6** | 1.96e-6 | **8.94e-6** | **1.43e-6** | **3.67e-6** | **1.43e-2** | **1.89e-2** | **1.75e-2** | **1.43e-2** | **2.17e-2** | **4.16e-3** | **1.96e-2** | **1.58e-2** |
| - | Random | 4.56e-2 | 4.81e-2 | 4.23e-2 | 4.54e-2 | 4.54e-2 | 6.23e-2 | 5.82e-2 | 6.40e-2 | 6.12e-2 | 5.99e-2 | 6.52e-2 | 6.85e-2 | 6.28e-2 |

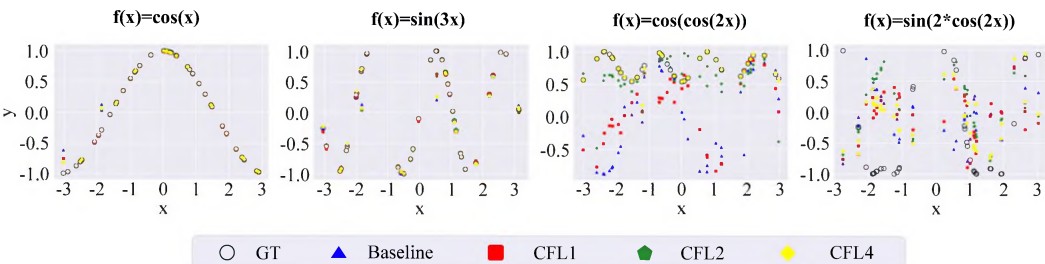

Figure 4: Function curves on the compositional combinations of base functions fitted by the Baseline and ComFuncLearner models after ICL.

curves learned after 39 examples and reveals a distinct trend: as the number of seen combinations in training increases, the model's fit to the objective function steadily improves. With 4 combinations observed, the ComFuncLearner model's fit closely approximates the function's true curve. This suggests that **transformers fail to achieve inter-problem generalization but show intra-problem generalization, which can be improved as the number of learned combinations increases**.

### 3.3 GENERALIZATION TO COMPOSITIONS OF LEARNED FUNCTIONS

We then investigate the generalization of transformers when selecting composition combinations as the operation on base functions. Compared to convex combinations and products of functions, compositional combinations (i.e., $\mathcal{T}_{com}^{\circ}$) present a significantly greater challenge due to a critical distinction: the distribution of input to the outer function is dynamically transformed by the inner function, creating additional complexity for analysis and optimization.

To further investigate the relationship between generalization capabilities and training patterns, we progressively increase the number (1, 2, 3, and 4) of compositional combinations presented to the ComFuncLearner model from combinational function classes including $\{\mathcal{F}_1^{(0)} \circ \mathcal{F}_3^{(0)}, \mathcal{F}_1^{(0)} \circ \mathcal{F}_2^{(0)}, \mathcal{F}_2^{(0)} \circ \mathcal{F}_4^{(0)}, \mathcal{F}_3^{(0)} \circ \mathcal{F}_4^{(0)}\}$ in subsequent experiments. CFL1 has seen the combination $\mathcal{F}_1^{(0)} \circ \mathcal{F}_3^{(0)}$. CFL2 has seen $\mathcal{F}_1^{(0)} \circ \mathcal{F}_3^{(0)}$ and $\mathcal{F}_1^{(0)} \circ \mathcal{F}_2^{(0)}$. While CFL4 has seen all four function pairs. As shown in Table 3 and Figure 4, While ComFuncLearner's improvement on compositional combinations is not as significant as on convex or product combinations compared to the Baseline, its SE decreases notably as the number of encountered combinations increases. We believe this is due to the nature of compositional combinations. Since the output of the inner function in a composite function serves as the input for the outer function, the input distribution for the outer function changes significantly compared to the non-composite case, making it difficult for the model to fit the composite function well. We discuss the impact of input distribution shifts on ICL generalization in Appendix C.

### 3.4 GENERALIZATION OF PRETRAINED LLaMA-3 ON FUNCTION FITTING

To investigate the impact of extensive pretraining data on model generalization, we finetune the pretrained LLaMA-3 model for function-fitting and evaluate its performance with the composition

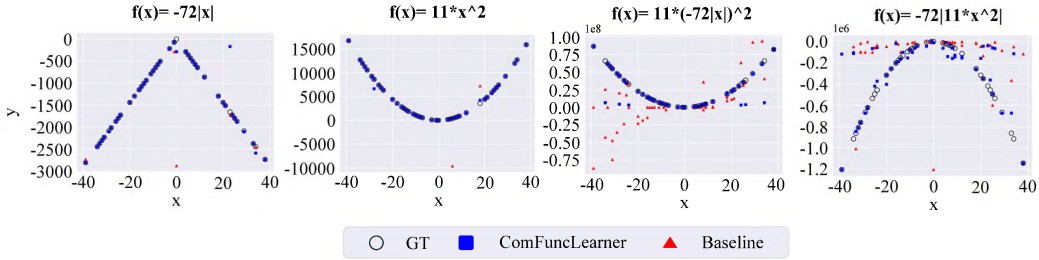

Figure 5: The fitted curves of LLaMa-3 on composition combinations of learned functions (*absolute value functions* $-|\mathbf{x}|$ and *quadratic function* $\mathbf{x^2}$).

combination of functions following the setting in Section 3.3. Detailed experimental procedures are shown in Appendix B.2.

As shown in Figure 5 and Appendix D.2, our findings provide compelling evidence that while finetuned LLaMA-3 exhibits exceptional proficiency in intra-problem and intra-task generalization, it notably struggles with inter-problem generalization. In more precise terms, the model's ability to learn via ICL is contingent upon prior exposure during finetuning and novel combinations of functions that are absent in the finetuning data cannot be acquired through ICL. However, if the model has been exposed to a limited number of combinatorial forms during finetuning, it demonstrates the capacity to generalize and learn other types of combinatorial forms via ICL. This implies that **despite its extensive pretraining on massive datasets, the model has not fully acquired the ability to transfer knowledge and apply learned patterns to dissimilar problem domains**, i.e., from question-answering problems to function-fitting. However, incorporating a small amount of function composition data into the finetuning dataset can stimulate the model's ability to handle new problems. This suggests that **the diversity of finetuning data and its representativeness for application tasks is crucial for activating the knowledge acquired by the model during pretraining**.

## 4 GENERALIZATION OF ICL ON REAL-WORLD EXPERITMENS

To ensure the real-world relevance of our findings, we investigate the generalization capabilities of transformers with ICL on complex, real-world tasks. Our primary focus is on tool-calling tasks for pretrained models and translation tasks for models trained from scratch. Unless otherwise stated, the results reported in this section are the average of three independent runs.

### 4.1 TOOL-CALLING

It is challenging to assess the generalization ability of LLMs on novel tasks, as they are typically pretrained and supervised finetuned (SFT) on massive text corpora. To address this, we selected tool-calling tasks, which differ significantly from the question-answering tasks commonly encountered by LLMs. Tool-calling tasks offer a high degree of controllability and comparability, making them ideal for delineating the boundaries of LLMs' generalization capabilities across problems and tasks where LLMs' initial performance has considerable room for improvement (Qin et al., 2023).

We categorized Tool-calling tasks into single-API calls and multi-API calls. Single-API calls refer to tasks that require only a single invocation to fulfill the instructions in the prompt, while multi-API calls necessitate considering the combined use of APIs to achieve more complex functionalities. Mirroring the definitions of generalization problem and task, we define single-API calls as the basic generalization problem $\mathcal{T}_{base}$, and the combination of APIs as the composite generalization problem $\mathcal{T}_{com}$. This allows the Baseline model to be trained only on the basic generalization problem, while ComFuncLearner is trained on both the basic and composite generalization problems. To ensure a fair comparison, we guarantee that the Baseline and ComFuncLearner are trained on the same number of samples and that ComFuncLearner has not encountered any of the API combination forms present in the test data. Further experimental details can be found in the Appendix B.3.

The results are in Table 4. When both are based on LLaMA-2 and trained on the same number of samples, incorporating ICL can easily surpass the results of ToolLLaMA (Qin et al., 2023). Compared to the Baseline model, ComFuncLearner with ICL demonstrates a significant boost in performance on

Table 4: API-calling accuracy on ToolBench. Following Qin et al. (2023), we report the results of I1-inst, I1-tool, I2-inst, I2-Cat, I3-Inst. During training, *I1, I2, I3* denote single-tool, intra-category multi-tool, and intra-collection multi-tool instructions. For inference, data is divided into levels: *Inst.* for new instructions on trained tools, *Tool* for new tools in known categories, and *Cat.* for new tools in new categories. Tools in the same *category* share a common theme (e.g., sports, finance), while those in the same *collection* have similar functionality (e.g., database manipulation, web crawling). *Baseline* refers to model trained with *single-tool* examples while *CFL* with *multi-tool* examples. $ICL_s$ means 16-shot evaluation with single-API examples and $ICL_m$ with multi-API examples. The difference highlighted refers to the improvement of ICL.

| Model | I1-Inst. | I1-Tool | I1-Cat. | Average$_s$ | I2-Inst. | I2-Cat. | I3-Inst. | Average$_m$ |
|---|---|---|---|---|---|---|---|---|
| ToolLLaMA[1] | 25.0 | 29.0 | 33.0 | 29.0 | 30.5 | 31.5 | 25.0 | 29.0 |
| Baseline | 24.2 | 27.5 | 32.6 | 28.1 | 26.2 | 25.6 | 20.6 | 24.1 |
| CFL | 25.2 | 28.7 | 32.8 | 28.9 | 30.5 | 31.7 | 24.8 | 29.0 |
| Baseline+ICL$_s$ | 26.9 | 29.3 | 33.4 | 29.8 (+1.8) | 27.0 | 26.6 | 22.5 | 25.4 (+1.3) |
| Baseline+ICL$_m$ | 26.6 | 28.5 | 33.6 | 29.5 (+1.4) | 28.1 | 27.5 | 23.6 | 26.3 (+2.2) |
| CFL+ICL$_s$ | **26.8** | 31.2 | **34.7** | **30.9** (+2.0) | 31.8 | 33.1 | 27.3 | 30.7 (+1.7) |
| CFL+ICL$_m$ | 26.2 | **31.3** | 33.4 | 30.3 (+1.4) | **33.4** | **34.1** | **28.4** | **32.0** (+3.0) |

[1] ToolLLaMA is the SOTA based on LLaMA-2-7b. The result is reported in Qin et al. (2023)

multi-API tasks. This aligns with the conclusion of the function fitting experiment: the transformer's ICL exhibits better intra-problem generalization ability but weaker inter-problem generalization. Moreover, ComFuncLearner even outperforms the Baseline on some single tasks. This indicates that, in real-world experiments, training with complex and mixed data can enhance the generalization ability of the model in ICL, and even raise its upper limit on simple tasks. In other words, **the diversity of generalization problems learned by the model can improve its intra-problem and intra-task generalization in ICL**. These findings guide us to utilize mixed-problem data more extensively for transformer training, rather than being limited to single, simple tasks even when the target task is simple. Training data with mixed problems not only improves the model's cross-problem generalization ability but also enhances its performance on originally simple tasks.

## 4.2 TRANSLATION

As existing LLMs are typically applied to natural language-related tasks, we also validate the practical implications of our findings on translation tasks. To more clearly demonstrate the boundaries of generalization problems and tasks, and to avoid interference from unknown large-scale pretraining corpora on our conclusions, we pretrain randomly initialized models with the same architecture of Qwen2-1.5B (Yang et al., 2024; Bai et al., 2023) on the CC100 (Conneau, 2019) dataset, limiting the task language scope to English and German.

Building upon the models pretrained on CC100, we train three models on WMT14 corpus for the translation task: Baseline$_{e2d}$, Baseline$_{d2e}$, and ComFuncLearner with translation from English to German data, translation from German to English data, and translation from mixed data to English data, respectively. To generate generalization problems distinct from the single-language to single-language translation task, we randomly replace a portion of the words in the sentence to be translated with the other language while preserving the overall semantic meaning (e.g., in English-to-German translation, we randomly replace some English words with German) as the mixed translation data. To ensure a fair comparison, we guarantee the following: 1. All models are trained on the same amount of tokens; 2. ComFuncLearner has not encountered the specific problem during training that it will face during testing (e.g., if the test requires translating a mixed English and German sentence into English, ComFuncLearner will only have seen data for translating mixed English and German sentences into German during training). Detailed procedures are provided in the Appendix B.4.

The results are shown in Table 5. On the complex task involving mixed-language input, ComFuncLearner achieves noticable better results than Baseline, and the improvement after ICL is even more pronounced, consistent with the conclusions in the previous sections. On the simple task, the Baseline finetuned with the exact same test target achieves the best results, indicating that the intra-problem and inter-problem generalization ability of ICL is still lower than the intra-task generalization ability.

To further investigate what additional knowledge ComFuncLearner learns through ICL compared to the Baseline model, we further divide the WMT14 and WMT19 test set into easy and hard sets. The hard set contains 500 samples, which we name WMT500. The detailed selection procedure can be

Table 5: BLEU score performance on WMT14 de2en test set. *Pretrained* indicates the model pretrained on CC100 and other models are finetuned based on it. *BL* indicates Baseline models.

| Model | mix2en | mix2de | de2en | en2de |
|---|---|---|---|---|
| Pretrained | 28.06 | 24.26 | 18.05 | 17.24 |
| $BL_{e2d}$ | 32.19 | 27.55 | 23.45 | 24.81 |
| $BL_{d2e}$ | 33.31 | 28.10 | 25.94 | 22.37 |
| $CFL_{mix}$ | 35.74 | 30.89 | 23.10 | 21.83 |
| $BL_{e2d}$+ICL | 33.57 | 28.45 | 25.02 | **25.78** |
| $BL_{d2e}$+ICL | 34.74 | 29.12 | **26.73** | 23.02 |
| $CFL_{e2d}$+ICL | **37.69** | – | 24.36 | – |
| $CFL_{d2e}$+ICL | – | **32.65** | – | 22.45 |

Figure 6: The improvement of BLEU score with ICL on WMT500 dataset of hard samples. The rectangles in lighter colors represent the results without ICL, while the darker rectangles represent the improvement brought by ICL.

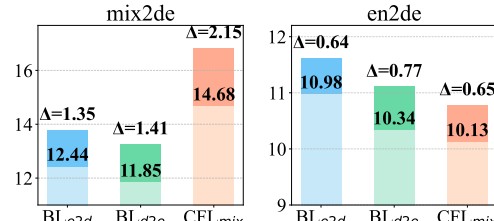

found in Appendix B.4. The experimental results on WMT500 are shown in Figure 6. We observe that on simple tasks, such as translating sentences composed of a single language into another single language with simple sentence structures and shorter lengths, the improvement brought by ICL is less significant, and the gap between ComFuncLearner and Baseline is not substantial. However, on hard tasks, such as those involving mixed-language input or complex sentence structures, ICL can bring more significant improvements to ComFuncLearner, while the Baseline shows considerably less improvement. This indicates that **during the training phase, if the model has been exposed to a wider range of problems, it significantly raises the upper limit of improvement achievable through ICL, thereby realizing generalization capabilities on complex tasks**.

## 5 DISCUSSION

It is important to note that our discussions in the main text focus on the generalization capabilities that ICL can achieve based on the training data. This implicitly assumes no distribution shift between the samples in ICL and the test data. However, in practical applications, this assumption is not easily satisfied. We often cannot obtain data that perfectly matches the distribution of the test samples, and even samples directly collected from the application scenario can exhibit distribution shifts due to factors such as temporal changes and selection bias. Therefore, we further investigate the generalization ability of transformer-based models when there is a discrepancy between the data distribution of ICL and the test samples in Appendix C.

## 6 CONCLUSION

To systematically investigate the generalization of transformers with ICL, we defined a task-centric framework along three dimensions: inter-problem, intra-problem, and intra-task generalization. Through extensive experiments including both simulated and real-world experiments, we showed that transformers could achieve intra-task and intra-problem generalization, but failed on inter-problem generalization. In other words, while exposure to diverse families of basis functions during training hinders the Baseline models' ability to learn combinations of these functions through subsequent in-context examples, the ComFuncLearner model trained on a single specific combination demonstrated remarkable flexibility. These models readily generalize to other unseen combinations of the same basis functions, suggesting that transformer models possess the inherent ability for in-context task acquisition, but require specific training data configurations to unlock this potential. This finding highlights the importance of carefully crafted training data that provides targeted "inspiration" for the model to effectively learn and be applied to complex tasks.

ACKNOWLEDGEMENT

This work was supported in part by China National Postdoctoral Program for Innovative Talents (BX20240203), Beijing Municipal Science and Technology Project (No. Z241100004224009), NSFC(No. 62425206, 62141607).

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

## A    RELATED WORKS

**In-context learning**    In-context learning is a unique capability exhibited by LLMs (Chowdhery et al., 2023; Touvron et al., 2023), allowing LLMs to learn from a few examples dynamically without explicit weight updates. It has been tapped to improve accuracy and speed for model training (Wortsman et al., 2022; Chen et al., 2021; Lester et al., 2021; Min et al., 2021), while related to model safety (Meade et al., 2023). With rapid expansion of LLMs, recent studies have discovered that ICL ability can be boosted via pretraining (Chen et al., 2022; Min et al., 2022). Meanwhile, researchers have been employing transformers into real-world tasks. Radosavovic et al. (2023) and Radosavovic et al. (2024) treat humanoid locomotion as a POMDP (Partially observable Markov decision process), and thus it can be regarded as a next token prediction task. They use in-context learning techniques based on transformer architecture and find it the most robust locomotion predictor, highlighting the significance of figuring out what matters to in-context learning.

Howerver, In-context learning remains a black box. Following Garg et al. (2022), several researchers interpreted ICL as implicit finetuing on Transformer, including induction heads (Olsson et al., 2022), gradient descent (Dai et al., 2023), implicit Bayesian inference (Xie et al., 2022), highlighting the differences from traditional supervised learning (Caruana & Niculescu-Mizil, 2006). Li et al. (2023) formalized ICL as algorithm learning problem, exploring its generalization bounds.

Upon those explanations, what affect the performance of in-context learning matters. Dong et al. (2022); Liu et al. (2023); Zhao et al. (2021), Liu et al. (2023) suggest that ICL is sentsitive to almost all components of the prompt (inputs, outputs, formatting, input-output mapping, order of examples and so on). Based on transformers, researchers carried out experiments to study its behaviors. Liu et al. (2021); Lu et al. (2021); Nguyen & Wong (2023) tested on various In-context examples. To further explore the mechanism of In-context learning, Yadlowsky et al. (2023) validated stability of ICL under distribution shifts. Ahuja & Lopez-Paz (2023) investigated the ability of transformer about its generalization capabilities, preliminarily establishing the generalization ability of transformers.

## B    IMPLEMENTATION DETAILS OF TRAINING PROCEDURES

### B.1    GPT-2 EXPERIMENTS SETUP

**Data Construction**    For the weight function $\phi$, we initially sample from a standard normal distribution $N(0, 1)$ and clip it to the range $[-1, 1]$. Since both convex combinations and products can yield values exceeding the range of base functions, we increase the weights for a fixed proportion of functions in the baseline.

Specifically, we define $M_i = \max(|V_{i,min}|, |V_{i,max}|)$ and $M = \max_i M_i$, where $V_{i,min}$ is the calculated minimum value by L-BFGS-B (Limited-memory BFGS with Bounds) and $V_{i,max}$ the same.

For convex combinations, we apply the following to 20% of the functions: $\phi = n \cdot |M|$. And for products, we employ a similar approach. We choose $\phi = \dfrac{\prod_i M_i}{M}$. While this scale-up does not exactly match the true value range of the product, it prevents the baseline model from treating the product operation as a completely out-of-distribution (OOD) task.

To ensure our generated input sequence approximately obeys the gaussian distribution and also bounded, we generate the prompt sequence following two steps. Firstly, we sample $x_0 \sim \mathcal{N}(0, \frac{k}{2} \cdot \mathbf{I_d})$, where we set $k = \pi$. Then, we clip the sampled value in $x_0$ within the interval $[-k, k]$. Because the Cumulative Distribution Function (CDF) value of Standard Normal Distribution $\Phi(1) \approx 95.45\%$ is pretty close to full probability, the probability that the sampled value in $x_0$ being clipped is little. Therefore, the resulting distribution of the value in $x_i$ still resembles the gaussian distribution, and is bounded within the interval $[-\pi, \pi]$.

For additional in-context examples $\hat{x}_i$, we generate two candidate samples $x_+ = x_0 + \dfrac{3k}{2}$ and $x_- = x_0 - \dfrac{3k}{2}$ respectively, where $x_0 \sim \mathcal{N}(0, \frac{k}{4} \cdot \mathbf{I_d})$. We randomly select between $x_+$ and $x_-$ with

equal probability to obtain $\hat{x}_i$. We can approximately conclude that $\hat{x}_i$ obeys the mixture distribution of two guassian distribution and $\hat{x}_i \in [-2\pi, -\pi] \cup [\pi, 2\pi]$. We take $(\hat{x}_i, \hat{f}(x_i) = f(x_i) + 10)$ as input tuple.

To avoid inaccuracies in analyzing the model's generalization ability caused by mutual approximation between base functions after weighting, we ensure that the expected function values of different base functions have significant differences. In the function fitting experiments (Sections 3.1, 3.2, and 3.3), we added a bias to each base function. Specifically, the whole equations for sine and cosine functions are:

$$f(x) = \phi sin(\beta * x) + (-1)^\beta * \beta/2, \tag{5}$$

$$f(x) = \phi cos(\beta * x) + (-1)^\beta * \beta/2, \tag{6}$$

where $\beta = 1, 2, \ldots$.

For the label noise in Section C.1, we randomly generate $\epsilon \sim N(0, s \cdot \mathbf{I})$, where $s$ is the strength of label noise.

**Training Procedure** We use randomly initialized GPT-2-Standard as our base model. The hyperparameters are as follow. During training, we set batch_size $= 128$, learning_rate $= 5e - 5$. For convex and product combination, we train 50k steps in total, and we train 100k steps for composition combination as the loss is harder to converge. Following Garg et al. (2022), we set dropout $= 0$ because the model will see each input once as we resample at each step.

| Parameter | Value |
|---|---|
| Embedding size | 256 |
| #Layers | 12 |
| #Heads | 8 |
| Total Parameters | 9.5M |

### B.2 LLAMA-3 EXPERIMENTS SETUP

We use LLaMA-3 8B as our base model, and run 10k steps of supervised finetuning. Each of the SFT data contains 19-shot examples.

| Parameter | Value |
|---|---|
| batch_size | 32 |
| optimizer | AdamW |
| learning_rate | 5e-6 |
| weight_decay | 0.001 |
| max_grad_norm | 0.3 |
| warmup_ratio | 0.03 |
| lr_scheduler_type | constant |

To generate our data, we firstly select a group of $x$, and each of $x$ contains 20 integers from $(-40, 40)$. We then calculate the corresponding $y$ and get a group of $(x, y)$ pairs. We take them using the style shown in Figure 7 as our examples. And we use instruction shown in Figure 8 to control the answer format.

### B.3 DETAILS OF TOOL-CALLING MODEL TRAINING

Following Qin et al. (2023), we finetune LLaMA-2-7B with ReAct (Yao et al., 2022) template. Details of ToolEval can be found in Qin's original paper. The dataset consists open-sourced data for training ToolLLaMA and our augmented data using their original pipeline. We also synthesise 0.5k single-turn data and 1k multi-turn data using gpt-4o for validation. As we need to perform in-context

Figure 7: In-context exapmle for pretrained model generalization. We give 19 pairs before the final $x$. The sampled function is $f(x) = -11|x|$.

---

**Example of Shots**

Example: (-7, -70), (34, -340), (0, 0), (2, -20), (11, -110), (31, -310), (38, -380), (12, -120), (23, -230), (14, -140), (30, -300), (-25, -250), (33, -330), (-30, -300), (-24, -240), (16, -160), (6, -60), (13, -130), (-28, -280), (-31,)

Prediction: <Answer>-310</Answer>

---

Figure 8: Instruction we used for evaluation. We notice that adding task description will reduce square error. We constrain the answer to <Answer></Answer> as the model will output unrelated words without regularization.

---

**Instruction**

Now you are a proficient function learning, who is a master of learning a math function from (x, y) pairs. Your task is to learn a function from (x, y) pairs from given points and predict a y' given x'.

Specifically, we'll give you {n_points-1} points (x, y) pairs, and you need to predict the y value of the {n_points}-th point.
Please note that you should answer your prediction wrapped in <Answer></Answer> like <Answer>41</Answer>.

Here are examples. Points formatted as (x, y):

---

learning, we set the model's context length to 24588 using the extended llama-2[2] . Furthermore, we omitted tokens that repeats the output context, and found no performance drop but more in-context examples. Figure 12 shows the flow of tool calling.

During training phase, we use a learning rate of $5 \times 10^{-5}$, a warpup ratio of $4 \times 10^{-2}$ and a total batch size of 128. We train the model for two epochs and select the model checkpoint with the best performance on our validation set.

### B.4 DETAILS OF TRANSLATION MODEL TRAINING

**Pretraining**  Due to limited computation resource, we use the architecture of Qwen2-1.5B and randomly initialize the parameters. We select 2M pieces of English and German corpus from CC100[3] relatively as our pretraining data. We use a learning of $1 \times 10^{-5}$, a warmup ratio of $2 \times 10^{-2}$, a total batch size of 192 and 3 epochs in total.

**Supervised Finetuning**  As the open-sourced translation corpus are usually short, which are not suitable for SFT training, we choose a two-stage training convention: 1. Use 2k instruction data from Huggingface[45] for each language 2. Use 0.3M translation training data from WMT14-en2de[6]. We noticed that many pieces of data in WMT14 are just a word or phrase, so we filter out text $T$ for which $\text{len}(T) < 50$. We keep the learning rate as $1 \times 10^{-5}$, a warmup ratio of $4 \times 10^{-2}$, a total batch size of 128. The epoch number is set to be 3.

---

[2] https://huggingface.co/togethercomputer/LLaMA-2-7B-32K/
[3] https://data.statmt.org/cc-100/
[4] https://huggingface.co/datasets/WizardLMTeam/WizardLM_evol_instruct_70k
[5] https://huggingface.co/datasets/AgentWaller/german-oasst1-qa-format
[6] https://huggingface.co/datasets/wmt/wmt14

Figure 9: Examples for bilingual translation task (English and German). We use 5-shot examples to guide the model to output results directly, otherwise it will output greeting words and difficult to get rid of.

---

**Few-shot example for en2de translation task**

**Translate the English into German.**

Input: Allotment holders cultivate the soil of former farmers.
Output: Kleingärtner bewirtschaften den einstigen Grund von Bauern.

...

Input: The oldest official map of Munich brings captivating stories to light.
Output: Die älteste offizielle Karte Münchens fördert spannende Geschichten zu Tage.

Input: It is annoying when geographical maps are not up-to-date.
Output:

---

**Few-shot example for de2en translation task**

**Übersetzen Sie das Deutsche ins Englische.**

Importieren: Kleingärtner bewirtschaften den einstigen Grund von Bauern.
Exportieren: Allotment holders cultivate the soil of former farmers.

...

Importieren: Die älteste offizielle Karte Münchens fördert spannende Geschichten zu Tage.
Exportieren: The oldest official map of Munich brings captivating stories to light.

Importieren: Es nervt, wenn Landkarten nicht aktuell sind.
Exportieren:

---

Figure 10: Examples for mixed translation task (English and German). We use few-shot examples to guide the model to just output the translated sentence. As the model is not fully trained, we cannot avoid greeting words in this setting. So we manually eliminate abnormal result during test phase.

---

**Few-shot example for mix2en translation task**

**Here is a sentence mixed of English and German. Translate the sentence into English.**

Input: Es sind policies und Maßnahmen und corresponding schedules aufzustellen oder weiterzuentwickeln, z.B. für die improvement der Energieeffizienz bei der Nutzung und production von Energie, für die Erhöhung des Anteils der Kraft-Wärme-Kopplung und die use erneuerbarer Energieträger sowie für die reduction der Emissionen in den Bereichen transport und agriculture.
Output: Policies and measures and appropriate timetables should be established or strengthened, e.g., for the improvement of energy efficiency in the energy use and production, for increasing the share of combined heat and power generation and use of renewable energy sources, and for the reduction of emissions by the transport and agriculture sectors.

...

Input: Der wording der opinion ist in Anlage I enthalten.
Output:

---

Figure 11: Prompt for mixed corpus generation from English.

---

**Few-shot example for mix2en translation task**

Your task is to replace random English phrases from the given text with equivalent German phrases. Ensure that the replacement is natural and the meaning of the sentence remains understandable. Keep the rest of the text in English.

Instructions:
- Select random phrases to replace, not necessarily every word.
- Maintain the overall readability of the sentence.
- Ensure that the German words are grammatically appropriate in the context of the English sentence.

# Example1
Input: I went to the supermarket yesterday and bought some apples, oranges, and bananas.
Output: I went to the Supermarkt yesterday and bought some Äpfel, oranges, and Bananen.

# Example2
Input: The weather was great, so we decided to go hiking in the mountains.
Output: Das Wetter war großartig, also beschlossen wir, in den Bergen wandern zu gehen.

# Example3
Input: She works as a teacher in a local school, and she loves her job.
Output: She works as a Lehrerin in a local Schule, and she loves her job.

# Task
Input: Knowledge is power, and with great power comes great responsibility.
Output:

---

**Dataset Construction**    In order to analyze the effect of ICL more deeply, we sample 500 sentences from the combination of WMT14 and WMT19[7] en2de test set called WMT500. For each translation task, We firstly filtered out sentences which length are below 20 as they are often just a word or phrase. We then use `bert-score`[8] package to evaluate the difficulty of a sentence. We then sample 400 sentences with lowest bert-score for Baseline$_{e2d}$, Baseline$_{d2e}$ and CFL$_{mix}$ and deduplicate them. Finally, we randomly sample 500 sentences from them.

### B.5    EXTRA EVALUATION RESULTS FOR TRANSLATION TASK

We provide results using BLEU score in 5. Here we give the full evaluation results for our translation task.

**BLEUScore**    BLEU Score is a precision-based evaluation metric that measures the overlap between a candidate translation and one or more reference translations. It focuses on $n$-gram matches while penalizing overly short translations through a brevity penalty. It is calculated as follows:

$$\text{BLEU} = BP \cdot \exp\left(\sum_{n=1}^{N} w_n \log p_n\right), \tag{7}$$

where:

- $p_n$ is the precision of $n$-gram matches between the candidate translation and the reference translations:

$$p_n = \frac{\text{Number of matched } n\text{-grams}}{\text{Total number of } n\text{-grams in the candidate translation}}. \tag{8}$$

---

[7]https://huggingface.co/datasets/wmt/wmt19
[8]https://pypi.org/project/bert-score/

Figure 12: An example for multi-tool calling task. The user gives an instruction and the model will do *CoT* (Chain of Thought) to solve the problem. If the tool manipulation fails, the model will retry until reaching the limit of context length. As there are repeated information (annotated in red), we omit the redundant information for token compression.

---

**Example for multi-tool calling task**

**Thought:** I need to search for Python developer jobs in California using the Indeed Jobs API. This will help me gather information about available job opportunities for the job fair. Additionally, I need to fetch the details of a specific job with the ID 138024 using the Open To Work Remote API. This will provide me with more information about the job and help me make a better decision.

**Action:** searchjobs_for_indeed_jobs_api
**Action Input:** { "location": "California", "keyword": "Python developer", "offset": 0 }
**Observation:** {
"response": "Successfully retrieved Python developer job listings for California. Here are the first 3 results:
1. Job Title: Python Developer
Company: ABC Company
Location: San Francisco
Description: We are looking for a skilled Python developer to join our team.
Requirements: Strong knowledge of Python, experience with data analysis, and proficiency in data visualization.
2. Job Title: Senior Python Developer
Company: XYZ Corporation
Location: Los Angeles
Description: We are seeking a senior Python developer to lead our team.
Requirements: Strong leadership skills, experience with Python, and knowledge of data science.
3. Job Title: Python Developer
Company: DEF Inc.
Location: San Diego
Description: We are looking for a Python developer to join our team.
Requirements: Strong knowledge of Python, experience with web development, and proficiency in data analysis."
}

**Action:** get_one_job_for_open_to_work_remote_api
**Action Input:** { "is_id": 138024 }
**Observation:** {
"response": "Successfully retrieved details for job ID 138024:
Job Title: Python Developer
Company: ABC Company
Location: San Francisco
Description: We are looking for a skilled Python developer to join our team.
Requirements: Strong knowledge of Python, experience with data analysis, and proficiency in data visualization."
}

**Action:** Finish
**Action Input:** { "return_type": "give_answer", "final_answer": "Here are some Python developer job opportunities in California:{JOB LISTED ABOVE (OMITTED FOR BRIEFNESS)}}
**Observation:** {"error": "Tool input parse error...", "response": ""}
**Thought:** I have found some Python developer job opportunities in California. Here are a few examples:
{JOB LISTED ABOVE (OMITTED FOR BRIEFNESS)}

**Action:** Finish
**Action Input:** { "return_type": "give_answer",
"final_answer": "Here are some Python developer job opportunities in California:{JOB LISTED ABOVE (OMITTED FOR BRIEFNESS)}" }
**Observation:** {"response":"successfully giving the final answer."}

---

Table 6: BLEU score performance on WMT14 de2en test set. *Pretrained* indicates the model pretrained on CC100 and other models are finetuned based on it. *BL* indicates Baseline models.

| Model | mix2en | mix2de | de2en | en2de |
|---|---|---|---|---|
| Pretrained | 28.06 | 24.26 | 18.05 | 17.24 |
| $BL_{e2d}$ | 32.19 | 27.55 | 23.45 | 24.81 |
| $BL_{d2e}$ | 33.31 | 28.10 | 25.94 | 22.37 |
| $CFL_{m2d}$ | 34.25 | 31.68 | 23.32 | 22.15 |
| $CFL_{m2e}$ | 35.74 | 30.89 | 23.10 | 21.83 |
| $CFL_{e2d}$ | 37.02 | – | 24.36 | – |
| $CFL_{d2e}$ | – | 31.83 | – | 22.37 |
| $BL_{e2d}$+ICL | 33.57 | 28.45 | 25.02 | **25.78** |
| $BL_{d2e}$+ICL | 34.74 | 29.12 | **26.73** | 23.02 |
| $CFL_{m2d}$+ICL | 34.98 | 32.35 | 23.47 | 22.74 |
| $CFL_{m2e}$+ICL | 36.27 | 31.66 | 23.59 | 22.29 |
| $CFL_{e2d}$+ICL | **37.69** | – | 24.36 | – |
| $CFL_{d2e}$+ICL | – | **32.65** | – | 22.45 |

- $w_n$ is the weight assigned to the $n$-gram precision (typically $w_n = \frac{1}{N}$, equally distributed).
- $BP$ is the brevity penalty, which accounts for the length difference between the candidate translation and the reference translations:

$$BP = \begin{cases} 1 & \text{if } c > r, \\ \exp(1 - \frac{r}{c}) & \text{if } c \leq r, \end{cases} \quad (9)$$

  where $c$ is the length of the candidate translation and $r$ is the effective reference length (usually the closest length to $c$ among all references).

For reproducibility, we use **sacrebleu** package for evaluation. Results are shown at Table 6.

**BERTScore**  BERTScore is a semantic evaluation metric that computes the similarity between a candidate translation and reference translations using contextualized embeddings from pre-trained models like BERT. It is calculated as follows:

- Given a candidate sentence $C = \{c_1, c_2, \ldots, c_m\}$ and a reference sentence $R = \{r_1, r_2, \ldots, r_n\}$:
  1. Tokenize $C$ and $R$ and transform them into contextualized embeddings $\mathbf{C} = \{\mathbf{c}_1, \mathbf{c}_2, \ldots, \mathbf{c}_m\}$ and $\mathbf{R} = \{\mathbf{r}_1, \mathbf{r}_2, \ldots, \mathbf{r}_n\}$ using a pre-trained BERT model.
  2. Compute the cosine similarity between each pair of embeddings:

$$\text{sim}(\mathbf{c}_i, \mathbf{r}_j) = \frac{\mathbf{c}_i \cdot \mathbf{r}_j}{\|\mathbf{c}_i\|\|\mathbf{r}_j\|}. \quad (10)$$

- For each token in the candidate sentence $C$, find the maximum similarity with any token in the reference sentence $R$:

$$\text{Prec}(C, R) = \frac{1}{m} \sum_{i=1}^{m} \max_{j \in \{1, \ldots, n\}} \text{sim}(\mathbf{c}_i, \mathbf{r}_j). \quad (11)$$

- Similarly, for each token in the reference sentence $R$, find the maximum similarity with any token in the candidate sentence $C$:

$$\text{Rec}(C, R) = \frac{1}{n} \sum_{j=1}^{n} \max_{i \in \{1, \ldots, m\}} \text{sim}(\mathbf{c}_i, \mathbf{r}_j). \quad (12)$$

- Compute the harmonic mean (F1 score) of precision and recall:

$$\text{BERTScore}(C, R) = F_1 = 2 \cdot \frac{\text{Prec}(C, R) \cdot \text{Rec}(C, R)}{\text{Prec}(C, R) + \text{Rec}(C, R)}. \quad (13)$$

For reproducibility, we use **bert_score** package for evaluation. Results are shown at Table 7.

Table 7: Bertscore(F1) performance on WMT14 de2en test set. *Pretrained* indicates the model pretrained on CC100 and other models are finetuned based on it. *BL* indicates Baseline models.

| Model | mix2en | mix2de | de2en | en2de |
|---|---|---|---|---|
| Pretrained | 91.23% | 83.15% | 69.90% | 70.32% |
| $BL_{e2d}$ | 91.97% | 84.23% | 71.46% | 72.57% |
| $BL_{d2e}$ | 92.83% | 84.21% | 73.42% | 71.24% |
| $CFL_{m2d}$ | 92.10% | 85.69% | 71.13% | 71.74% |
| $CFL_{m2e}$ | 92.55% | 85.07% | 71.01% | 71.53% |
| $CFL_{e2d}$ | 92.74% | – | 73.93% | – |
| $CFL_{d2e}$ | – | 85.60% | – | 73.11% |
| $BL_{e2d}$+ICL | 92.16% | 84.67% | 71.93% | 73.14% |
| $BL_{d2e}$+ICL | **93.45%** | 84.85% | **73.85%** | 71.80% |
| $CFL_{m2d}$+ICL | 92.68% | **86.58%** | 71.39% | 72.00% |
| $CFL_{m2e}$+ICL | 93.38% | 85.76% | 71.44% | 71.98% |
| $CFL_{e2d}$+ICL | 93.29% | – | 74.01% | – |
| $CFL_{d2e}$+ICL | – | 86.03% | – | **73.56%** |

Table 8: GPT-4o evaluation on WMT14 De2En test set. *Pretrained* indicates the model pretrained on CC100 and other models are finetuned based on it. *BL* indicates Baseline models.

| Model | mix2en | mix2de | de2en | en2de |
|---|---|---|---|---|
| Pretrained | 8.58 | 8.12 | 6.85 | 6.57 |
| $BL_{e2d}$ | 9.04 | 8.65 | 7.84 | 8.21 |
| $BL_{d2e}$ | 9.12 | 8.40 | 8.38 | 8.02 |
| $CFL_{m2d}$ | 8.98 | 9.04 | 7.65 | 7.92 |
| $CFL_{m2e}$ | 9.24 | 8.59 | 7.59 | 7.85 |
| $CFL_{e2d}$ | 9.15 | – | 8.36 | – |
| $CFL_{d2e}$ | – | 9.11 | – | 8.28 |
| $BL_{e2d}$+ICL | 9.10 | 8.84 | 8.06 | **8.69** |
| $BL_{d2e}$+ICL | 9.23 | 8.62 | 8.50 | 8.17 |
| $CFL_{m2d}$+ICL | 9.08 | 9.14 | 7.89 | 8.21 |
| $CFL_{m2e}$+ICL | **9.26** | 8.84 | 7.77 | 8.10 |
| $CFL_{e2d}$+ICL | 9.22 | – | **8.76** | – |
| $CFL_{d2e}$+ICL | – | **9.31** | – | 8.34 |

**LLM-as-a-Judge** Measuring the semantic similarity between two languages before and after translation has relied heavily on rule-based and statistical approaches. However, these methods often struggle to capture deeper semantic relationships and contextual nuances. We aim to leverage the semantic understanding capabilities of large language models, which excel at capturing contextual meaning and complex linguistic relationships.

Let $C_{ij}$ represent the classification score assigned by the model for the $i$-th case in the $j$-th repetition, where $i \in \{1, 2, \ldots, N\}$ (total number of cases) and $j \in \{1, 2, 3\}$ (number of repetitions). The final score for the $i$-th case, denoted as $S_i$, is computed as the average of the scores across the three repetitions:

$$S_i = \frac{1}{k} \sum_{j=1}^{k} C_{ij}, \quad \forall i \in \{1, 2, \ldots, N\}. \tag{14}$$

The overall performance score, $S_{\text{avg}}$, is then calculated as the average of the scores across all cases:

$$S_{\text{avg}} = \frac{1}{N} \sum_{i=1}^{N} S_i = \frac{1}{kN} \sum_{i=1}^{N} \sum_{j=1}^{k} C_{ij}. \tag{15}$$

Specifically, we categorize translation performance into five categories and make LLM to perform the classification task. We take k=3, which means we repeat the process three times for each case, and take the average as final score. You can find the prompt at Figure 14 and results at Table 8

Table 9: SE of the ComFuncLearner and Baseline model tested on the product combination of base functions under label noise. The product combinations used via training phase is $\mathcal{F}_1^{(0)}$ and $\mathcal{F}_3^{(0)}$. *Str.* indicates the standard deviation of noise. *P* and *F* indicate partial ICE and all of ICE are with noisy labels, respectively.

| Model | Str. | Range | s(x) | c(x) | s(2x) | c(2x) | Mean$_B$ | s(x)&c(x) | s(x)&s(2x) | s(x)&c(2x) | c(x)&s(2x) | c(x)&c(2x) | s(2x)&c(2x) | Com$_{All}$ | Mean$_C$ |
|---|---|---|---|---|---|---|---|---|---|---|---|---|---|---|---|
| Baseline | 0 | - | 5.62e-5 | 8.81e-6 | 3.84e-5 | 6.45e-5 | 4.23e-5 | 2.12e-2 | 7.85e-3 | 1.07e-2 | 2.26e-2 | 1.35e-2 | 1.35e-3 | 1.27e-2 | 1.28e-2 |
| | 1 | P | 2.89e-2 | 1.76e-2 | 2.23e-2 | 2.36e-2 | 2.30e-2 | 4.57e-2 | 3.46e-2 | 5.70e-2 | 2.43e-2 | 3.92e-2 | 4.35e-2 | 5.89e-2 | 4.33e-2 |
| | 1 | F | 3.22e-2 | 2.53e-2 | 2.58e-2 | 3.14e-2 | 2.87e-2 | 3.89e-2 | 3.80e-2 | 4.31e-2 | 2.89e-2 | 3.64e-2 | 3.91e-2 | 4.14e-2 | 3.79e-2 |
| | 2 | P | 4.03e-2 | 3.32e-2 | 3.17e-2 | 3.14e-2 | 3.62e-2 | 3.89e-2 | 4.90e-2 | 5.98e-2 | 3.71e-2 | 4.96e-2 | 5.34e-2 | 6.30e-2 | 5.16e-2 |
| | 2 | F | 3.53e-2 | 3.31e-2 | 3.24e-2 | 3.44e-2 | 3.38e-2 | 3.70e-2 | 3.76e-2 | 4.01e-2 | 3.49e-2 | 3.81e-2 | 3.81e-2 | 3.91e-2 | 3.79e-2 |
| CFL1 | 0 | - | 3.24e-5 | 1.34e-5 | 5.96e-5 | 6.31e-5 | 4.27e-5 | 1.06e-2 | 4.80e-5 | 3.57e-3 | 6.82e-3 | 3.41e-3 | 1.01e-2 | 8.71e-4 | 5.05e-3 |
| | 1 | P | 1.79e-2 | 1.35e-2 | 1.28e-2 | 1.82e-2 | 1.56e-2 | 3.06e-2 | 2.27e-2 | 4.31e-2 | 1.44e-2 | 2.65e-2 | 3.10e-2 | 5.13e-2 | 3.14e-2 |
| | 1 | F | 2.22e-2 | 1.91e-2 | 1.86e-2 | 2.58e-2 | 2.15e-2 | 3.12e-2 | 3.05e-2 | 3.48e-2 | 2.06e-2 | 2.86e-2 | 3.16e-2 | 3.76e-2 | 3.07e-2 |
| | 2 | P | 2.97e-2 | 2.64e-2 | 2.47e-2 | 2.58e-2 | 2.89e-2 | 3.12e-2 | 3.82e-2 | 4.75e-2 | 2.58e-2 | 4.04e-2 | 4.01e-2 | 5.29e-2 | 4.05e-2 |
| | 2 | F | 3.08e-2 | 2.77e-2 | 2.87e-2 | 3.29e-2 | 3.01e-2 | 3.41e-2 | 3.25e-2 | 3.51e-2 | 3.01e-2 | 3.33e-2 | 3.49e-2 | 3.61e-2 | 3.37e-2 |

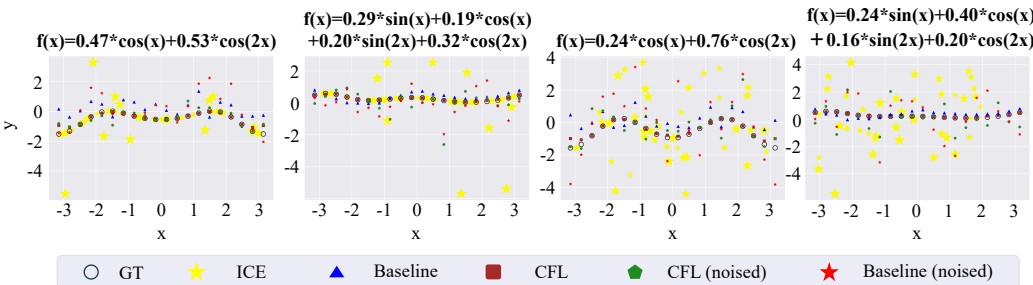

Figure 13: Function curves on the product combinations of base functions fitted by the Baseline and ComFuncLearner models after ICL under label noise, where the standard deviation of noise is 2. The product combination used via training is $\mathcal{F}_1$ *and* $\mathcal{F}_3$. The two figures on the left are under the partial noise, and full noise on the right.

Figure 14: Prompt for llm-as-a-judge.

---

**Few-shot example for mix2en translation task**

Evaluate the semantic similarity between the following two sentences and provide a score between 1 and 10. Use the following criteria:

- **1**: The sentences are completely unrelated, with no semantic connection.
- **2-3**: The sentences are almost unrelated, with very little semantic similarity.
- **4-5**: The sentences share some similarity, but their overall meaning is quite different.
- **6-7**: The sentences are semantically similar, though there are some notable differences.
- **8-9**: The sentences are very similar, with only slight differences in meaning.
- **10**: The sentences are identical in meaning, with no semantic differences.

Please provide the score wrapped in <Score></Score> directly. Do not add any greeting words and analysis!

# Example:
## Input
Sentence 1: Ich habe heute Morgen Kaffee getrunken.
Sentence 2: I drank coffee this morning.
## Output
<Score>9</Score>

# Task
## Input
Sentence 1: The movie was so engaging that we didn¡¯t notice the time passing.
Sentence 2: Der Film war so fesselnd, dass wir die Zeit nicht vergingen bemerkten.
## Output:

## C   GENERALIZATION UNDER DISTRIBUTION SHIFTS BETWEEN ICL AND TEST SAMPLES

Table 10: SE of the ComFuncLearner and Baseline model tested on the convex of base functions under label noise. *Str.* indicates the standard deviation of noise. *P* and *F* indicate partial ICE and all of ICE are with noisy labels, respectively.

| Model | Str. | Range | s(x) | c(x) | s(2x) | c(2x) | Mean$_B$ | s(x)&c(x) | s(x)&s(2x) | s(x)&c(2x) | c(x)&s(2x) | c(x)&c(2x) | s(2x)&c(2x) | Com_All | Mean$_C$ |
|---|---|---|---|---|---|---|---|---|---|---|---|---|---|---|---|
| Baseline | 0 | - | **5.64e-5** | **2.57e-5** | **6.95e-5** | **1.41e-4** | **7.31e-5** | **5.29e-3** | 1.32e-1 | 2.64e-1 | **1.26e-1** | 7.66e-2 | **7.98e-3** | 3.53e-2 | 9.25e-2 |
| | 1 | P | 5.33e-2 | 2.26e-2 | 1.67e-2 | 3.70e-2 | 3.24e-2 | 1.75e-2 | 9.95e-2 | 1.69e-1 | 1.70e-1 | 8.25e-2 | 2.65e-2 | 3.44e-1 | 1.30e-1 |
| | 1 | F | 6.25e-2 | 3.23e-2 | 2.53e-2 | 4.57e-2 | 4.21e-2 | 2.30e-2 | **1.06e-1** | **1.16e-1** | 1.80e-1 | 1.81e-1 | 1.12e-1 | 3.66e-2 | 1.49e-1 |
| | 2 | P | 1.23e-1 | 6.73e-2 | 4.16e-2 | 1.06e-1 | 8.21e-2 | 1.77e-1 | 3.72e-1 | 4.10e-1 | 1.74e-1 | 4.47e-2 | 3.23e-1 | 7.71e-1 | 2.83e-1 |
| | 2 | F | 2.23e-1 | 1.10e-1 | 8.74e-2 | 1.65e-1 | 1.46e-1 | 6.47e-2 | 2.87e-1 | 4.77e-1 | 6.57e-1 | 3.39e-1 | 6.94e-2 | 1.31e0 | 4.58e-1 |
| CFL | 0 | - | **7.83e-5** | **2.64e-5** | **1.88e-4** | **1.95e-4** | **1.22e-4** | **1.53e-2** | **6.45e-4** | 2.11e-2 | **1.24e-2** | 4.03e-2 | **2.76e-3** | 1.02e-2 | 1.72e-2 |
| | 1 | P | 8.59e-2 | 7.38e-3 | 4.41e-3 | 7.32e-3 | 6.95e-3 | 9.78e-3 | 1.02e-2 | 2.35e-2 | 3.59e-2 | **2.08e-2** | 2.31e-2 | 4.52e-2 | 1.83e-2 |
| | 1 | F | 1.22e-2 | 7.74e-3 | 7.83e-3 | 9.05e-3 | 9.21e-3 | 8.19e-3 | 1.39e-2 | **1.39e-2** | 2.71e-2 | 3.70e-2 | 2.12e-2 | 1.17e-2 | 2.53e-2 |
| | 2 | P | 7.55e-3 | 4.63e-3 | 3.02e-3 | 5.94e-3 | 5.29e-3 | 6.03e-3 | 1.49e-2 | 2.02e-2 | 4.22e-2 | 4.48e-2 | 4.96e-3 | 3.17e-2 | 1.36e-2 |
| | 2 | F | 6.54e-2 | 3.07e-2 | 3.61e-2 | 4.30e-2 | 4.368e-2 | 2.28e-2 | 7.86e-2 | 1.20e-1 | 1.53e-1 | 8.05e-2 | 3.17e-2 | 3.14e-1 | 1.14e-1 |

Upon our investigation in the main text, we explore the inter- and intra-problem generalization of transformers with ICL, i.e., the performance of the model in ICL scenarios where the distribution of in-context examples (ICE) differs from the target test data. In real-world applications, the ICE provided to the model may not perfectly reflect the target application's data distribution. Therefore, a crucial question arises: how do biased input or noisy labels within ICL impact the intra-task, intra-, and inter-problem generalization of transformers?

We consider three critical scenarios when ICL is biased: (1) label noise within ICE, (2) distribution shifts in input data between ICE and test samples, and (3) the combined effect of distribution shifts and noisy labeling. We aim to investigate how these challenges influence the model's ability to fit known functions and generalize to unseen function combinations. This may help to understand the model's robustness and susceptibility to various forms of data uncertainty for assessing its real-world applicability in scenarios where perfect data alignment is often unattainable.

### C.1   LABEL NOISE

We explore the impact of noisy labels within ICE on the efficacy of ICL. Following Ahuja & Lopez-Paz (2023), we add randomly generated noise $\epsilon \sim N(0, I)$ to the outputs (labels) of ICE, where $I$ is the strength of noise and we set it to 1 and 2 in our experiments.

To investigate the effects of varying degrees of label corruption, we implemented two noise injection strategies: partial addition and full addition. In the partial addition setting, 10 noisy samples were introduced within the ICE, while the remaining samples maintained their original, unbiased labels. Conversely, the full addition strategy involved contaminating all ICE with noisy labels.

Table 9 and 10 shows the SE of the Baseline and ComFuncLearner model at the 40th point with noisy ICE, where each model is fed 39 ICE and predicts the 40th output. Figure 13 shows the curves the models learn after 39 ICE. They show a significant negative impact of label noise on the generalization ability of both models, particularly in the fitting of base functions. This suggests that the presence of inaccurate labels disrupts the models' intra-task and intra-problem generalization.

Moreover, we find a critical aspect of ICL: surprisingly high sensitivity to relatively few noisy samples. Introducing as few as 10 out of 39 ICE with label noise significantly compromised the model's generalization ability, irrespective of their position within the sequence.

### C.2   BIASED INPUT

We explore the effect of perturbing ICE with samples outside the test data's input range. Specifically, we control the test sample input sampled in the range $(-\pi, \pi)$ and the additional ICE at $(-2\pi, -\pi)$ and $(\pi, 2\pi)$, respectively and add them at the beginning and end of the in-context sequence.

Please note that the outputs of these additional ICE are still generated with the groud-truth function. While it might be expected that these additional inputs, due to their non-overlapping input range with the test samples and still retaining the ground truth function, would not significantly influence the predictions, our experiments revealed a surprising result. Adding these out-of-range ICE significantly hampered the model's generalization to the test samples.

Table 11: SE of the ComFuncLearner and Baseline model tested on the product combinations of base functions with biased ICE. The product combinations used via training phase is $\mathcal{F}_1^{(0)}$ and $\mathcal{F}_3^{(0)}$. *Type* indicates the type of bias in ICE and 0 indicates no bias. *In* and *I&O* indicate based input and based input and output, respectively.

| Model | Type | Pos | s(x) | c(x) | s(2x) | c(2x) | $Mean_B$ | s(x)&c(x) | s(x)&s(2x) | s(x)&c(2x) | c(x)&s(2x) | c(x)&c(2x) | s(2x)&c(2x) | $Com_{All}$ | $Mean_C$ |
|---|---|---|---|---|---|---|---|---|---|---|---|---|---|---|---|
| Baseline | 0 | - | 6.15e-5 | 8.56e-6 | 3.95e-5 | 6.99e-5 | 4.53e-5 | 1.91e-2 | 2.19e-3 | 1.02e-2 | 1.27e-2 | 1.62e-2 | 1.04e-3 | 9.87e-3 | 1.02e-2 |
| | In | 20 | 1.36e-1 | 8.59e-2 | 2.11e-1 | 6.07e-2 | 2.60e-1 | 1.44e-1 | 2.11e-1 | 5.41e-1 | 9.83e-2 | 2.11e-1 | 2.01e-1 | 1.07 | 3.54e-1 |
| | In | 40 | 1.51e-2 | 1.33e-2 | 4.91e-2 | 4.91e-2 | 2.47e-2 | 8.87e-2 | 1.99e-2 | 1.75e-1 | 1.56e-2 | 3.91e-2 | 7.20e-2 | 1.97e-1 | 8.70e-2 |
| | I&O | 20 | 9.95e-2 | 1.89e-1 | 9.73e-2 | 5.73e-2 | 8.14e-2 | 4.51e-2 | 2.21e-1 | 4.61e-1 | 1.60e-1 | 1.96e-1 | 1.83e-1 | 4.31e-2 | 1.87e-1 |
| | I&O | 40 | 2.78e-2 | 5.55e-2 | 1.72e-2 | 3.54e-2 | 3.40e-2 | 8.39e-2 | 1.03e-1 | 2.15e-1 | 5.39e-2 | 4.64e-2 | 4.87e-2 | 3.44e-1 | 1.27e-1 |
| CFL | 0 | - | 3.24e-5 | 1.34e-5 | 6.50e-5 | 7.31e-5 | 4.61e-5 | 8.45e-3 | 4.24e-5 | 6.19e-3 | 3.39e-3 | 8.93e-3 | 7.24e-4 | 4.86e-3 | 4.64e-3 |
| | In | 20 | 5.65e-2 | 3.25e-2 | 8.78e-2 | 4.86e-2 | 5.67e-2 | 1.96e-1 | 8.78e-2 | 4.68e-1 | 4.93e-2 | 8.73e-2 | 1.69e-1 | 6.97e-1 | 2.60e-1 |
| | In | 40 | 9.75e-3 | 2.13e-2 | 2.97e-2 | 2.92e-2 | 1.79e-2 | 1.01e-1 | 8.51e-3 | 7.02e-2 | 1.87e-2 | 3.93e-2 | 3.28e-2 | 1.03e-1 | 5.35e-2 |
| | I&O | 20 | 2.62e-1 | 1.91e-1 | 4.85e-2 | 2.73e-1 | 2.05e-1 | 4.76e-2 | 5.43e-1 | 1.24 | 2.52e-1 | 3.59e-1 | 4.04e-1 | 4.92e-2 | 4.15e-1 |
| | I&O | 40 | 3.62e-2 | 6.27e-2 | 1.87e-2 | 2.85e-2 | 3.65e-2 | 4.15e-2 | 6.07e-2 | 1.86e-1 | 5.03e-2 | 5.13e-2 | 3.56e-2 | 1.64e-1 | 8.58e-2 |

Table 12: SE of the ComFuncLearner and Baseline model tested on the convex combinations of base functions with biased ICE. *Type* indicates the type of bias in ICE and 0 indicates no bias. *In* and *I&O* indicate based input and based input and output respectively.

| Model | Type | Pos | s(x) | c(x) | s(2x) | c(2x) | $Mean_B$ | s(x)&c(x) | s(x)&s(2x) | s(x)&c(2x) | c(x)&s(2x) | c(x)&c(2x) | s(2x)&c(2x) | Com_All | $Mean_C$ |
|---|---|---|---|---|---|---|---|---|---|---|---|---|---|---|---|
| Baseline | 0 | - | 1.85e-5 | 1.15e-5 | 7.02e-5 | 8.29e-5 | 4.67e-5 | 3.37e-3 | 3.64e-2 | 2.54e-1 | 1.01e-1 | 1.39e-2 | 8.13e-3 | 1.67e-1 | 8.35e-2 |
| | In | 20 | 2.48e-1 | 1.16e-1 | 1.21e-1 | 3.53e-1 | 2.10e-1 | 9.36e-2 | 4.12e-1 | 6.32e-1 | 4.70e-1 | 3.89e-1 | 9.77e-2 | 1.65e0 | 5.353-1 |
| | In | 40 | 1.90e-1 | 1.04e-1 | 7.15e-2 | 2.69e-1 | 1.59e-1 | 7.73e-2 | 3.04e-1 | 6.06e-1 | 5.60e-1 | 3.21e-1 | 5.95e-2 | 9.88e-1 | 4.16e-1 |
| | I&O | 20 | 2.69e-1 | 1.21e-1 | 1.33e-1 | 3.90e-1 | 2.28e-1 | 7.91e-2 | 4.69e-1 | 9.84e-1 | 7.68e-1 | 4.79e-1 | 1.59e-1 | 1.74e0 | 6.71e-1 |
| | I&O | 40 | 1.86e-1 | 1.14e-1 | 5.68e-2 | 2.07e-1 | 1.41e-1 | 6.11e-2 | 3.04e-1 | 6.37e-1 | 5.80e-1 | 4.02e-1 | 7.19e-2 | 1.28e0 | 4.71e-1 |
| CFL | 0 | - | 4.47e-5 | 1.17e-5 | 1.69e-4 | 1.26e-4 | 8.79e-5 | 5.06e-3 | 3.78e-4 | 1.23e-3 | 5.65e-3 | 2.17e-3 | 1.09e-3 | 4.43e-3 | 2.86e-3 |
| | In | 20 | 1.52e-2 | 1.49e-2 | 6.52e-3 | 2.26e-2 | 1.48e-2 | 1.54e-1 | 8.93e-3 | 4.05e-2 | 1.39e-2 | 5.72e-3 | 1.37e-2 | 8.26e-2 | 3.32e-2 |
| | In | 40 | 1.31e-3 | 2.67e-3 | 5.06e-4 | 6.02e-3 | 2.63e-3 | 2.29e-3 | 6.46e-4 | 5.43e-3 | 9.24e-3 | 9.35e-3 | 2.21e-3 | 6.57e-3 | 5.11e-3 |
| | I&O | 20 | 2.94e-2 | 2.67e-2 | 2.31e-2 | 8.39e-2 | 4.09e-2 | 3.27e-2 | 4.03e-2 | 9.12e-2 | 6.62e-2 | 6.32e-2 | 7.76e-2 | 1.27e-1 | 7.13e-2 |
| | I&O | 40 | 2.59e-3 | 3.09e-3 | 1.25e-3 | 2.23e-2 | 7.32e-3 | 3.14e-3 | 3.50e-3 | 1.27e-2 | 1.08e-2 | 6.34e-3 | 2.02e-2 | 1.38e-1 | 1.01e-2 |

Table 11, 12 and Figure 15 show the effect of augmenting ICE with 10 out-of-range input samples both before and after the standard sequence of 20 examples, respectively. Here we set the combination of base functions to the product combinations and more results with other combinations are in Appendix.

Both the Baseline and ComFuncLearner model exhibits a vulnerability to augmentation with out-of-range samples. Notably, their performance significantly declines not only on unseen combinations of base functions but also on the fitting of seen base functions. This counterintuitive observation holds regardless of whether the out-of-range samples are placed at the beginning or end of the in-context sequence. Thus the presence of biased input in ICE disrupts both intra-task and intra-problem generalization of transformers.

## C.3  BIASED INPUT AND OUTPUT

To investigate the influence of severely biased examples on ICL, we implement a scenario where some ICE intentionally lie outside the test data's input range whose output is biased. Specifically, we confine the test sample input range to $(-\pi, \pi)$ while placing additional OOD examples at $(-2\pi, -\pi)$ and $(\pi, 2\pi)$. we deliberately introduce a constant offset of 10 to their outputs, modifying them to $y = f_i(x; \phi_i) + 10$, where $f_i(x; \phi_i)$ is the ground-truth function for the current sequence and $y$ is the outputs of ICE. These OOD examples are inserted at the beginning and end of the in-context sequence. Similar to Section C.2, as shown in Table 11 and Figure 15, both Baseline and ComFuncLearner models demonstrate remarkable susceptibility to ICE with biased input and output. The trend persists regardless of the placement of out-of-range samples within the sequence, suggesting that the presence of biased input disrupts both intra-task and intra-problem generalization of transformers.

## D  SUPPLEMENTARY RESULTS

### D.1  FUNCTION FITTING ON PRETRAINED GPT-2

At Section 3.1 and Section 3.2, we take the base functions $\mathcal{F}_1^{(0)}, \mathcal{F}_2^{(0)}, \mathcal{F}_3^{(0)}, \mathcal{F}_4^{(0)}$ defined in 2.3, and the combination of $\mathcal{F}_1^{(0)}, \mathcal{F}_3^{(0)}$. To support the phenomenon that transformers gain ability to solve intra-problem generalization when trained with corresponding combinations of base functions, we test on different settings to further validate the conclusion.

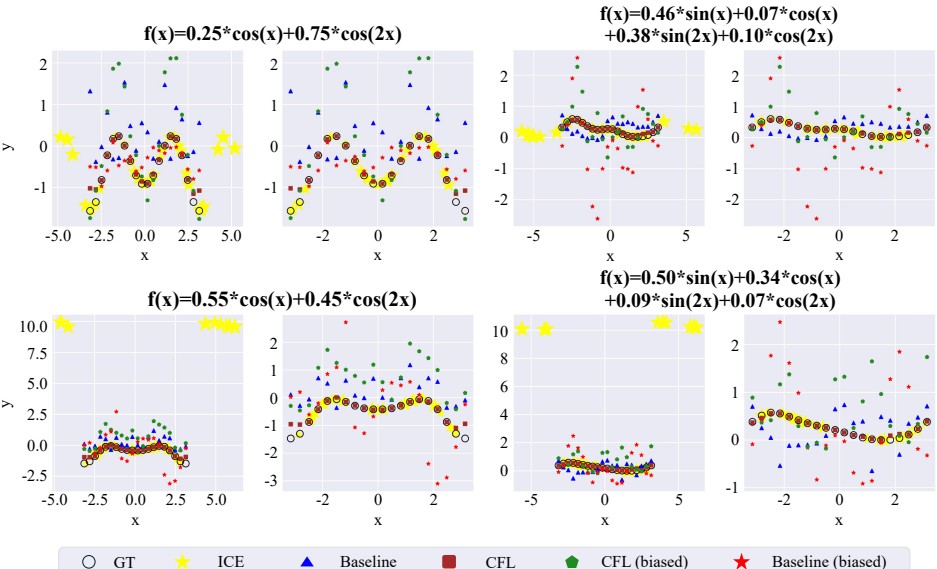

Figure 15: Function curves on the product combinations of base functions fitted by the Baseline and ComFuncLearner models after ICL with biased ICE. The product combination used via training is $\mathcal{F}_1$ and $\mathcal{F}_3$. Upper figures give points outside the domain of definition during the training phase (*Type I*), and the lower figures additionally modify the value of y outside the domain. (*Type I&O*)

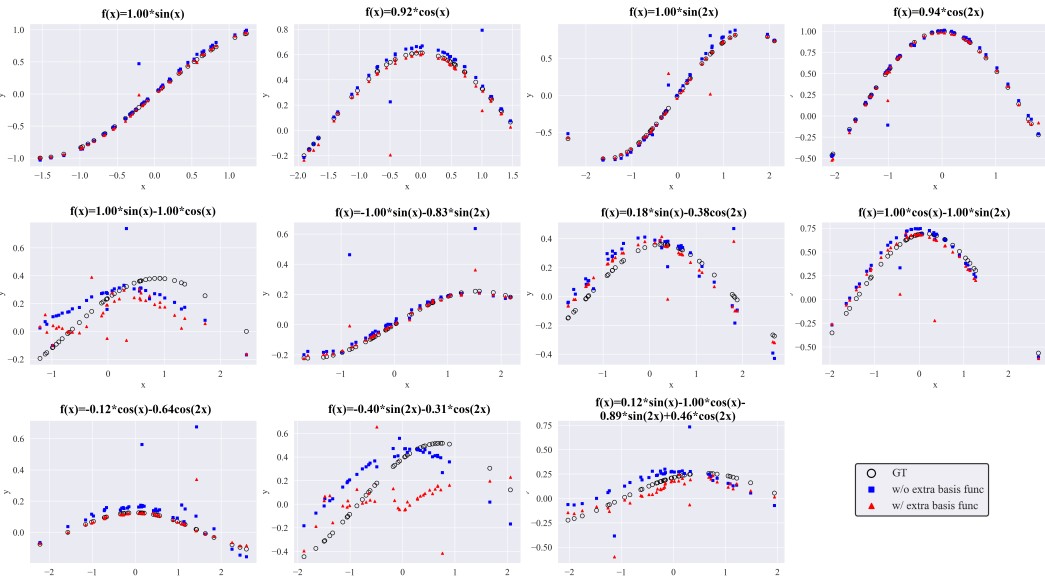

Figure 16: Function curves of the two Baseline models tested on the convex combinations of base functions. The shared base functions are $\mathcal{F}_1^{(0)}, \mathcal{F}_2^{(0)}, \mathcal{F}_3^{(0)}, \mathcal{F}_4^{(0)}$. Model w/o extra basis function means it is trained on $\mathcal{F}_5^{(0)}$ additionally, while model w/ extra basis function indicates that it is trained only on shared base functions. We keep total training steps the same.

**Baseline Learned with 4 Base Functions**  To verify whether cross-problem generalization can be achieved solely by increasing the amount of data, we conducted additional experiments where the model was trained with more data from only 4 base functions and tested on various function fitting tasks. The results are shown in Table 13. The conclusion drawn from these experiments is consistent with the main text: the transformer model only achieves intra-task and intra-problem generalization, but not inter-problem generalization.

Table 13: SE of the ComFuncLearner and Baseline model tested on the convex combinations of base functions, where the Baseline model is learned on 4 base functions and the same number of samples as the ComFuncLearner model.

| Range | Model | s(x) | c(x) | s(2x) | c(2x) | Mean$_B$ | s(x)&c(x) | s(x)&s(2x) | s(x)&c(2x) | c(x)&s(2x) | c(x)&c(2x) | s(2x)&c(2x) | Com$_{All}$ | Mean$_C$ |
|---|---|---|---|---|---|---|---|---|---|---|---|---|---|---|
| 1-10 | Baseline | 1.31e-2 | 3.82e-2 | 1.26e-2 | 4.26e-2 | 2.66e-2 | 1.98e-2 | 4.48e-3 | 1.99e-2 | 1.49e-2 | 5.95e-3 | 1.16e-2 | 1.78e-2 | 1.35e-2 |
|  | CFL | 2.63e-1 | 2.48e-1 | 6.46e-2 | 1.81e-1 | 1.89e-1 | 8.01e-2 | 8.02e-3 | 5.34e-2 | 1.39e-2 | 1.38e-2 | 2.69e-2 | 8.63e-3 | 2.93e-2 |
| 11-20 | Baseline | 5.99e-4 | 8.37e-4 | 1.94e-4 | **4.75e-4** | 5.26e-4 | **2.11e-3** | 5.24e-5 | 4.07e-3 | 7.33e-3 | 5.92e-5 | 1.97e-3 | 8.19e-3 | 3.40e-3 |
|  | CFL | 6.86e-3 | 1.07e-2 | 2.11e-3 | 1.39e-2 | 8.38e-3 | 3.32e-3 | 2.31e-4 | 3.82e-3 | 3.82e-3 | 4.48e-4 | 2.18e-3 | 4.89e-3 | 2.67e-3 |
| 21-30 | Baseline | **4.40e-4** | 1.00e-3 | 1.95e-4 | 5.77e-4 | 5.53e-4 | 2.27e-3 | 4.24e-5 | 3.66e-3 | 6.99e-3 | **5.31e-5** | 1.91e-3 | 7.60e-3 | 3.22e-3 |
|  | CFL | 6.73e-3 | 9.45e-3 | 2.58e-3 | 1.56e-2 | 8.60e-3 | 2.70e-3 | 2.41e-4 | **3.44e-3** | 4.00e-3 | 5.07e-4 | 2.49e-3 | 4.71e-3 | **2.58e-3** |
| 31-40 | Baseline | 4.60e-4 | **7.78e-4** | **1.56e-4** | 5.44e-4 | **4.84e-4** | 2.24e-3 | **3.79e-5** | 4.05e-3 | 7.27e-3 | 5.94e-5 | **1.87e-3** | 7.81e-3 | 3.33e-3 |
|  | CFL | 7.09e-3 | 9.16e-3 | 3.05e-3 | 1.24e-2 | 7.94e-3 | 2.99e-3 | 2.24e-4 | 3.57e-3 | **3.70e-3** | 5.53e-4 | 3.10e-3 | **4.70e-3** | 2.69e-3 |

Table 14: SE of the ComFuncLearner and Baseline model tested on the convex combinations of base functions, where the Baseline model is learned on 4 base functions and the same number of samples as the ComFuncLearner model. The convex combination is $\mathcal{F}_1^{(0)} + \mathcal{F}_2^{(0)}$

| Range | Model | s(x) | c(x) | s(2x) | c(2x) | Mean$_B$ | s(x)&c(x) | s(x)&s(2x) | s(x)&c(2x) | c(x)&s(2x) | c(x)&c(2x) | s(2x)&c(2x) | Com$_{All}$ | Mean$_C$ |
|---|---|---|---|---|---|---|---|---|---|---|---|---|---|---|
| 1-10 | Baseline | 5.21e-2 | 9.57e-2 | 8.29e-2 | 2.18e-2 | 6.31e-2 | 1.41e-2 | 5.23e-3 | 2.07e-2 | 1.71e-2 | 4.85e-3 | 1.33e-2 | 2.02e-2 | 1.36e-2 |
|  | CFL | 8.64e-2 | 1.17e+00 | 2.37e-1 | 8.76e-2 | 3.95e-1 | 7.03e-3 | 7.64e-3 | 7.66e-3 | 2.45e-3 | 1.99e-3 | 1.82e-3 | 1.58e-3 | 4.31e-3 |
| 11-20 | Baseline | 7.87e-4 | 4.63e-3 | 7.01e-4 | 5.32e-4 | 1.66e-3 | 2.64e-3 | 1.71e-4 | 5.67e-3 | 8.18e-3 | 4.33e-5 | 5.03e-3 | 8.90e-3 | 4.38e-3 |
|  | CFL | 4.36e-4 | 1.82e-3 | **2.98e-4** | 1.81e-4 | 6.85e-4 | 3.99e-5 | 3.51e-5 | 4.28e-5 | 2.60e-5 | 1.17e-5 | 1.71e-5 | 1.83e-5 | 2.73e-5 |
| 21-30 | Baseline | 4.42e-4 | 5.08e-3 | 4.66e-4 | 4.00e-4 | 1.60e-3 | 2.73e-3 | 8.21e-5 | 5.18e-3 | 7.53e-3 | 4.30e-5 | 5.27e-3 | 7.97e-5 | 4.12e-3 |
|  | CFL | **1.72e-4** | 2.17e-3 | 4.16e-4 | 1.79e-4 | 7.35e-4 | **3.41e-5** | 2.88e-5 | 2.94e-5 | 2.36e-5 | **1.13e-5** | **1.52e-5** | 1.62e-5 | 2.27e-5 |
| 31-40 | Baseline | 4.53e-4 | 3.02e-3 | 5.14e-4 | 5.63e-4 | 1.14e-3 | 2.43e-3 | 1.60e-4 | 5.47e-3 | 7.14e-3 | 4.53e-5 | 4.95e-3 | 8.52e-3 | 4.10e-3 |
|  | CFL | 1.91e-4 | **1.81e-3** | 3.36e-4 | **1.78e-4** | **6.29e-4** | 3.79e-5 | **2.30e-5** | **2.90e-5** | **2.13e-5** | 1.14e-5 | 1.56e-5 | **1.60e-5** | **2.20e-5** |

**Convex Combination Generalization with ComFuncLearner Learning other Combinations** Here, we keep the base function classes defined in Section 3.1, and let the ComFuncLearner model leverages convex combination of $\mathcal{F}_2^{(0)}$ and $\mathcal{F}_4^{(0)}$, i.e., $\mathcal{F}_2^{(0)} + \mathcal{F}_4^{(0)}$. The Baseline model is trained on all base function classes and $\mathcal{F}_5^{(0)} = \{\phi\sin(3x)\}, \phi \in \mathbb{R}$. Results are shown in Figure 17. We also carry out experiment on $\mathcal{F}_1^{(0)} + \mathcal{F}_2^{(0)}$. Results can be found at Figure 18 and Table 14.

**Composition Generalization** At Section 3.3, we select four base function classes $\mathcal{F}_1^{(0)}, \mathcal{F}_2^{(0)}, \mathcal{F}_3^{(0)}, \mathcal{F}_4^{(0)}$. To evaluate the significance of baseline function classes, we implemented class numbers to 16. The base functions classes now include $\mathcal{F}_i^{(0)}, i = 1, 2, .., 16$, namely $\{\phi\sin(kx), k = 1, 2, ..8\} \cup \{\phi\cos(k \cdot x), k = 1, 2, ..8\}$. Other settings keep the same. The result showed at Figure 19.

## D.2 FUNCTION FITTING ON PRETRAINED LLAMA SERIES

To further validate our findings, we conducted additional experiments focusing on convex and multiply combinations on pretrained LLaMA-3. These experiments aimed to assess the model's performance for intra-problem and intra-task generalization.

Figure 20 indicates that when exposed to convex combinations, the model still struggles with inter-problem generalization, while successfully leveraging its prior knowledge to address intra-problem and intra-task generalization.

These additional experiments reinforce our assertion that the diversity and representativeness of the finetuning dataset play a pivotal role in enhancing the model's generalization capabilities.

We carried the same experiments on LLaMA-2-7b-hf and found similar conclusion. Results can be found at Figure 21, 22.

## D.3 REVERSE INTER-PROBLEM AND INTRA-PROBLEM

To investigate whether the observed inter- and intra-problem generalization behaviors hold when the roles of base functions and their combinations are reversed, we conducted the following supplementary experiment. In contrast to the experimental setup in the main text, we now explore whether a model exposed to diverse combinations of base functions can generalize to base function fitting tasks.

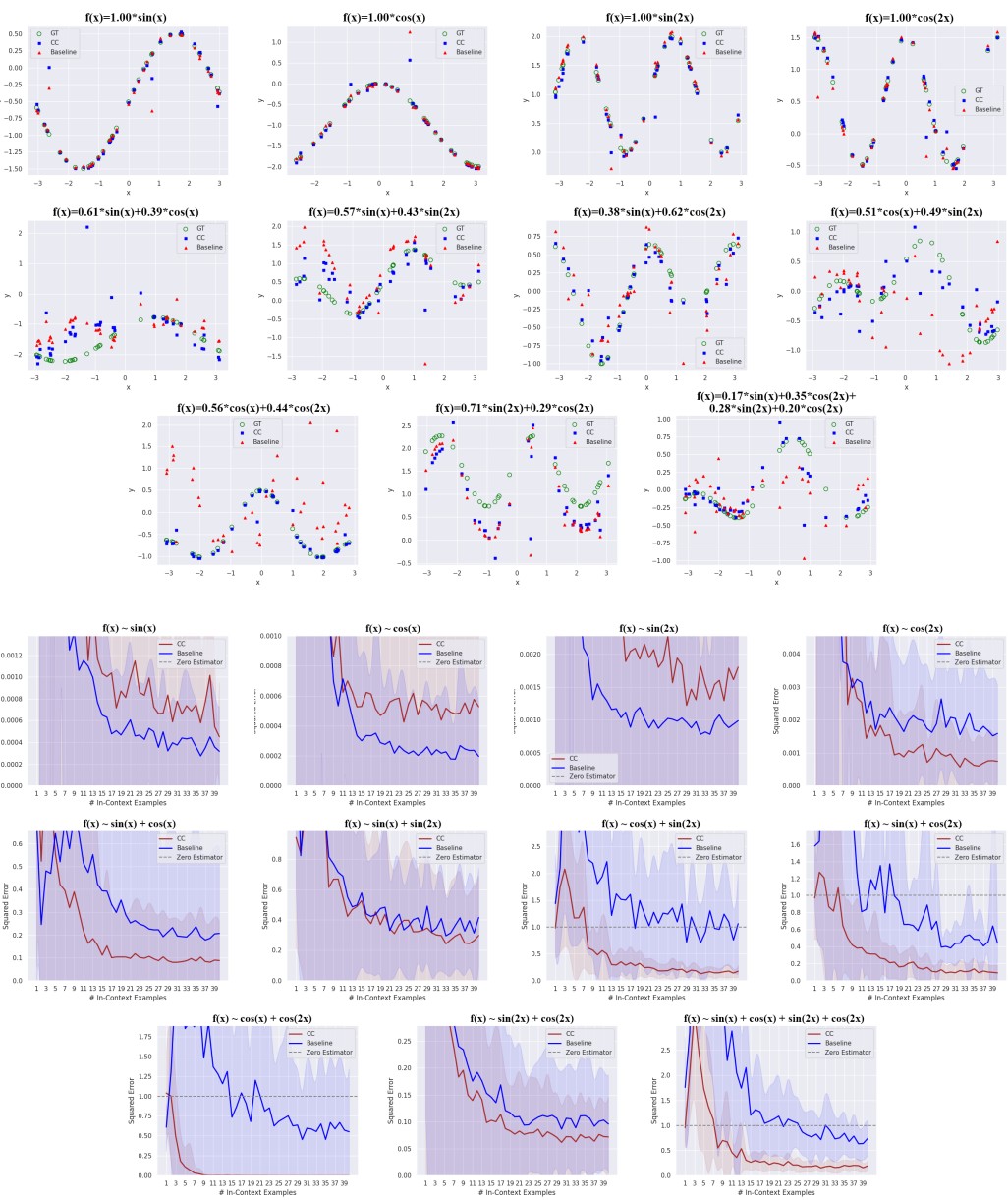

Figure 17: Results of the ComFuncLearner and Baseline model tested on the convex combinations of base functions. The convex combinations used via training phase is $\mathcal{F}_2^{(0)}$ and $\mathcal{F}_4^{(0)}$. *GT* refers to ground truth. *CC* refers to ComFuncLearner and *Zero Estimator* refers to constant function $f(x) = 1$.

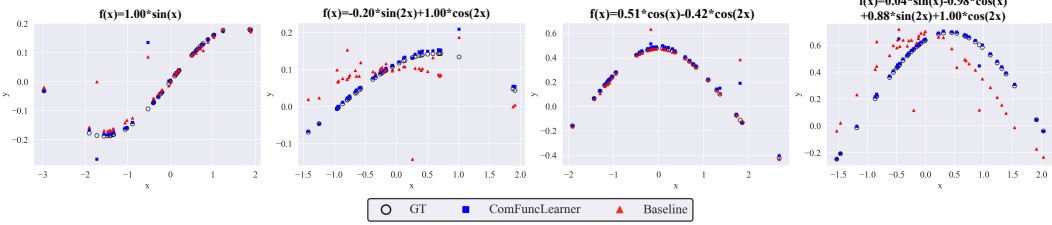

Figure 18: Function curves of the two Baseline models tested on the convex combinations of base functions. The base functions are $\mathcal{F}_1^{(0)}, \mathcal{F}_2^{(0)}, \mathcal{F}_3^{(0)}, \mathcal{F}_4^{(0)}$. We change the convex combination from $\mathcal{F}_1^{(0)} + \mathcal{F}_3^{(0)}$ to $\mathcal{F}_1^{(0)} + \mathcal{F}_2^{(0)}$

Table 15: SE of the ComFuncLearner and Baseline model, which trained on reversed data, tested on the product combinations of base functions.

| Model | Steps | s(x) | c(x) | s(2x) | c(2x) | Mean$_B$ | s(x)&c(x) | s(x)&s(2x) | s(x)&c(2x) | c(x)&s(2x) | c(x)&c(2x) | s(2x)&c(2x) | Com$_{All}$ | Mean$_C$ |
|---|---|---|---|---|---|---|---|---|---|---|---|---|---|---|
| Baseline | 10k | 2.08e-2 | 2.70e-2 | 2.03e-2 | 2.92e-2 | 2.43e-2 | 3.14e-2 | 2.16e-2 | 3.08e-2 | 3.11e-2 | 1.96e-1 | 2.75e-2 | 1.63e-1 | 7.17e-2 |
| | 20k | 1.49e-2 | 1.67e-2 | 1.53e-2 | 1.82e-2 | 1.63e-2 | 1.82e-2 | 1.51e-2 | 1.70e-2 | 1.72e-2 | 1.50e-1 | 1.68e-2 | 1.79e-1 | 5.90e-2 |
| | 30k | 1.30e-2 | 1.57e-2 | 1.32e-2 | 1.46e-2 | 1.41e-2 | 1.63e-2 | 1.30e-2 | 1.59e-2 | 1.64e-2 | 1.67e-1 | 1.66e-2 | 1.50e-1 | 5.64e-2 |
| | 40k | 1.23e-2 | **1.33e-2** | 1.18e-2 | 1.36e-2 | 1.28e-2 | **1.33e-2** | 1.08e-2 | **1.26e-2** | 1.36e-2 | 1.61e-1 | **1.37e-2** | 1.69e-1 | 5.63e-2 |
| | 50k | 1.04e-2 | 1.62e-2 | 1.04e-2 | 1.79e-2 | 1.37e-2 | 1.51e-2 | 1.06e-2 | 1.54e-2 | 1.63e-2 | 1.48e-1 | **1.37e-2** | 1.05e-1 | 4.62e-2 |
| CFL | 10k | 2.10e-2 | 2.42e-2 | 2.07e-2 | 2.46e-2 | 2.26e-2 | 2.54e-2 | 2.05e-2 | 2.46e-2 | 2.43e-2 | 2.04e-1 | 2.49e-2 | 2.29e-1 | 7.90e-2 |
| | 20k | 1.04e-2 | 1.57e-2 | 1.11e-2 | 1.63e-2 | 1.34e-2 | 1.54e-2 | 1.12e-2 | 1.54e-2 | 1.58e-2 | 1.71e-1 | 1.73e-2 | 1.16e-1 | 5.18e-2 |
| | 30k | 9.20e-3 | 1.43e-2 | 8.24e-3 | 1.55e-2 | 1.18e-2 | 1.34e-2 | 8.89e-3 | 1.51e-2 | **1.28e-2** | 1.40e-1 | 1.46e-2 | 1.77e-1 | 5.46e-2 |
| | 40k | 1.02e-2 | 1.66e-2 | 9.33e-3 | 1.51e-2 | 1.28e-2 | 1.42e-2 | 9.14e-3 | 1.35e-2 | 1.56e-2 | **1.11e-1** | 1.52e-2 | 1.73e-1 | 5.02e-2 |
| | 50k | **8.57e-3** | 1.38e-2 | **7.33e-3** | **1.36e-2** | **1.08e-2** | 1.36e-2 | **8.50e-3** | 1.31e-2 | 1.44e-2 | 1.65e-1 | 1.51e-2 | 1.35e-1 | 5.21e-2 |

The results demonstrate that our findings hold even when we reverse the roles of composite functions and base functions, treating function combinations as $\mathcal{T}_{base}$ and the base functions (which can be approximated as combinations of multiple combinations of functions) as $\mathcal{T}_{com}$. This further reinforces our conclusion that transformers with ICL enable intra-problem generalization but not inter-problem generalization.

- The **Baseline model** is trained solely on combinations of 4 functions: $\{\mathcal{F}_1^{(0)}+\mathcal{F}_2^{(0)}, \mathcal{F}_1^{(0)}+\mathcal{F}_4^{(0)}, \mathcal{F}_2^{(0)}+\mathcal{F}_3^{(0)}, \mathcal{F}_3^{(0)}+\mathcal{F}_4^{(0)}\}$.
- The **ComFuncLearner** is trained on the same 4 function combinations as the Baseline model, but is additionally exposed to one base function: $\{\mathcal{F}_1^{(0)}\}$.

The results are shown in table 15.

### D.4 CURRICULUM LEARNING

To investigate the influence of data point order and sample difficulty on the generalization ability of the transformer, we designed the following experiments:

For the **ComFuncLearner**, we designed a two-stage training process. In the first 25,000 iterations, the model learns only the base functions. Subsequently, in the next 25,000 iterations, it is exposed to the combined functions. We then compare the performance of this model with a ComFuncLearner trained on a random sequence of samples. The results are presented in Table 16.

We find that training the model on base functions first, followed by combined functions, does not further improve its generalization ability. In fact, it results in worse generalization performance compared to a model trained with randomly sampled data. This could be attributed to the need for careful tuning of the ratio of iterations dedicated to learning base functions versus combined functions to achieve optimal performance.

### D.5 GENERALIZATION ON OTHER BASE FUNCTION CLASSES

Aiming at analyzing the model's ability to generalize across different function classes, we conduct experiments on another set of base functions. Inspired by Legendre polynomials, we select new base functions as follows.

- $y_{\text{L}}^{(1)} = \frac{\sqrt{30}}{50} \cdot \mathbf{x} \cdot |\mathbf{w}|$

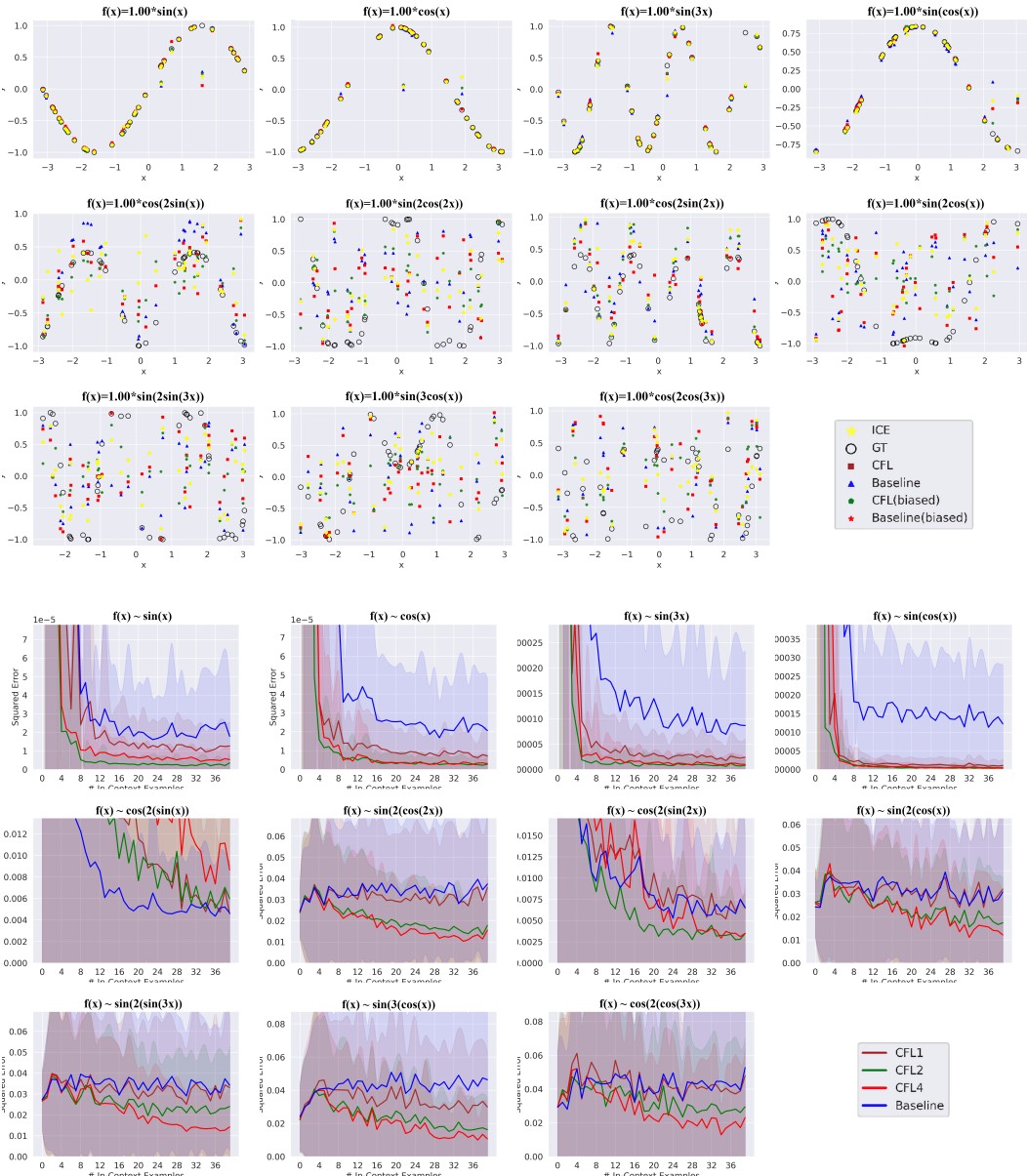

Figure 19: Result of of the ComFuncLearner and Baseline model tested on the composition combinations of base functions. To delve deeper into the effect of base function diversity, we increase the function number to 16 ($\mathcal{F}_i^{(0)}, i = 1, 2, .., 16$). However, we do not find significantly drop in SE compared to six functions in 3.3 with at most four combinations

Table 16: SE of the model tested on the product combinations of base functions. CFL1_CL means CFL1 learned with curriculum learning.

| Model | s(x) | c(x) | s(2x) | c(2x) | Mean_B | s(x)&c(x) | s(x)&s(2x) | s(x)&c(2x) | c(x)&s(2x) | c(x)&c(2x) | s(2x)&c(2x) | Com_All | Mean_C |
|---|---|---|---|---|---|---|---|---|---|---|---|---|---|
| Baseline | 2.52e-5 | 7.24e-6 | **1.61e-5** | 3.71e-5 | **2.17e-5** | 2.69e-2 | 7.76e-3 | 2.29e-2 | 8.86e-3 | **8.52e-3** | 5.21e-3 | 5.06e-2 | 1.98e-2 |
| CFL1 | **1.82e-5** | **6.75e-6** | 3.65e-5 | **3.07e-5** | 2.31e-5 | **3.16e-3** | **4.99e-5** | 7.58e-3 | 2.84e-2 | 1.16e-2 | **1.59e-3** | **9.88e-3** | **5.24e-3** |
| CFL1_CL | 3.38e-3 | 7.62e-3 | 3.45e-3 | 7.52e-3 | 5.49e-3 | 7.81e-3 | 2.89e-3 | **7.46e-3** | **8.02e-3** | 1.60e-2 | 8.34e-3 | 4.06e-2 | 9.92e-3 |

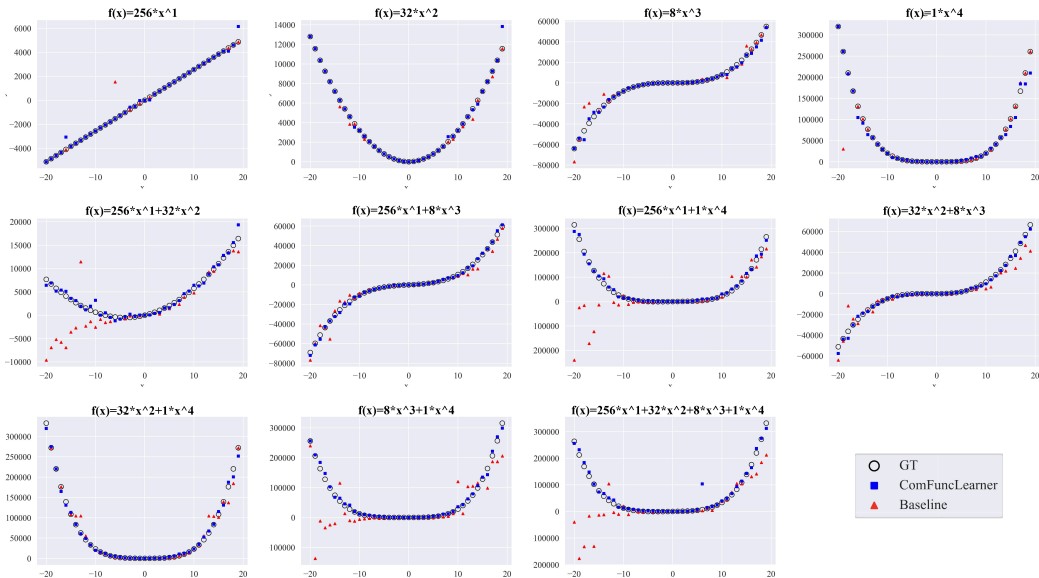

Figure 20: The performance of LLaMA-3 on convex combinations of learned functions ($x$, $x^2$, $x^3$ and $x^4$). *Baseline* indicates LLaMA-3-8B finetuned with base functions. *ComFuncLearner* indicates LLaMA-3-8B finetuned with base functions and the combination of $x^2$ and $x^3$.

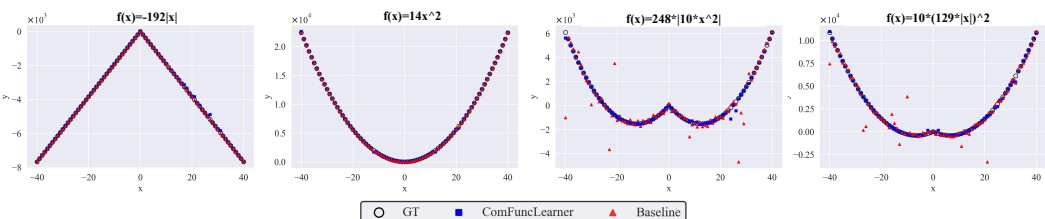

Figure 21: The fitted curves of LLaMa-2 on composition combinations of learned functions (*absolute value functions* $-|\mathbf{x}|$ and *quadratic function* $\mathbf{x^2}$).

- $y_L^{(2)} = \frac{\sqrt{2}}{50} \cdot \left(3\mathbf{x}^2 - 25\right) \cdot \mathbf{w}$

- $y_L^{(3)} = \frac{\sqrt{70}}{500} \cdot \left(\mathbf{x}^3 - 15\mathbf{x}\right) \cdot \mathbf{w}$

- $y_L^{(4)} = \frac{3\sqrt{10}}{10000} \cdot \left(7\mathbf{x}^4 - 150\mathbf{x}^2 + 375\right) \cdot \mathbf{w}$

The training data was derived from these functions, and the model was subsequently evaluated on various function fitting tasks. The results, presented in Figure 23 and Table 17, align with the findings in the main text: while the transformer model demonstrates strong intra-task and intra-problem generalization, it fails to achieve inter-problem generalization, even when trained on mathematically diverse base functions like the polynomials.

Table 17: SE of models tested on the convex combinations of polynomials.

| Range | Model | $x$ | $x^2$ | $x^3$ | $x^4$ | $\text{Mean}_B$ | $x\&x^2$ | $x\&x^3$ | $x\&x^4$ | $x^2\&x^3$ | $x^2\&x^4$ | $x^3\&x^4$ | $\text{Com}_{All}$ | $\text{Mean}_C$ |
|---|---|---|---|---|---|---|---|---|---|---|---|---|---|---|
| 1-10 | Baseline | 2.11e-1 | 5.96e-2 | 1.81e-1 | 3.58e-1 | 2.02e-1 | 1.54e-2 | 2.40e-2 | 7.55e-3 | 2.31e-2 | 8.59e-3 | 3.95e-3 | 4.95e-3 | 1.25e-2 |
| | CFL | 2.02e-1 | 5.57e-2 | 1.76e-1 | 2.52e-1 | 1.72e-1 | 1.52e-2 | 1.26e-2 | 7.40e-3 | 1.21e-2 | 8.24e-3 | 3.42e-3 | 4.91e-3 | 9.12e-3 |
| 11-20 | Baseline | 2.48e-2 | 3.35e-3 | 1.21e-3 | 2.10e-2 | 1.26e-2 | 8.98e-3 | 7.23e-3 | 3.02e-3 | 7.53e-3 | 5.10e-3 | 8.14e-4 | 3.92e-3 | 4.88e-3 |
| | CFL | 2.36e-2 | 1.60e-2 | 6.72e-3 | 5.02e-2 | 2.41e-2 | 6.53e-3 | 3.52e-3 | 4.26e-3 | 2.15e-3 | 5.02e-3 | 7.22e-4 | 3.94e-3 | 3.73e-3 |
| 21-30 | Baseline | 1.50e-2 | 5.67e-2 | 1.18e-3 | **1.79e-2** | 9.93e-3 | 7.47e-3 | 7.41e-3 | 2.55e-3 | 7.94e-3 | 3.97e-3 | 8.90e-4 | 4.35e-3 | 4.71e-3 |
| | CFL | 2.11e-2 | 2.01e-2 | **6.56e-3** | 3.45e-2 | 2.05e-2 | 5.85e-3 | **3.47e-3** | 3.53e-3 | 1.91e-3 | 4.26e-3 | 6.53e-4 | **4.09e-3** | 3.39e-3 |
| 31-40 | Baseline | 5.69e-3 | 4.25e-3 | 1.42e-3 | 1.84e-2 | **7.45e-3** | 7.34e-3 | 6.65e-3 | 3.99e-3 | 8.26e-3 | 4.99e-3 | 7.99e-4 | 4.64e-3 | 4.72e-3 |
| | CFL | **1.91e-2** | **1.43e-2** | 7.72e-3 | 2.37e-2 | 1.62e-2 | **5.42e-3** | 3.71e-3 | **3.00e-3** | **1.85e-3** | **3.93e-3** | 6.51e-4 | 4.11e-3 | **3.23e-3** |

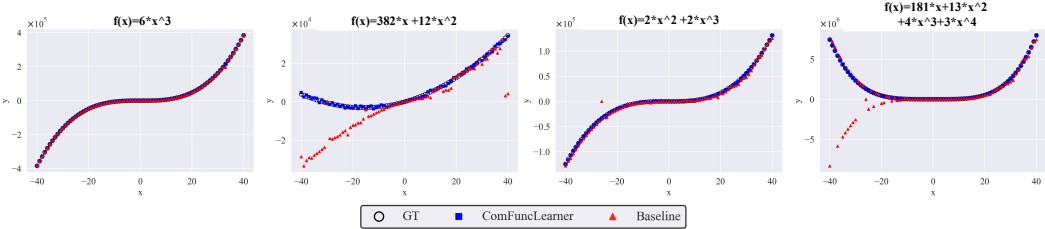

Figure 22: The fitted curves of LLaMa-2 on composition combinations of learned functions ($\mathbf{x}, \mathbf{x^2}, \mathbf{x^3}$ and $\mathbf{x^4}$)

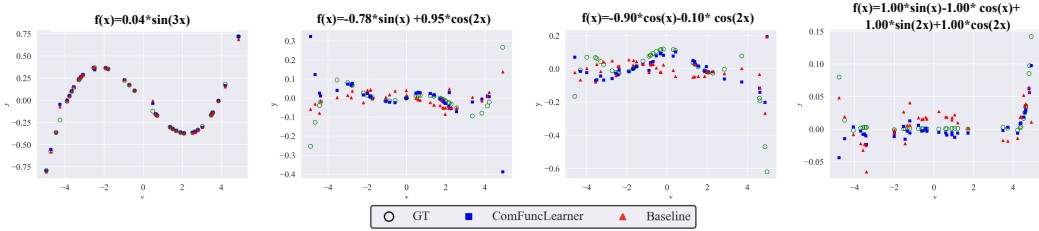

Figure 23: Function curves of the ComFuncLearner and Baseline model tested on the convex combinations of base polynomials.

### D.6 GENERALIZATION ON OTHER STRUCTURES

We conduct further experiments with other model architectures, including MLP and KAN. The results are presented in Table 18, 19 and Figure 24, 25.

We are surprised to find that the generalization behavior observed in transformers persists in these other network architectures. This opens up exciting avenues for future research. It is important to note that this should not be strictly termed "in-context learning" for MLP and KAN. Since these models require fixed input and output dimensions, our experimental setup necessitates feeding 39 (x, f(x)) pairs along with the 40-th x as input, and the model predicts only the 40th f(x). This differs from the sequential prediction capability of transformers in ICL. However, we believe that our findings regarding generalization limitations may extend beyond transformers and ICL, potentially applying to other deep learning architectures and tasks.

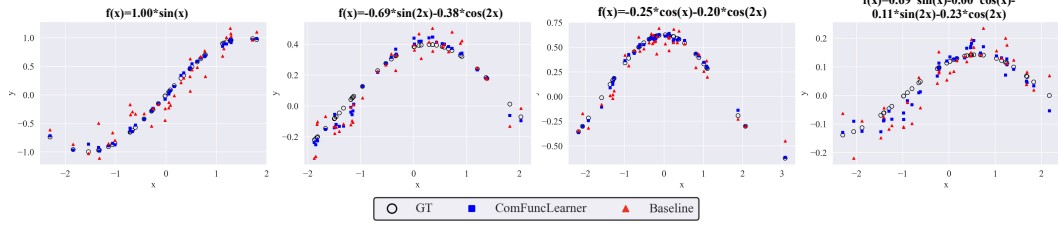

Figure 24: Function curves of models trained on MLP architecture. The setting is the same as Section 3.

Table 18: SE of models trained on MLP architecture. The setting is the same as Section 3.

| Range | Model | s(x) | c(x) | s(2x) | c(2x) | Mean_B | s(x)&c(x) | s(x)&s(2x) | s(x)&c(2x) | c(x)&s(2x) | c(x)&c(2x) | s(2x)&c(2x) | Com_All | Mean_C |
|---|---|---|---|---|---|---|---|---|---|---|---|---|---|---|
| 1-10 | Baseline | 3.07e-1 | 1.77e-2 | 1.67e-2 | 1.32e-1 | 1.18e-1 | 7.83e-3 | 8.70e-4 | 1.03e-3 | 1.41e-3 | 2.52e-4 | 3.06e-3 | 2.64e-4 | 2.10e-3 |
|  | CFL | 5.98e-2 | 1.34e-3 | **1.57e-3** | 2.16e-2 | 2.11e-2 | 4.42e-4 | 1.59e-4 | 1.86e-4 | 2.88e-4 | 7.21e-5 | 3.78e-4 | 4.61e-5 | 2.24e-4 |
| 11-20 | Baseline | 3.00e-1 | 1.19e-2 | 1.01e-2 | 1.47e-1 | 1.17e-1 | 4.59e-3 | 9.85e-4 | 9.64e-4 | 3.36e-3 | 4.16e-4 | 2.91e-3 | 2.70e-4 | 1.93e-3 |
|  | CFL | 4.89e-2 | 9.13e-4 | 2.08e-3 | 3.08e-2 | 2.07e-2 | 6.89e-4 | 1.57e-4 | 2.16e-4 | 2.60e-4 | 7.99e-5 | 2.81e-4 | 5.83e-5 | 2.49e-4 |
| 21-30 | Baseline | 3.89e-1 | 1.00e-2 | 8.93e-3 | 1.76e-1 | 1.46e-1 | 5.79e-3 | 8.44e-4 | 9.00e-4 | 1.56e-3 | 4.24e-4 | 2.35e-3 | 2.76e-4 | 1.74e-3 |
|  | CFL | 4.70e-2 | 1.88e-3 | 1.72e-3 | **1.78e-2** | 1.71e-2 | 5.90e-4 | 1.67e-4 | 1.91e-4 | **2.40e-4** | 6.67e-5 | 3.28e-4 | **4.41e-5** | 2.32e-4 |
| 31-40 | Baseline | 1.82e-1 | 2.23e-2 | 1.71e-2 | 1.76e-1 | 9.93e-2 | 4.59e-3 | 8.41e-4 | 1.05e-3 | 2.25e-3 | 3.90e-4 | 3.86e-3 | 2.54e-4 | 1.89e-3 |
|  | CFL | **3.65e-2** | **9.05e-4** | 2.13e-3 | 2.48e-2 | **1.61e-2** | **3.23e-4** | **1.42e-4** | **1.78e-4** | 2.86e-4 | **5.08e-5** | **2.65e-4** | 4.65e-5 | **1.84e-4** |

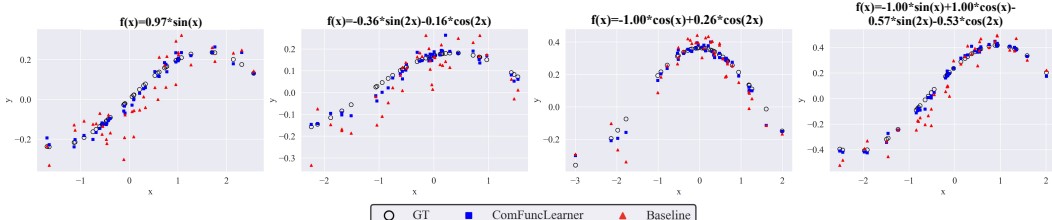

Figure 25: Function curves of models trained on KAN architecture. The setting is the same as Section 3.

Table 19: SE table of models trained on KAN architecture. The setting is the same as Section 3.

| Range | Model | s(x) | c(x) | s(2x) | c(2x) | Mean_B | s(x)&c(x) | s(x)&s(2x) | s(x)&c(2x) | c(x)&s(2x) | c(x)&c(2x) | s(2x)&c(2x) | Com_All | Mean_C |
|---|---|---|---|---|---|---|---|---|---|---|---|---|---|---|
| 1-10 | Baseline | 1.25e-2 | 2.26e-2 | 2.34e-2 | 9.50e-3 | 1.70e-2 | 3.98e-3 | 5.84e-3 | 4.42e-3 | 6.48e-3 | 1.19e-3 | 3.59e-3 | 2.92e-3 | 4.06e-3 |
|  | CFL | 7.50e-4 | 9.26e-4 | 1.79e-3 | 6.65e-4 | 1.04e-3 | 3.41e-4 | 2.73e-4 | 3.26e-4 | 4.93e-4 | 1.26e-4 | 2.58e-4 | 2.15e-4 | 2.90e-4 |
| 11-20 | Baseline | 9.84e-3 | 1.14e-2 | 2.31e-2 | 8.17e-3 | 1.32e-2 | 2.85e-3 | 3.95e-3 | 2.88e-3 | 4.63e-3 | 9.22e-4 | 2.31e-3 | 2.10e-3 | 2.81e-3 |
|  | CFL | 8.46e-4 | 1.27e-3 | 7.93e-4 | **3.85e-4** | 8.23e-3 | **2.52e-4** | **2.36e-4** | 4.16e-4 | 3.82e-4 | **1.09e-4** | 2.53e-4 | **2.03e-4** | **2.64e-4** |
| 21-30 | Baseline | 1.62e-2 | 3.59e-2 | 6.94e-3 | 1.32e-2 | 3.37e-2 | 6.01e-3 | 6.18e-3 | 6.18e-3 | 9.09e-3 | 1.86e-3 | 4.98e-3 | 3.60e-3 | 5.41e-3 |
|  | CFL | 7.85e-4 | 1.22e-3 | 1.30e-3 | 7.76e-4 | 1.02e-3 | 3.81e-4 | 3.82e-4 | 4.17e-4 | 4.37e-4 | 1.34e-4 | 2.93e-4 | 2.33e-4 | 3.25e-4 |
| 31-40 | Baseline | 1.10e-2 | 1.88e-2 | 2.68e-2 | 9.97e-3 | 1.67e-2 | 3.62e-3 | 4.88e-3 | 4.11e-3 | 5.18e-3 | 1.03e-3 | 2.76e-3 | 2.22e-3 | 3.40e-3 |
|  | CFL | **7.16e-4** | **7.69e-4** | **7.34e-4** | 6.34e-4 | **7.13e-4** | 3.57e-4 | 2.50e-4 | **2.95e-4** | **3.28e-4** | 1.39e-4 | 3.36e-4 | 2.94e-4 | 2.86e-4 |

