# OpenReview forum: "Understanding the Generalization of In-Context Learning in Transformers: An Empirical Study"
_ICLR.cc/2025/Conference — ICLR 2025 Poster_

### Official Review · Reviewer_Htfn · 2024-10-25

**Soundness:** 2
**Presentation:** 3
**Contribution:** 2
**Rating:** 6
**Confidence:** 4

**Summary:**

The present work studies in-context learning (ICL) generalization. The authors’ ICL problems considered sinusoidal functions with varying frequencies and amplitudes (Eq. 2) as basis functions, and additive, multiplicative, or compositional combinations of two of such sinusoidal basis functions (Eq. 3). They considered 3 axes for ICL generalization:
* Inter-problem generalization: Can models trained on the basis functions alone generalize to additive, multiplicative, or compositional combinations of functions?
* Intra-problem generalization: Can a model trained on some functions generalize to other, functionally very distinct functions (e.g., combinations of functions with different frequencies)?
* Intra-task generalization: Can a model trained on some functions generalize to functionally similar functions (e.g., only changes in amplitude that don’t change the x-positions of characteristic points like minima)?

Throughout the experiments, the authors found that transformers show intra-task and intra-problem but lacked inter-problem generalization (Secs. 3 & 4). They partially verified that these findings on synthetic data also hold on real data (Sec. 4). They conclude that task diversity is essential for improved ICL generalization.

**Strengths:**

* **S1**: The work’s systematic setup across the three axes is very well-designed (Secs. 2.2 & 2.3). The real-word experiments (Sec. 4) are also set up sensibly.

* **S2**: The results for compositional combinations are validated on real data and commonly used LLMs. Findings and conclusions from the synthetic experiments from Sec. 3.3 also hold on real data for compositional tasks.

* **S3**: The paper is generally well-written and clear (except the minor points from the questions/suggestions below).

* **S4**: The paper studies a problem that I believe is relevant to the research community. Understanding when ICL will generalize and when it will not is an important research question.

**Weaknesses:**

* **W1** (also see **Q4**): The paper claims that “transformers lack inter-problem generalization with ICL” (l. 19), yet Fig. 2 left or Fig. 3 left show that transformers generalize to y-axis shifted sinusoids (to a certain extent). This seems contradictory. A more nuanced analysis of when transformers can and cannot generalize across ICL problems would strengthen the paper.

* **W2**: The finding that task diversity aids inter-problem generalization is neither novel nor surprising. This is a known reason, e.g., for training LLMs on large, diverse text corpora, as diversity is known to be a key ingredient for generalization. Other popular research areas (e.g., model robustness) also found that data is a key factor. That this is also the case for ICL generalization is expected.

* **W3**: The paper does not offer new insights when transformers do or do not generalize in ICL, particularly in inter-problem generalization. For instance, I think it is clear that the baseline transformer won’t generalize to convex combinations with basis functions with different frequencies (e.g., $sin(x)+sin(2x)$) when trained only on sinusoidals with the same frequency, as these combinations are not well-supported by the training data.

* **W4**: It seems that the combined ICL problems in Eq. 3 will have function values outside the range of those in models trained only on the basis function from Eq. 2. However, this may also be a misunderstanding, as the values for $\phi$ in the experiments are not provided (see **Q2**).

* **W5**: The experiments on convex combinations (Sec. 3.1) and product combinations (Sec. 3.2) are unrelated to the real-word experiments in Sec. 4. It is unclear whether the findings on the synthetic data also hold in the real-world setting for these combinations.

* **W6**: Code is not provided and certain details are missing, e.g., range of $\phi$. Providing code or ensuring all needed experimental details are provided would enhance the paper’s reproducibility.

**Questions:**

* **Q1**: What is the meaning of the superscripts in Eq. 2 and other equations? I.e., what does $(0)$ mean? It appears that it is always the same throughout the paper and seems unnecessary (e.g., superscripts are absent in Eq. 3).

* **Q2**: What is the range of $\phi$ in the experiments?

* **Q3**: Are the GPT-2 models trained with permutation invariance? If either yes or no, why? I think it’d be meaningful to train it with permutation invariance, as it better fits the considered sinusoidal ICL problems (Eqs. 2 & 3).

* **Q4**: How are test functions generated, e.g., in Fig. 2? For example, Fig. 2 left (similar Fig. 3 left) shows $f(x)=sin(x)-0.5$. However, this is different from the functions defined in Eqs. 2 & 3 or Fig. 1, e.g., $\mathcal{F}^{(0)}_1=\lbrace\phi sin(x): \phi\in\mathbb{R}\rbrace$. Importantly, that may influence the results. This could impact the results, as seen in Fig. 2 (second plot from the left), where both methods closely follow the sinusoidal shape but appear shifted.

* **Q5**: l. 255-258: should it be Baseline instead of ComFuncLearner? If not, could you please clarify what is meant in that sentence?

* **Q6**: Do the findings from Sec. 3.1 also hold when we train the ComFuncLearner on the convex combination $\mathcal{F}_1^{(0)}+\mathcal{F}_2^{(0)}$ instead of $\mathcal{F}_1^{(0)}+\mathcal{F}_3^{(0)}$?

* **Q7**: Why was Llama-3 chosen in Sec. 3.4? The experiments would be more consistent if they also used Llama 2, as in Sec. 4 (as they must for fair comparison with ToolLlama).

* **Q8**: Can you provide results for the mixed translation task (Sec. 4.2)?

* **Q9**: How do the authors account for synonyms/paraphrases in the translation experiments (Sec. 4.2)? I tried to input the English output sentence (“Policies and measures…”, Fig. 10) into different translators and translated them to German and back to English. They often yielded very different sentences but the core meaning remained the same. Other metrics than BLUE, e.g., BertScore, may be better fit.

## Suggestions

* **S1**: Most experimental Figs. & Tabs: I suggest writing out the functions (e.g., f(x)=sin(x)) instead of the current format of the subplot titles or table entries. For example, in Tab. 3 & Fig. 4 for compositional combinations, it’s challenging for readers’ to figure out what’s the inner and outer function at a glance.

* **S2**: The paragraph in l. 263-265 appears to address two separate things that could be separated for clarity.

* **S3**: It'd be good to add the description of the ComFuncLearner model variants in l. 364-366 also in Sec. 3.2.

* **S4**: The meaning of the red numbers in Table 4 should be clarified, especially since the number for Baseline+ICL_s under Average_s is not directly computable from the table (likely due to rounding).

---

> ### Author Response · Authors · 2024-11-21
> **Response to Reviewer Htfn (Part 1)**
>
> **W1**: (also see Q4): The paper claims that “transformers lack inter-problem generalization with ICL” (l. 19), yet Fig. 2 left or Fig. 3 left show that transformers generalize to y-axis shifted sinusoids (to a certain extent). This seems contradictory.
>
> **R1**: Thank you for your suggestion. Our primary focus is on the generalization ability of transformers under the three shift patterns we investigate, as outlined in Section 2.2. These patterns, namely inter-problem, intra-problem, and inter-task, represent a decreasing order of generalization difficulty. Our findings indicate that transformers can achieve intra-task and intra-problem generalization while fail in inter-problem generalization.
>
> Regarding Figures 2 and 3, we have clarified the research questions addressed by the different parts of the figures. As stated in Lines 230-233, "The performance of the Baseline model on base function classes answers Question 2.3 on intra-task generalization, while its performance on combinations of base function classes with T+ answers Question 2.1 on inter-problem generalization...". Furthermore, the left-hand side portions of Figures 2 and 3 demonstrate the generalization capability of the transformer, which is one of our key conclusions. In these instances, the model encounters the same types of functions during both training and testing, illustrating the strong intra-task generalization ability of transformers.
>
> This observation aligns with our conclusion that "Transformers fail to show inter-problem generalization yet succeed in achieving strong intra-problem generalization during ICL" (Lines 64-66). The left-hand side plots in Figures 2 and 3 represent intra-task generalization within our experimental setup, where the model is tested on the same types of functions it encountered during training. This confirms the generalization capability of transformers in this context, consistent with our findings.
>
> Regarding the y-axis shifts, we provide a detailed description of our data generation mechanism in Appendix B.1, Lines 759-769. To avoid inaccuracies in analyzing the model's generalization ability caused by mutual approximation between base functions after weighting, we ensure that the expected function values of different base functions have significant differences. In the function fitting experiments (Sections 3.1, 3.2, and 3.3), we added a bias to each base function. Specifically, the whole equation for sine and cosine functions is:
>
> $f(x) = \alpha sin(\beta*x) + (-1)^\beta * \beta / 2$.
>
> **W2**: Other popular research areas (e.g., model robustness) also found that data is a key factor. That this is also the case for ICL generalization is expected.
>
> **R2**: It's important to note that we are not simply stating that more or more diverse data generally improves model generalization. Instead, we provide a nuanced analysis of the generalization behavior of transformers in ICL. We categorize different generalization scenarios and investigate specific areas where transformers exhibit limitations and how data design can be used to address these limitations.
>
> Specifically, we find that increasing the diversity of tasks within a problem does not lead to significant improvements in generalization. However, exposing the model to a wider range of problems can broaden the boundaries of its capabilities across different tasks. This detailed analysis is presented in Line 64-78.
>
> **W3**: The paper ... I think it is clear that the baseline transformer won’t generalize to convex combinations with basis functions with different frequencies (e.g., sin(x)+sin(2x)) when trained only on sinusoidals with the same frequency, as these combinations are not well-supported by the training data.
>
> **R3**: We believe there might be a misunderstanding. Please note that our experiments demonstrate the ability of the transformer to generalize to combinations of functions with different frequencies, even when trained only on functions with the same frequency. For instance, as shown in Appendix D.1, Lines 1372-1377, after being exposed to functions like sin(x) + cos(x), the model can generalize to sin(x) + sin(2x).
>
> As you pointed out, this might seem counterintuitive. However, this is precisely what we aim to demonstrate. In our definition (Section 2.3, Lines 155-180), fitting convex combinations of base functions constitutes a single problem, regardless of the frequencies of the individual functions. Different frequencies simply represent different base functions. Therefore, our conclusion is that once the model has been trained on one type of convex combination, it can generalize to other convex combinations, irrespective of the frequencies involved.

---

> ### Author Response · Authors · 2024-11-21
> **Response to Reviewer Htfn (Part 2)**
>
> **W4**: It seems that the combined ICL problems in Eq. 3 will have function values outside the range of those in models trained only on the basis function from Eq. 2. However, this may also be a misunderstanding, as the values for \phi in the experiments are not provided (see Q2).
>
> **R4**: During training, if the sampled combination function yields values exceeding the range of the base functions, we discard that sample. This effectively prevents any OOD issues arising from mismatched value domains.
>
> **W5**: The experiments on convex combinations (Sec. 3.1) and product combinations (Sec. 3.2) are unrelated to the real-word experiments in Sec. 4. It is unclear whether the findings on the synthetic data also hold in the real-world setting for these combinations.
>
> **R5**: The motivation and purpose behind the design of Our synthetic and real-world experiments are identical. The research questions we raise in Sections 2.2 and 2.3, along with our definitions of base functions and function combinations, apply to both the synthetic data in Section 3 and the real-world settings in Section 4.
>
> Trigonometric functions are simply an example we use in our function fitting experiments. Our goal is to analyze the generalization behavior of in-context learning in transformers broadly, and provide guidance for real-world applications. For instance, in Section 4.1, we state: "Mirroring the definitions of generalization problem and task, we define single-API calls as the basic generalization problem $\mathcal T_{base}$, and the combination of APIs as the composite generalization problem $\mathcal T_{com}$" (Line 423 - Line 431 ). Here, single-API calls correspond to the base functions in Section 3. Similarly, in Section 4.2, translation tasks between single languages, such as En2De, also correspond to the base function fitting in Section 3.
>
> **W6**: Code is not provided and certain details are missing.
>
> **R6**: Thank you for your suggestion. To ensure reproducibility, we have open-sourced our code at https://anonymous.4open.science/r/Generalization-of-Transformers-8227/.
>
> **Q1**: What is the meaning of the superscripts in Eq. 2 and other equations? I.e., what does (0) mean? It appears that it is always the same throughout the paper and seems unnecessary (e.g., superscripts are absent in Eq. 3).
>
> **R1**: The superscript 0 referring to the trigonometric family of functions is simply one instance of the function set F. Our experiments and considerations extend far beyond trigonometric functions, as described in Lines 143-145.
>
> Our research is not limited to trigonometric functions. For example, in Section 4.1, Lines 426-429, we state: "Mirroring the definitions of generalization problem and task, we define single-API calls as the basic generalization problem $\mathcal T_{base}$, and the combination of APIs as the composite generalization problem $\mathcal T_{com}$." This demonstrates the broader applicability of our framework.
>
> Furthermore, we also consider fitting other families of functions. As shown in Appendix D5, Lines 1508-1654, we explore fitting more complex functions, further demonstrating the generalizability of our findings.
>
> **Q2**: What is the range of ϕ in the experiments?
>
> **R2**: For the weight function $\phi$, we initially sample from a standard normal distribution $N(0, 1)$ and clip it to the range $[-1, 1]$. We simply discard combination function samples with function values exceeding the range of the base functions.
>
> **Q3**: Are the GPT-2 models trained with permutation invariance?
>
> **R3**: Yes, we train GPT-2 with permutation invariance because the order of samples presented to the model does not affect the underlying function being learned.
>
> **Q4**: How are test functions generated.
>
> **R4**: Please refer to the response to **W1**.
>
> **Q5**: l. 255-258: should it be Baseline instead of ComFuncLearner? If not, could you please clarify what is meant in that sentence?
>
> **R5**: We believe "ComFuncLearner" is indeed the correct term here, and there is no error. This passage provides an analysis of the experimental results. We observe that ComFuncLearner, after being trained on only the base functions and one type of convex combination of functions, can generalize to other unseen convex combinations.
>
> For example, in our experiment, ComFuncLearner is trained on the following function types: {sin(x), cos(x), sin(2x), cos(2x), α*sin(x)+(1-α)*sin(2x)}, where α ∈ (0,1). However, it successfully generalizes to unseen convex combinations like sin(2x)+cos(2x), sin(x)+cos(2x), etc.
>
> Conversely, the baseline model only demonstrates fitting ability on the base functions. This aligns with our findings: the transformer model exhibits intra-problem generalization but fails to achieve inter-problem generalization.

---

> ### Author Response · Authors · 2024-11-21
> **Response to Reviewer Htfn (Part 3)**
>
> **Q6**: Do the findings from Sec. 3.1 also hold when we train the ComFuncLearner on the convex combination F1(0)+F2(0) instead of F1(0)+F3(0)?
>
> **R6**: Yes, the conclusion still holds. In the original manuscript, we presented the results of ComFuncLearner trained on F2(0)+F4(0) in Appendix D.1, Lines 1372-Line 1376. Following your suggestion, we conducted additional experiments with F1(0)+F2(0), and the results are shown in the table below. We have also updated these experimental results in the appendix, as detailed in Appendix D.1, Lines 1376-1377.
>
> | Range   | Model     | s(x)    | s(2x)    | c(2x)    | Mean$\rm{_B}$ | s(x)\&c(x) | s(x)\&c(2x) | c(x)\&c(2x) | s(2x)\&c(2x) | Com$_{\text{All}}$ | Mean$\rm{_C}$ |
> |---------|-----------|----------|----------|----------|----------|-------------|-------------|-------------|-------------|-------------------|---------------|
> | **1-10** | Baseline  | 5.21e-2| 8.29e-2  | 2.18e-2  | 6.31e-2       | 1.41e-2    | 2.07e-2     | 4.85e-3     | 1.33e-2     | 2.02e-2           | 1.36e-2       |
> |         | CFL       | 8.64e-2  | 2.37e-1  | 8.76e-2  | 3.95e-1       | 7.03e-3    | 7.66e-3          | 1.99e-3     | 1.82e-3     | 1.58e-3           | 4.31e-3       |
> | **11-20**| Baseline  | 7.87e-4   | 7.01e-4  | 5.32e-4  | 1.66e-3       | 2.64e-3    |  5.67e-3      | 4.33e-5     | 5.03e-3     | 8.90e-3           | 4.38e-3       |
> |         | CFL       | 4.36e-4  | **2.98e-4** | 1.81e-4  | 6.85e-4    | 3.99e-5    |  4.28e-5          | 1.17e-5     | 1.71e-5     | 1.83e-5           | 2.73e-5       |
> | **21-30**| Baseline  | 4.42e-4    | 4.66e-4  | 4.00e-4  | 1.60e-3       | 2.73e-3    | 5.18e-3      | 4.30e-5     | 5.27e-3     | 7.97e-3           | 4.12e-3       |
> |         | CFL       | **1.72e-4**  | 4.16e-4  | 1.79e-4 | 7.35e-4 | **3.41e-5** |  2.94e-5        | **1.13e-5** | **1.52e-5** | 1.62e-5           | 2.27e-5       |
> | **31-40**| Baseline  | 4.53e-4    | 5.14e-4  | 5.63e-4  | 1.14e-3       | 2.43e-3    | 5.47e-3     | 4.53e-5     | 4.95e-3     | 8.52e-3           | 4.10e-3       |
> |         | CFL       | 1.91e-4  | 3.36e-4  | **1.78e-4** | **6.29e-4** | 3.79e-5    |  **2.90e-5**  | 1.14e-5     | 1.56e-5     | **1.60e-5**       | **2.20e-5**   |
>
> ***
>
> **Q7**: Why was Llama-3 chosen in Sec. 3.4? The experiments would be more consistent if they also used Llama 2, as in Sec. 4 (as they must for fair comparison with ToolLlama).
>
> **R7**: We chose LLaMA-3 because it represented the state-of-the-art base model at the time of our study. Our intention was to verify that our conclusions hold even for advanced models pre-trained on large-scale datasets.
>
> To further validate our findings on LLaMA-2 and maintain consistency in our experimental setup, we conducted additional experiments using LLaMA-2. The results are presented in the table below. We have also updated the paper with these results in Appendix D.2, Lines 1387-1397.
>
> **Q8**: Can you provide results for the mixed translation task (Sec. 4.2)?
>
> **R8**: Thank you for your suggestion. The complete results are shown in the table below. We have also updated the paper with these results in Appendix B.5, Lines 1026-1042.
> | Model                  | mix2en  | mix2de  | de2en   | en2de   |
> |------------------------|---------|---------|---------|---------|
> | Pretrained             | 28.06   | 24.26   | 18.05   | 17.24   |
> | BL$_{\text{e2d}}$      | 32.19   | 27.55   | 23.45   | 24.81   |
> | BL$_{\text{d2e}}$      | 33.31   | 28.10   | 25.94   | 22.37   |
> | CFL$_{\text{m2d}}$     | 34.25   | 31.68   | 23.32   | 22.15   |
> | CFL$_{\text{m2e}}$     | 35.74   | 30.89   | 23.10   | 21.83   |
> | CFL$_{\text{e2d}}$ | 37.02   | --      | 24.36   | --      |
> | CFL$_{\text{d2e}}$| --      | 31.83   | --      | 22.37   |
> | BL$_{\text{e2d}}$+ICL  | 33.57   | 28.45   | 25.02   | **25.78** |
> | BL$_{\text{d2e}}$+ICL  | 34.74   | 29.12   | **26.73** | 23.02   |
> | CFL$_{\text{m2d}}$+ICL     | 34.98   | 32.35   | 23.47   | 22.74   |
> | CFL$_{\text{m2e}}$+ICL     | 36.27   | 31.66   | 23.59   | 22.29   |
> | CFL$_{\text{e2d}}$+ICL | **37.69** | --    | 24.36   | --      |
> | CFL$_{\text{d2e}}$+ICL | --      | **32.65** | --    | 22.45   |
>
> ***

---

> ### Author Response · Authors · 2024-11-21
> **Response to Reviewer Htfn (Part 4)**
>
> **Q9**: How do the authors account for synonyms/paraphrases in the translation experiments (Sec. 4.2)?
>
> **R9**: Thank you for your suggestion. We adopted additional metrics beyond BLEU, including BertScore and GPT-4o scoring, to assess the accuracy of the model's output. The detailed experimental setup is described in Appendix B.5 Line 1054 - Line 1133. The experimental results are shown in the table below. Our conclusions remain valid.
>
> Bert score:
> | Model                  | mix2en   | mix2de   | de2en   | en2de   |
> |------------------------|----------|----------|---------|---------|
> | Pretrained             | 91.23\%  | 83.15\%  | 69.90\% | 70.32\% |
> | BL$_{\text{e2d}}$      | 91.97\%  | 84.23\%  | 71.46\% | 72.57\% |
> | BL$_{\text{d2e}}$      | 92.83\%  | 84.21\%  | 73.42\% | 71.24\% |
> | CFL$_{\text{m2d}}$     | 92.10\%  | 85.69\%  | 71.13\% | 71.74\% |
> | CFL$_{\text{m2e}}$     | 92.55\%  | 85.07\%  | 71.01\% | 71.53\% |
> | CFL$_{\text{e2d}}$     | 92.74\%  | --       | 73.93\% | --      |
> | CFL$_{\text{d2e}}$     | --       | 85.60\%  | --      | 73.11\% |
> | BL$_{\text{e2d}}$+ICL  | 92.16\%  | 84.67\%  | 71.93\% | 73.14\% |
> | BL$_{\text{d2e}}$+ICL  | **93.45\%** | 84.85\%  | **73.85\%** | 71.80\% |
> | CFL$_{\text{m2d}}$+ICL | 92.68\%  | **86.58\%** | 71.39\% | 72.00\% |
> | CFL$_{\text{m2e}}$+ICL | 93.38\%  | 85.76\%  | 71.44\% | 71.98\% |
> | CFL$_{\text{e2d}}$+ICL | 93.29\%  | --       | 74.01\% | --      |
> | CFL$_{\text{d2e}}$+ICL | --       | 86.03\%  | --      | **73.56\%** |
>
>
> gpt4o eval:
> | Model                  | mix2en | mix2de | de2en  | en2de  |
> |------------------------|--------|--------|--------|--------|
> | Pretrained             | 8.58   | 8.12   | 6.85   | 6.57   |
> | BL$_{\text{e2d}}$      | 9.04   | 8.65   | 7.84   | 8.21   |
> | BL$_{\text{d2e}}$      | 9.12   | 8.40   | 8.38   | 8.02   |
> | CFL$_{\text{m2d}}$     | 8.98   | 9.04   | 7.65   | 7.92   |
> | CFL$_{\text{m2e}}$     | 9.24   | 8.59   | 7.59   | 7.85   |
> | CFL$_{\text{e2d}}$ | 9.15   | --     | 8.36   | --     |
> | CFL$_{\text{d2e}}$ | --     | 9.11   | --     | 8.28   |
> | BL$_{\text{e2d}}$+ICL  | 9.10   | 8.84   | 8.06   | **8.69** |
> | BL$_{\text{d2e}}$+ICL  | 9.23   | 8.62   | 8.50   | 8.17   |
> | CFL$_{\text{m2d}}$+ICL    | 9.08   | 9.14   | 7.89   | 8.21   |
> | CFL$_{\text{m2e}}$+ICL     | **9.26** | 8.84 | 7.77   | 8.10   |
> | CFL$_{\text{e2d}}$+ICL | 9.22   | --     | **8.76** | --     |
> | CFL$_{\text{d2e}}$+ICL | --     | **9.31** | --   | 8.34   |
>
> **S1**: Most experimental Figs. & Tabs: ** I suggest writing out the functions.
>
> **R1**: Thank you for your suggestion. We have specified the exact forms of the composite functions in the relevant section, as detailed in Figure 2, 3, 4, 5 in the paper.
>
> **S2**: The paragraph in l. 263-265 appears to address two separate things that could be separated for clarity.
>
> **R2**: Thank you for pointing this out. We have revised the statement in Line 263 - Line 265 as follows:
>
> "For a detailed experimental setup, please refer to Appendix B.1. Except for convex combinations, we also observe remarkably similar conclusions regarding the model's ability to learn multiplicative combinations of functions as shown in the following section. "
>
> **S3**: It'd be good to add the description of the ComFuncLearner model variants in l. 364-366 also in Sec. 3.2.
>
> **R3**: We apologize, but we did not fully understand your point. Could you please elaborate further? Thank you for your clarification.
>
> **S4**: The meaning of the red numbers in Table 4 should be clarified, especially since the number for Baseline+ICL_s under Average_s is not directly computable from the table (likely due to rounding).
>
> **R4**: We have indicated in the table caption that "The difference highlighted refers to the improvement of ICL." The improvement of 1.8 for Baseline + ICL_s over Baseline that you mentioned is indeed due to rounding. Specifically, the red number means improvement over its corresponding baseline. Baseline + ICL -> Baseline, CFL + ICL -> CFL. The sum of previous items is 89.6, and 89.6 / 3 = 29.866. And we discarded the fractional part instead of rounding it to 29.9.

---

> > ### Comment · Reviewer_Htfn · 2024-11-24
> >
> > I would like to thank the authors for their thorough responses to my initial review and those of the other reviewers. Specifically, I want to thank the authors for their clarifications, providing anonymous code, and the extended evaluations for the translation task. Additionally, I thank the authors for incorporating experiments on non-transformer architectures, as suggested by reviewer udL3.
> >
> > That said, I still have a few remaining questions and concerns:
> >
> > * Regarding **W3** & **Q6**: Thank you for pointing me to Appendix D.1 and for the additional results for $\mathcal{F}_1^{(0)}+\mathcal{F}_2^{(0)}$. However, I find the experimental results (Fig, 18, Tab. 14) somewhat surprising, since any linear combination $\phi_1 sin(x) + \phi_2 cos(x)$ can be equivalently expressed as a single sinusoidal function. Thus, I’d have expected the ComFuncLearner to perform similarly to the baseline model, which was trained solely on single sinusoidal functions. Instead, the results show that ComFuncLearner generalizes significantly better. I’d appreciate a more detailed explanation for this result.
> > * Regarding **W3** & **Q6** ctd: Beyond the above, I’m skeptical of the generality of the claim that “it can generalize to other convex combinations, irrespective of the frequencies”. If the authors assert that the model generalizes to *any* convex combination of base functions, this would include combinations like $sin(42x)+cos(27x)$, which involves vastly different frequencies of the basis functions. If the authors indeed mean that, they would need to provide evidence for that. Otherwise, I suggest revising the claim to make it more accurate.
> > * Regarding **W1**: It would enhance clarity to include certain details, such as the bias term, directly in the main text of the paper. I could not find where the bias term is explicitly mentioned, either in the main text or the supplementary material. I strongly recommend adding this information for completeness.
> > * Regarding **S1**: I appreciate that the authors have explicitly written out the functions in the figures. However, this detail still seems to be missing from the tables and some appendix figures. Including these descriptions consistently would improve clarity.
> > * Regarding **S3**: I must apologize for overlooking the description of the ComFuncLearner model variants in Section 3.2 in the initial submission.

---

> > > ### Author Response · Authors · 2024-11-26
> > > **Thank you for your positive response!**
> > >
> > > We greatly appreciate you taking the time to read our response and for your positive feedback and constructive suggestions!  We have provided responses to your additional questions below.
> > >
> > > **Q1**:  I find the experimental results (Fig, 18, Tab. 14) somewhat surprising.
> > >
> > > **R1**: It's true that sin(x) + cos(x) can be replaced by a single sinusoidal function, but this introduces a phase shift in the function (specifically,
> > > $a\* sin(x) + b \* cos(x) = \sqrt{(a^2 + b^2)}\*sin(x + arctan(b/a))$
> > > ). This phase shift is a completely unfamiliar operation for the Baseline model because it has never encountered such a pattern during training. However, ComFuncLearner can learn the pattern of "convex combination" of different functions and generalize this pattern to convex combinations of other functions. Therefore, it is reasonable that there is a significant difference in their performance.
> > >
> > > **Q2**: Beyond the above, I’m skeptical of the generality of the claim that “it can generalize to other convex combinations, irrespective of the frequencies”.
> > >
> > > **R2**: We apologize for the inaccurate description in the rebuttal. We assume that all experiments are conducted within the constraints outlined in Section 2 of the main text. Therefore, a more accurate description of "it can generalize to other convex combinations, irrespective of the frequencies" would be "it can generalize to other convex combinations, irrespective of the frequencies, **as long as the base functions are seen by ComFuncLearner**."
> > >
> > > Consistent with the main text, when testing the model's ability, we ensure that the base functions of the combination are seen by the model. This is because the model can only generalize to other combination patterns by learning the pattern of a certain combination, **not by generalizing from some base functions to unseen base functions**.
> > >
> > > Therefore, as long as ComFuncLearner has seen the base functions sin(42x) and cos(27x), and only the combination of sin(x) and cos(x), it can generalize to the combination of sin(42x) + cos(27x). The experimental results are shown in the table below, which demonstrate that ComFuncLearner can exhibit significantly stronger generalization ability on the convex combination of sin(42x) + cos(27x).
> > >
> > > | Range | Model | s(x) | s(42x) | c(27x) | Mean$\rm{_B}$ | s(x)&c(x) | s(x)&s(42x) | s(x)&c(27x) | c(x)&s(42x) | s(42x)&c(27x) | Com$_{\text{All}}$ | Mean$\rm{_C}$ |
> > > | --- | --- | --- | --- | --- | --- | --- | --- | --- | --- | --- | --- | --- |
> > > | **1-10** | Baseline | **1.56e-3**  | 2.38e-3 | 2.42e-3 | 3.34e-3 | 2.57e-3 | 4.79e-3 | 4.90e-3 | 6.74e-3 |  4.05e-3 | 2.09e-3 | 3.77e-3 |
> > > |  | CFL | 1.83e-3  | 2.12e-3 | 1.75e-3 | **1.73e-3** | 4.93e-4 | 6.54e-4 | 6.33e-4 | 1.00e-3 | 7.09e-4 | 3.84e-4 | 5.91e-4 |
> > > | **11-20** | Baseline | 2.27e-3 | 1.35e-3 | 1.99e-3 | 2.15e-3 | 1.94e-3 | 2.94e-3 | 4.42e-3 |  6.60e-3 | 1.33e-3 | 5.79e-3 | 2.17e-3 | 3.98e-3 |
> > > |  | CFL | 1.75e-3  | 1.99e-3 | 3.08e-3 | 2.67e-3 | 5.21e-4 | 6.97e-4 | 6.17e-4 | 1.06e-3 | 7.45e-4 | 4.20e-4 | 6.22e-4 |
> > > | **21-30** | Baseline | 1.67e-3 | 4.28e-3 | 1.89e-3 | 3.07e-3 | 2.73e-3 | 2.48e-3 | 4.19e-3 | 3.54e-3 | 5.49e-3 | 3.87e-3 | 1.79e-3 | 3.20e-3 |
> > > |  | CFL | 2.28e-3| **1.82e-3** | **1.65e-3** | 2.39e-3 | 4.72e-4 | 7.89e-4 | 6.27e-4 | 1.13e-3 |**6.63e-4** | 3.77e-4 | 6.13e-4 |
> > > | **31-40** | Baseline | 1.69e-3  | 2.03e-3 | 2.15e-3 | 1.98e-3 | 2.88e-3 | 4.14e-3 | 4.65e-3 | 8.35e-3 |  4.91e-3 | 2.46e-3 | 4.12e-3 |
> > > |  | CFL | 1.78e-3 | 2.04e-3 | 1.76e-3 | 2.31e-3 | **4.64e-4** | **5.92e-4** | **5.76e-4** | **8.66e-4** |  9.73e-4 | **3.69e-4** | **5.86e-4** |
> > > ***
> > >
> > > **Q3**: It would enhance clarity to include certain details, such as the bias term, directly in the main text of the paper.
> > >
> > > **R3**: Thank you for pointing this out. We have added a description of the bias term in Appendix B.1, lines 759-769, as follows:
> > >
> > > To avoid inaccuracies in analyzing the model's generalization ability caused by mutual approximation between base functions after weighting, we ensure that the expected function values of different base functions have significant differences. In the function fitting experiments (Sections 3.1, 3.2, and 3.3), we added a bias to each base function. Specifically, the whole equations for sine and cosine functions are:
> > >
> > > $f(x) = \phi sin(\beta*x)+(-1)^\beta * \beta/2$
> > >
> > > $f(x) = \phi cos(\beta*x)+(-1)^\beta * \beta/2$
> > >
> > > **Q4**: This detail still seems to be missing from the tables and some appendix figures.
> > >
> > > **R4**: Thank you for pointing this out. However, the tables in the paper usually present the average values of multiple test samples. The data in each table represents the average performance over 128 randomly sampled base functions, so we cannot provide the exact function expressions.
> > >
> > > In the appendix, we have added function formulas to most figures where specific function forms can be provided. Due to time constraints, the update is not yet complete, and we will update the remaining figures in the next version.

---

> > > ### Author Response · Authors · 2024-11-27
> > >
> > > Thank you again for your response and positive feedback! If there are remaining questions or concerns, please let us know and we would like to engage in further discussions!

---

> > > ### Author Response · Authors · 2024-12-01
> > >
> > > Dear Reviewer Htfn,
> > >
> > > We would like to thank you for taking the time to review our paper and provide your valuable feedback! We truly appreciate your insights and look forward to your response.
> > >
> > > In our revisions, we have carefully considered the feedback provided by all reviewers and incorporated a number of changes. We hope these revisions address the points raised by the reviewers and strengthen our contribution.  We are happy to address any further questions or concerns you may have during the remaining days of the discussion period.  Please don't hesitate to reach out if you would like to discuss any aspect of the paper in more detail.
> > >
> > > Thank you again for your time and efforts in the review process!

---

> > > > ### Comment · Reviewer_Htfn · 2024-12-02
> > > >
> > > > Thank you for your response and additional comments.
> > > >
> > > > Phase shift is indeed something that the model has not seen during training. However, it still remains unclear to me *why* it helps the model to generalize to convex combinations like $sin(x)+sin(2x)$. I find this very unintuitive, as the function of $sin(x)+sin(2x)$ looks very different to the simple sinusoidal functions seen during training (when using $sin(x)+cos(x)$). In other words, this seems like a very out-of-distribution case and it’s surprising that transformers work here. I’ve read the paper again and, unfortunately, found no explanation or even just hypothesis that (may) explain(s) why transformers generalize to other convex combinations that may look very different to the ones during training.
> > > >
> > > >
> > > > That said, I find the experimental findings intriguing. While it might be asking too much of the authors to provide explanations or conjectures for their observations, I believe this is a critical component for a paper that aims to deepen our understanding. After long consideration, I’ve decided to increase my score to a borderline accept (6), albeit hesitantly. However, in its current state, I remain uncertain about fully supporting the acceptance of this work and would also be okay with rejecting the work.

---

> > > > > ### Author Response · Authors · 2024-12-03
> > > > >
> > > > > We sincerely appreciate the time and efforts you have invested in reviewing our work, and we value the feedback and positive comments you have provided. We are happy to engage in the following discussions and clarifications regarding your concerns.
> > > > >
> > > > > We understand your surprise at our experimental conclusion: that once a transformer has encountered convex combinations of certain functions, it can generalize to convex combinations of other functions in ICL. This is actually one of the most important conclusions of our paper, which focuses on "the extent to which transformers can generalize in ICL." Our understanding of this conclusion is that as long as the transformer has seen a specific form of combination during training (e.g., the convex combination of sin(x) and sin(2x)), it can acquire the ability to generalize to other convex combinations in ICL (e.g., the convex combination of sin(42x) and cos(27x)). **This ability is independent of whether the curves of these other convex combinations are similar to the curve of the convex combination of sin(x) and sin(2x), which is precisely the Intra-problem generalization we defined in Section 2.**
> > > > >
> > > > > **We believe this may be an ability analogous to human learning and reasoning, as humans, once having learned the concept of "convex combination," would not fail to understand it simply because the curve of the convex combination of functions sin(42x) and cos(27x) does not conform to the curves of the convex combination examples we have seen. We believe the model is not simply fitting the curve, but rather learning how to combine two functions and acquiring the ability to learn the corresponding coefficients in ICL, just as we have proven in Section 4 that the transformer can learn the ability to combine different tools and the ability to mix different languages. Therefore, what we are trying to verify through experiments is precisely whether the transformer has the ability to learn an extrapolatable pattern like humans, rather than being limited to similar function curves.**
> > > > >
> > > > > Some theoretical papers also discuss the generalization problem of transformer ICL. For example, when there are differences between the samples in ICL and the training data, it can be proven that transformers can still achieve generalization under certain circumstances. Sang Michael Xie et al. [1] prove that when the pretraining distribution is a mixture of HMMs, the transformer can still generalize to new tasks in ICL with a distribution mismatch between prompts and pretraining data. However, theoretical proof is not the main focus of this paper. We leave theoretical explanations for our findings in future work.
> > > > >
> > > > > [1] Xie, Sang Michael, Aditi Raghunathan, Percy Liang, and Tengyu Ma. "An explanation of in-context learning as implicit bayesian inference." arXiv preprint arXiv:2111.02080 (2021).

---

### Official Review · Reviewer_XdrE · 2024-11-03

**Soundness:** 3
**Presentation:** 2
**Contribution:** 3
**Rating:** 6
**Confidence:** 3

**Summary:**

This work investigates the generalization boundary of transformers' in-context learning.

The authors propose systematically exploring this problem through three dimensions: (i) inter-problem: generalization on problem combination; (ii) intra-problem: generalization across problem combination with guidance; and (iii) intra-task: in-domain generalization.

They conduct experiments on function fitting, API calling, and translation and shed light on strategically designing training data by leveraging the intra-problem and intra-task generalization of ICL.

**Strengths:**

* **Well-scoped problem formulation**.
This paper studies three different dimensions of the generalization with ICL in well-scoped scenarios with a clearly defined experimental protocol. The task formulation corresponding to each research question is clear and adaptable, making it straightforward to generalize findings across the chosen real-world domains.

* **Comprehensive analysis and extended experiments**.
This paper conducts comprehensive experiments that strongly support their conclusions. There are also smart experiment designs in different scenarios to interpret the performance trend across different approaches. I also find the experimental setups sharp and non-trivial in real-world tasks such as tool calls.

* **Convincing and intriguing results**.
The analysis regarding intra-problem generalization and how it works in different tasks provides insights into how data diversity and combinations can augment training for better generalization.

**Weaknesses:**

* **Lack of analysis for a mechanistic understanding of intra-problem generalization**.
While intra-problem generalization appears crucial to ICL, the paper does not sufficiently explore the mechanisms behind this phenomenon. They show that transformers excel in intra-problem generalization across function-combination operators, yet it's unclear how to disentangle the effect of training data in terms of the knowledge derived from functions versus operators. For example, the operator is like a second-order function that can be generalized to higher-order ones, which is particularly pertinent in real-world applications. More in-depth analysis or discussion on this would enhance the work's generalizability and provide greater insights into the workings of intra-problem generalization.

* **Simple task formulation that may not apply to complex natural language tasks**.
While the experiments cover domains like tool usage and provide consistent results, how these findings apply to complex natural language tasks, such as reasoning, is still unclear. For example, results on Llama3 suggest that pre-trained language models may struggle with function fitting despite extensive training corpora. This highlights a domain shift challenge that could be especially relevant for NLP tasks due to the inherent diversity in language. Addressing this limitation or providing additional insights into generalization on natural language tasks could broaden the study’s applicability.

**Questions:**

* The performance improvement on simple tasks when incorporating combination data is intriguing.
    - To ensure a fair comparison, did you control the training data size to have an equal number of samples related to the target simple tasks? From my understanding, combination data should also be factored in as they provide supplementary knowledge.
    - Additionally, have you tried a reverse inter- and intra-problem generalization setup to explore how effectively the model can learn directly from combination data? This could help better understand the generalization mechanism across different conditions.

* Regarding the ComFuncLearner setting, did you observe any impact of data point order, such as when using a curriculum learning schedule? It seems feasible that learning samples from easy to hard could affect the model's performance, particularly for complex generalization tasks.

* Why choose $\mathcal{F}_5^{(0)}$ to be included in Baseline? It seems unrelated to the functions of interest and may hinder the performance.

---

> ### Author Response · Authors · 2024-11-21
> **Response to Reviewer XdrE (Part 1)**
>
> **W1**: More in-depth analysis or discussion on this would enhance the work's generalizability and provide greater insights into the workings of intra-problem generalization.
>
> **R1**: Thank you for your suggestion. We have explored the practical implications of our findings and how they can be applied to improve ICL generalization in Section 4.1 and Section 4.2. We believe that identifying base tasks within different domains and exposing the model to diverse combinations of these base tasks is crucial for improving generalization in real-world applications. For instance, in API call learning (as shown in Section 4.1), training on a small set of API combinations can significantly enhance the model's ability to generalize to unseen combinations. Here, we consider individual API calls as analogous to base functions, and API combinations as operators.
>
> However, we acknowledge the difficulty in establishing a universal definition of base tasks and operators applicable to all downstream tasks. This requires task-specific analysis. Our work primarily provides intuitive guidance and suggestions on this foundational level. Further investigation into the interplay between base tasks and operators within specific downstream tasks is an important direction for future work.
>
> **W2**: Simple task formulation that may not apply to complex natural language tasks
>
> **R2**: Thank you for your suggestion. To better align with real-world applications, such as common NLP tasks, we investigated the inter / intra-problem generalization in an NLP translation task, as detailed in Section 4.2 (Line 464-Line 513). Similar to the function fitting tasks, we trained models from random initialization to avoid potential overlap between pre-training corpora and the test data.
>
> As described in Line 506-Line 514, our findings on this task indicate that: "We observe that on simple tasks, such as translating sentences composed of a single language into another single language with simple sentence structures and shorter lengths, the improvement brought by ICL is less significant, and the gap between ComFuncLearner and Baseline is not substantial. However, on hard tasks, such as those involving mixed-language input or complex sentence structures, ICL can bring more significant improvements to ComFuncLearner, while the Baseline shows considerably less improvement. This indicates that during the training phase, if the model has been exposed to a wider range of problems, it significantly raises the upper limit of improvement achievable through ICL, thereby realizing generalization capabilities on complex tasks."
>
> This observation is consistent with our findings in the function-fitting tasks. Therefore, the experiments presented in this paper offer valuable guidance for real-world applications.
>
> **Q1**: To ensure a fair comparison, did you control the training data size to have an equal number of samples related to the target simple tasks?
>
> **R1**: Yes, to ensure a fair comparison, we trained both the baseline and ComFuncLearner using the same amount of training data, including the combined data, as described in Line 313 - Line 315.
>
> While maintaining the same total amount of training data, we explored two different approaches for selecting the training functions for the baseline:
>
> 1. **Increased samples from the same base functions:** The baseline was trained with a larger number of samples drawn from the same 4 base functions used to train ComFuncLearner. Detailed results are shown in Line 1345-Line 1349 in Appendix  D.1.
>
> 2. **Additional base function:** The baseline was trained with an additional base function beyond the 4 used for ComFuncLearner. Detailed results are shown in Line 234-Line 245 in Section 3.1.
>
> | Range   | Model     | s(x)    | s(2x)    | c(2x)    | Mean$\rm{_B}$ | s(x)&c(x) | s(x)&s(2x) | s(x)&c(2x)  | c(x)&c(2x) | s(2x)&c(2x) | Com$_{\text{All}}$ | Mean$\rm{_C}$ |
> |---------|-----------|----------|----------|----------|-------------|----------|----------|----------|---------|----------|-------------|-------------|
> | **1-10** | Baseline  | 1.31e-2   | 1.26e-2  | 4.26e-2  | 2.66e-2     | 1.98e-2  | 4.48e-3  | 1.99e-2   | 5.95e-3  | 1.16e-2  | 1.78e-2      | 1.35e-2     |
> |         | CFL       | 2.63e-1   | 6.46e-2  | 1.81e-1  | 1.89e-1     | 8.01e-2  | 8.02e-3  | 5.34e-2    | 1.38e-2  | 2.69e-2  | 8.63e-3      | 2.93e-2     |
> | **31-40**| Baseline  | 4.60e-4  | **1.56e-4** | 5.44e-4 | **4.84e-4** | 2.24e-3  | **3.79e-5** | 4.05e-3    | 5.94e-5  | **1.87e-3** | 7.81e-3      | 3.33e-3     |
> |         | CFL       | 7.09e-3   | 3.05e-3  | 1.24e-2  | 7.94e-3     | 2.99e-3  | 2.24e-4  | 3.57e-3  | 5.53e-4  | **3.10e-3** | **4.70e-3**  | 2.69e-3     |
>
> ***
> Both approaches yielded results consistent with our main findings: the transformer model exhibits intra-problem generalization but fails to achieve inter-problem generalization.

---

> ### Author Response · Authors · 2024-11-21
> **Response to Reviewer XdrE (Part 2)**
>
> **Q2**: Have you tried a reverse inter- and intra-problem generalization setup to explore how effectively the model can learn directly from combination data?
>
> **R2**: Thank you for your suggestion. To investigate whether the observed inter- and intra-problem generalization behaviors hold when the roles of base functions and their combinations are reversed, we conducted the following supplementary experiment, and have updated our paper with these findings in Appendix D.3, Line 1399-Line 1492.
>
> >**Experimental Setup:**
>
> >In contrast to the experimental setup in the main text, we now explore whether a model exposed to diverse combinations of base functions can generalize to base function fitting tasks.
>
> >The **Baseline model** is trained solely on combinations of 4 functions: \{ $\mathcal{F}^{(0)}_1 + \mathcal{F}^{(0)}_2$, $\mathcal{F}^{(0)}_1 + \mathcal{F}^{(0)}_4$, $\mathcal{F}^{(0)}_2 + \mathcal{F}^{(0)}_3$, $\mathcal{F}^{(0)}_3 + \mathcal{F}^{(0)}_4$ \}.
>
> >The **ComFuncLearner** is trained on the same 4 function combinations as the Baseline model, but is additionally exposed to one base function: \{ $\mathcal{F}^{(0)}_1$ \}.
>
> The results demonstrate that our findings hold even when we reverse the roles of composite functions and base functions, treating function combinations as T0 and the base functions (which can be approximated as combinations of multiple combinations of functions) as T1. This further reinforces our conclusion that transformers with ICL enable intra-problem generalization but not inter-problem generalization.
>
>
> **Q3**: Regarding the ComFuncLearner setting, did you observe any impact of data point order, such as when using a curriculum learning schedule?
>
> **R3**: Thank you for your suggestion. To investigate the influence of data point order and sample difficulty on the generalization ability of the transformer, we designed the following experiments:
>
> For the **ComFuncLearner**, we designed a two-stage training process. In the first 25,000 iterations, the model learns only the base functions. Subsequently, in the next 25,000 iterations, it is exposed to the combined functions. We then compare the performance of this model with a ComFuncLearner trained on a random sequence of samples. The results are presented in the following table. CFL1_CL refers to CFL1 trained with curriculum learning.
>
> | Model        | s(x)      | s(2x)       | c(2x)       | Mean$\rm{_B}$ | s(x)\&c(x) | s(x)\&s(2x) | s(x)\&c(2x)  | c(x)\&c(2x) | s(2x)\&c(2x) | Com$_{\text{All}}$| Mean\(_C\)  |
> |--------------|-------------|-------------|-------------|-------------|-------------|--------------|----------------|--------------|---------------|--------------------|--------------|
> | Baseline     | 2.52e-5    | **1.61e-5** | 3.71e-5     | **2.17e-5** | 2.69e-2     | 7.76e-3      | 2.29e-2         | **8.52e-3**   | 5.21e-3       | 5.06e-2            | 1.98e-2      |
> | CFL1        | **1.82e-5** | 3.65e-5    | **3.07e-5** | 2.31e-5     | **3.16e-3** | **4.99e-5**  | 7.58e-3       | 1.16e-2      | **1.59e-3**   | **9.88e-3**        | **5.24e-3**  |
> | CFL1_CL      | 3.38e-3    | 3.45e-3     | 7.52e-3     | 5.49e-3     | 7.81e-3     | 2.89e-3      | **7.46e-3**   | 1.60e-2      | 8.34e-3       | 4.06e-2          | 9.92e-3      |
>
> ***
> We find that training the model on base functions first, followed by combined functions, does not further improve its generalization ability. In fact, it results in worse generalization performance compared to a model trained with randomly sampled data. This could be attributed to the need for careful tuning of the ratio of iterations dedicated to learning base functions versus combined functions to achieve optimal performance.
>
>
> **Q4**: Why choose $\mathcal F_5^{(0)}$ to be included in Baseline? It seems unrelated to the functions of interest and may hinder the performance.
>
> **R4**: This is to ensure that the baseline and ComFuncLearner are exposed to the same diversity of data. While maintaining the same total amount of training data, we explored two different approaches for selecting the training functions for the baseline. For the comparison with Baseline trained on 4 base functions, please refer to the response to Q1 above.

---

> > ### Comment · Reviewer_XdrE · 2024-11-26
> >
> > Thanks for the detailed response! I maintain the recommendation of acceptance.

---

> > > ### Author Response · Authors · 2024-11-27
> > >
> > > Thank you for your response and recognition of our rebuttal! We would like to engage in further discussions if there are remaining questions or concerns!

---

### Official Review · Reviewer_udL3 · 2024-11-06

**Soundness:** 4
**Presentation:** 4
**Contribution:** 3
**Rating:** 8
**Confidence:** 3

**Summary:**

The paper presents an empirical study which seeks to advance our understanding of the in-context learning abilities of transformers.  It identifies three different types of generalization, depending on the training and test data: inter-problem (ICL of completely unseen problems), intra-problem (ICL of similar tasks) and intra-task generalization (ICL of already seen tasks).
Experimental results are presented on synthetic data, involving trigonometric functions, as well as more realistic settings, namely tool-calling and translation.

**Strengths:**

Pros: Addresses an important problem in a systematic way.; Clearly defined research questions and synthetic data. ;Extensive experiments.; Provides actionable insight for training transformers;

**Weaknesses:**

Cons:

It is not clear to me that the chosen synthetic functions are representative of ICL tasks that we care about, and that the conclusions drawn from them would be applicable to other synthetic functions.

Having a non-transformer baseline would have been useful in order to understand if the observed behaviours can differ, based on the architectures.

**Questions:**

On line 213, you state that you add a third class in order to balance the data for the Baseline model. Why did you not just sample more data from the 4 base function classes?

Do you expect your results to hold for a different class of functions, s.a. polynomials?

Do you expect similar results for non-transformer architectures?

---

> ### Author Response · Authors · 2024-11-21
> **Response to Reviewer udL3 (Part 1)**
>
> **Q1**: Why did you not just sample more data from the 4 base function classes?
>
> **R1**: Thank you for your constructive feedback. Our initial motivation for this experimental setup was to ensure that both models were exposed to the same number of functions, in order to demonstrate that cross-problem generalization cannot be achieved simply by increasing the diversity of tasks within a problem. To verify whether cross-problem generalization can be achieved solely by increasing the amount of data, we conducted additional experiments where the model was trained with more data from only 4 base functions and tested on various function fitting tasks. The results are shown in the following table. The conclusion drawn from these experiments is consistent with the main text: the transformer model only achieves intra-task and intra-problem generalization, but not inter-problem generalization.
>
> Furthermore, we have included the supplementary results in Line 1345-Line 1349 in Appendix  D.1.
>
>  | Range   | Model     | s(x)          | s(2x)    | c(2x)    | Mean$\rm{_B}$ | s(x)&c(x) | s(x)&s(2x) | c(x)&s(2x) | c(x)&c(2x) | s(2x)&c(2x) | Com$_{\text{All}}$ | Mean$\rm{_C}$ |
> |---------|-----------|----------|----------|----------|-------------|----------|----------|----------|----------|----------|-------------|-------------|
> | **1-10** | Baseline  | 1.31e-2    | 1.26e-2  | 4.26e-2  | 2.66e-2     | 1.98e-2  | 4.48e-3  | 1.49e-2  | 5.95e-3  | 1.16e-2  | 1.78e-2      | 1.35e-2     |
> |         | CFL       | 2.63e-1   | 6.46e-2  | 1.81e-1  | 1.89e-1     | 8.01e-2  | 8.02e-3  | 1.39e-2  | 1.38e-2  | 2.69e-2  | 8.63e-3      | 2.93e-2     |
> | **31-40**| Baseline  | **4.60e-4**  | **1.56e-4** | **5.44e-4** | **4.84e-4** | **2.24e-3**  | **3.79e-5** |  7.27e-3  | **5.94e-5**  | **1.87e-3** | 7.81e-3      | 3.33e-3     |
> |         | CFL       | 7.09e-3    | 3.05e-3  | 1.24e-2  | 7.94e-3     | 2.99e-3  | 2.24e-4  | **3.70e-3** | 5.53e-4  | **3.10e-3** | **4.70e-3**  | **2.69e-3**     |
>
> ***
>
> **Q2**: Do you expect your results to hold for a different class of functions, s.a. polynomials?
>
> **R2**: Thank you for your feedback. We have further investigated the generalization ability of the transformer model on other base functions. To ensure significant differences between the functions, we selected the following base functions:
>
>
>    *  $y^{\text{(1)}}_{\text{L}} = \frac{\sqrt{30}}{50} \cdot \mathbf{x} \cdot |\mathbf{w}|$
>    *  $y^{\text{(2)}}_{\text{L}} = \frac{\sqrt{2}}{50} \cdot \left( 3 \mathbf{x}^2 - 25 \right) \cdot \mathbf{w}$
>    *  $y^{\text{(3)}}_{\text{L}} = \frac{\sqrt{70}}{500} \cdot \left( \mathbf{x}^3 - 15 \mathbf{x} \right) \cdot \mathbf{w}$
>    *  $y^{\text{(4)}}_{\text{L}} = \frac{3 \sqrt{10}}{10000} \cdot \left( 7 \mathbf{x}^4 - 150 \mathbf{x}^2 + 375 \right) \cdot \mathbf{w}$
>
> Using these base functions, we trained and tested both the baseline and ComFuncLearner using convex combinations as the function composition method. The results are shown in the following table. We found that using different base functions, as long as there are significant differences between them, leads to conclusions consistent with those presented in the paper. We include this experiment in Appendix D.5.
>
> | Range   | Model     | x        | $\text{x}^2$ | $\text{x}^3$ | $\text{x}^4$ | Mean$\rm{_B}$ | x\&$\text{x}^2$ | x\&$\text{x}^3$ | x\&$\text{x}^4$ | $\text{x}^2$\&$\text{x}^3$ | $\text{x}^2$\&$\text{x}^4$ | $\text{x}^3$\&$\text{x}^4$ | Com$_{\text{All}}$ | Mean$\rm{_C}$ |
> |---------|-----------|----------|--------------|--------------|--------------|---------------|-----------------|-----------------|-----------------|-----------------|-----------------|-----------------|-----------------|---------------|
> | **1-10** | Baseline  | 2.11e-1  | 5.96e-2      | 1.81e-1      | 3.58e-1      | 2.02e-1       | 1.54e-2         | 2.40e-2         | 7.55e-3         | 2.31e-2         | 8.59e-3         | 3.95e-3         | 4.95e-3         | 1.25e-2      |
> |         | CFL       | 2.02e-1  | 5.57e-2      | 1.76e-1      | 2.52e-1      | 1.72e-1       | 1.52e-2         | 1.26e-2         | 7.40e-3         | 1.21e-2         | 8.24e-3         | 3.42e-3         | 4.91e-3         | 9.12e-3      |
> | **31-40**| Baseline  | 5.69e-3  | 4.25e-3      | **1.42e-3**      | **1.84e-2**      | **7.45e-3**   | 7.34e-3         | 6.65e-3         | 3.99e-3         | 8.26e-3         | 4.99e-3         | 7.99e-4         | 4.64e-3         | 4.72e-3      |
> |         | CFL       | **1.91e-2** | **1.43e-2** | 7.72e-3      | 2.37e-2      | 1.62e-2       | **5.42e-3**     | **3.71e-3**         | **3.00e-3**     | **1.85e-3**     | **3.93e-3**     | **6.51e-4**     | **4.11e-3**         | **3.23e-3**   |
>
> ***

---

> ### Author Response · Authors · 2024-11-21
> **Response to Reviewer udL3 (Part 2)**
>
> **Q3**: Do you expect similar results for non-transformer architectures?
>
> **R3**：Thank you for your insightful suggestion. Our main focus was to investigate the generalization ability of transformer models in the context of ICL, and other models were not originally within the scope of our study. However, prompted by your valuable feedback, we conducted further experiments with other model architectures, including MLP and KAN. The results are presented in the tables below.
>
> Results with MLP:
> | Range   | Model     | s(x)    | s(2x)    | c(2x)    | Mean$\rm{_B}$ | s(x)\&c(x) | s(x)\&s(2x) | s(x)\&c(2x) | c(x)\&c(2x) | s(2x)\&c(2x) | Com$_{\text{All}}$ | Mean$\rm{_C}$ |
> |---------|-----------|----------|---------|----------|---------------|------------|-------------|-------------|-------------|----------------|-------------------|---------------|
> | **1-10** | Baseline  | 3.07e-1  | 1.67e-2  | 1.32e-1  | 1.18e-1       | 7.83e-3    | 8.70e-4     | 1.03e-3     | 2.52e-4     | 3.06e-3     | 2.64e-4           | 2.10e-3       |
> |         | CFL       | 5.98e-2    | **1.57e-3**  | 2.16e-2  | 2.11e-2       | 4.42e-4    | 1.59e-4     | 1.86e-4    | 7.21e-5     | 3.78e-4     | 4.61e-5           | 2.24e-4       |
> | **11-20**| Baseline  | 3.00e-1   | 1.01e-2  | 1.47e-1  | 1.17e-1       | 4.59e-3    | 9.85e-4     | 9.64e-4    | 4.16e-4     | 2.91e-3     | 2.70e-4           | 1.93e-3       |
> |         | CFL       | 4.89e-2    | 2.08e-3  | 3.08e-2  | 2.07e-2       | 6.89e-4    | 1.57e-4     | 2.16e-4   | 7.99e-5     | 2.81e-4     | 5.83e-5           | 2.49e-4       |
> | **21-30**| Baseline  | 3.89e-1    | 8.93e-3  | 1.76e-1  | 1.46e-1       | 5.79e-3    | 8.44e-4     | 9.00e-4     | 4.24e-4     | 2.35e-3     | 2.76e-4           | 1.74e-3       |
> |         | CFL       | 4.70e-2    | 1.72e-3  | **1.78e-2**  | 1.71e-2   | 5.90e-4    | 1.67e-4     | 1.91e-4     | 6.67e-5     | 3.28e-4     | **4.41e-5**       | 2.32e-4       |
> | **31-40**| Baseline  | 1.82e-1    | 1.71e-2  | 1.76e-1  | 9.93e-2       | 4.59e-3    | 8.41e-4     | 1.05e-3        | 3.90e-4     | 3.86e-3     | 2.54e-4           | 1.89e-3       |
> |         | CFL       | **3.65e-2**  | 2.13e-3  | 2.48e-2  | **1.61e-2** | **3.23e-4** | **1.42e-4** | **1.78e-4**    | **5.08e-5** | **2.65e-4** | 4.65e-5           | **1.84e-4**   |
> ***
>
> Results with KAN:
> | Range   | Model     | s(x)     | s(2x)    | c(2x)    | Mean$\rm{_B}$ | s(x)\&c(x) | s(x)\&s(2x) | s(x)\&c(2x)  | c(x)\&c(2x) | s(2x)\&c(2x) | Com$_{\text{All}}$ | Mean$\rm{_C}$ |
> |---------|-----------|----------|----------|----------|----------|---------------|------------|------------|-------------|-------------|-------------------|---------------|
> | **1-10** | Baseline  | 1.25e-2  |   | 2.34e-2  | 9.50e-3  | 1.70e-2       | 3.98e-3    | 5.84e-3       | 1.19e-3     | 3.59e-3     | 2.92e-3           | 4.06e-3       |
> |         | CFL       | 7.50e-4  |   | 1.79e-3  | 6.65e-4  | 1.04e-3       | 3.41e-4    | 2.73e-4      | 1.26e-4     | 2.58e-4     | 2.15e-4           | 2.90e-4       |
> | **11-20**| Baseline  | 9.84e-3  |  | 2.31e-2  | 8.17e-3  | 1.32e-2       | 2.85e-3    | 3.95e-3           | 9.22e-4     | 2.31e-3     | 2.10e-3           | 2.81e-3       |
> |         | CFL       | 8.46e-4  |   | 7.93e-4  | **3.85e-4** | 8.23e-3    | **2.52e-4** | **2.36e-4**      | **1.09e-4** | **2.53e-4** | **2.03e-4**       | **2.64e-4**   |
> | **21-30**| Baseline  | 1.62e-2  |  | 6.94e-2  | 1.32e-2  | 3.37e-2       | 6.01e-3    | 6.18e-3         | 1.86e-3     | 4.98e-3     | 3.60e-3           | 5.41e-3       |
> |         | CFL       | 7.85e-4  |   | 1.30e-3  | 7.76e-4  | 1.02e-3       | 3.81e-4    | 3.82e-4        | 1.34e-4     | 2.93e-4     | 2.33e-4           | 3.25e-4       |
> | **31-40**| Baseline  | 1.10e-2  |  | 2.68e-2  | 9.97e-3  | 1.67e-2       | 3.62e-3    | 4.88e-3          | 1.03e-3     | 2.76e-3     | 2.22e-3           | 3.40e-3       |
> |         | CFL       | **7.16e-4** | | **7.34e-4** | 6.34e-4  | **7.13e-4** | 3.57e-4    | 2.50e-4      | 1.39e-4     | 3.36e-4     | 2.94e-4           | **2.86e-4**   |
>
> ***
>
> We are surprised to find that the generalization behavior observed in transformers persists in these other network architectures. This opens up exciting avenues for future research.
>
> It is important to note that this should not be strictly termed "in-context learning" for MLP and KAN. Since these models require fixed input and output dimensions, our experimental setup necessitates feeding 39 (x, f(x)) pairs along with the 40th x as input, and the model predicts only the 40th f(x). This differs from the sequential prediction capability of transformers in ICL. However, we believe that our findings regarding generalization limitations may extend beyond transformers and ICL, potentially applying to other deep learning architectures and tasks.

---

> > ### Comment · Reviewer_udL3 · 2024-11-28
> >
> > I thank the authors for their reply. I maintain my recommendation for acceptance.

---

> > > ### Author Response · Authors · 2024-11-29
> > >
> > > Thank you for your constructive suggestions and positive response! We would like to engage in further discussions if there are remaining questions or concerns!

---

### Meta-Review · Area_Chair_XGsL · 2024-12-18

**Metareview:**

This paper explores how well transformer models can learn from context. it provides systematic investigation of transformers' generalization capability in terms of inter-problem, intra-problem, and intra-task generalization. The study shows that that transformers show intra-task and intra-problem but lacked inter-problem generalization and task diversity is essential for improving ICL generalization.

This is a valuable study that provides key insights into a crucial aspect of in-context learning (ICL). The findings and well-designed experiments are likely to influence future research and spark new ways to use ICL.

That said, based on the feedback received, I recommend the authors provide more details about the study design and strengthen their arguments with additional evidence. This will further enhance the impact and clarity of their work.

**Additional Comments On Reviewer Discussion:**

The main concerns raised about this paper focused on two points:

Are the conclusions justified? Reviewers questioned whether the conclusions drawn from the experiments were fully supported by the evidence.

How broadly applicable are the findings? Reviewers wanted to know if the insights gained from this study could be generalized to other situations.

To address these concerns, the authors have expanded the paper with new experiments and explanations. This strengthens their arguments and provides further evidence to support their original conclusions.

---

### Decision · Program_Chairs · 2025-01-22

Accept (Poster)